# A simple model for the evolution of melt pond coverage on permeable Arctic sea ice

Predrag Popović[1] and Dorian Abbot[1]

[1]University of Chicago, Dept. of Geophysical sciences

*Correspondence to:* Predrag Popović (ppopovic@uchicago.edu)

**Abstract.**

As the melt season progresses, sea ice in the Arctic often becomes permeable enough to allow for nearly complete drainage of meltwater that has collected on the ice surface. Melt ponds that remain after drainage are hydraulically connected to the ocean and correspond to regions of sea ice whose surface is below sea level. We present a simple model for the evolution of melt pond coverage on such permeable sea ice floes in which we allow for spatially varying ice melt rates and assume the whole floe is in hydrostatic balance. The model is represented by two simple ordinary differential equations, where the rate of change of pond coverage depends on the pond coverage. All the physical parameters of the system are summarized by four strengths that control the relative importance of the terms in the equations. The model both fits observations and allows us to understand the behavior of melt ponds in a way that is often not possible with more complex models. Examples of insights we can gain from the model are: (1) the pond growth rate is more sensitive to changes in bare sea ice albedo than changes in pond albedo, (2) ponds grow slower on smoother ice, and (3) ponds respond strongest to freeboard sinking on first year ice and sidewall melting on multiyear ice. We also show that under a global warming scenario, pond coverage would increase, decreasing the overall ice albedo, and leading to ice thinning that is likely comparable to thinning due to direct forcing. Since melt pond coverage is one of the key parameters controlling the albedo of sea ice, understanding the mechanisms that control the distribution of pond coverage will help improve large-scale model parameterizations and sea ice forecasts in a warming climate.

## 1 Introduction

Over the past forty years, Arctic summer sea ice extent has reduced by 50 percent, making it one of the most sensitive indicators of man-made climate change (Serreze and Stroeve, 2015; Stroeve et. al., 2007; Perovich and Richter-Menge, 2009). This rapid decrease is at least partially due to the ice-albedo feedback (Zhang et. al., 2008; Screen and Simmonds, 2010; Perovich et. al., 2007). Moreover, if the ice-albedo feedback is strong enough it could lead to instabilities and abrupt changes in ice coverage in the future (North, 1984; Holland et. al., 2006; Eisenman and Wettlaufer, 2008; Abbot et al., 2011). The albedo of ice is significantly reduced by the presence of melt ponds on

its surface (Eicken et. al., 2004; Perovich and Polashenski, 2012; Yackel et. al., 2000). Therefore, understanding the evolution of melt ponds is essential for understanding the ice-albedo feedback, and consequently, the evolution of Arctic sea ice cover in a warming world. This means that accurate melt pond parameterizations must be incorporated into Global Climate Models (GCMs) to improve their sea ice forecasts (Flocco et. al., 2010; Holland et. al., 2012; Pedersen et. al., 2009). The main difficulties with including accurate melt pond parameterizations in large scale models are that pond evolution is nonlinear and that it is the result of a variety of different physical processes operating on a range of length and time scales. For these reasons, it is important to understand the mechanisms that drive the evolution of melt ponds.

Typically, the evolution of pond coverage on first-year ice proceeds in fairly consistent stages (Polashenski et. al., 2012; Perovich et. al., 2003; Landy et. al., 2014; Webster et. al., 2015). First the ponds grow quickly while the ice is impermeable. Next they drain quickly and pond coverage shrinks as the ice transitions from impermeable to permeable. Then the ponds grow slowly while the ice is permeable and pond water remains at sea level. Finally, the ponds either refreeze or the floe breaks up. The stage when ice is highly permeable is typically the longest, often longer than the first two stages combined. This stage is particularly suitable to model, since the ponds can be assumed to be at sea level and hydraulically connected to the ocean. On multiyear ice, ponds also experience a growth and a drainage stage, but often do not drain to sea level. On some occasions, however, ponds on multiyear ice can drain to sea level as well.

In this paper we will present a simple "0D" model for the evolution of melt pond coverage on sea ice floes. We will assume that ice is permeable, ponds are at sea level and hydraulically connected to the ocean, the whole ice floe is in hydrostatic balance, and different points on the ice surface may melt at different rates. The purpose of our model is: (1) to clarify the roles in the evolution of pond coverage played by energy fluxes, the ice thickness, bulk ice density, ice roughness, and initial pond coverage, (2) to provide a simple, yet accurate, way to estimate the pond coverage as a function of time, (3) to understand the behavior of melt ponds under general environmental conditions, and (4) to investigate different types of qualitative behavior that can arise from differential melting and maintaining hydrostatic balance.

Skyllingstad et. al. (2009) also describe pond growth on permeable ice, but include only pond growth by lateral melt of pond walls. This contrasts with our model, which includes includes pond growth by vertical changes of the topography. Our models are different, but complementary, and we will draw parallels between our two models when discussing the possibility of lateral melt. Aside from Skyllingstad et. al. (2009), previous melt pond modeling efforts include works by Taylor and Feltham (2004), Lüthje et. al. (2006), Scott and Feltham (2010), and Flocco and Feltham (2007), who all created comprehensive models that allowed for more realistic representations of physical processes such as heat and salt balance, and meltwater routing and drainage. The advantage of our

model is its simplicity, which makes it possible to clarify the roles of each of the physical parameters

involved.

This paper is organized in the following way. In section 2 we build a simple model for the evolution of pond coverage. In section 3, we compare the model to observations. In section 4 we discuss realistic values of physical parameters and solve the model numerically. In section 5 we assess the impacts of sea ice roughness and develop a simple parameterization to estimate mean pond

coverage after a certain amount of time without solving the model. In section 6 we analyze the model analytically to gain a better understanding of the factors influencing pond evolution. In section 7 we discuss lateral melt and internal melt combined with effect of density variations. Finally in section 8 we summarize our results and conclude. In appendices A, B, C, and D we discuss some of the more technical aspects of our model.

## 2  Building the simple 0D model

In this section, we build the model for the evolution of melt pond coverage, and then solve it using realistic physical parameters. Before we proceed to build the quantitative model, we will first state the assumptions, and discuss the physical mechanisms driving pond evolution.

### 2.1  Assumptions of the model

Our model focuses on the stage of pond evolution when ice is highly permeable and all the meltwater created can be quickly removed to the ocean. The beginning of this stage can be identified as the point in time when the meltwater on the ice surface has drained to sea level, such that the remaining ponds correspond to places on the ice surface that are below sea level. We will assume that from this point on, the ponds are hydraulically connected with the ocean, and the only way for pond

coverage to increase is for the points on the ice surface which were above sea level to sink or melt below sea level. In reality, ponds can also grow through horizontal melting of their sidewalls. As some observations suggest that this type of growth is small at least on first year ice (Polashenski et. al., 2012; Landy et. al., 2014), we neglect it (see section 7.1 for further discussion). Furthermore, we will assume that all the melt occurs at the surface or the bottom of the ice. We thereby neglect

the possibility of internal melt. We will also assume that ice has a uniform bulk density throughout the vertical column, and we discuss the effects of vertical non-uniformity in bulk density together with effects of internal melt in subsection 7.2. Finally, we will assume that the entire ice floe is in hydrostatic balance.

The main goal of our model is to determine the fraction of the ice surface above sea level that

falls below sea level after some time. Therefore, we focus on the vertical displacements of points on the surface of the ice in response to melt. To this end, we define the ice topography, $s(\boldsymbol{r})$, as the elevation of the ice surface above sea level at the point $\boldsymbol{r}$, and we define melt ponds as those regions

where $s(\boldsymbol{r}) < 0$. There are two main reasons why the topography might change in response to ice melt:

1. First, the topography at a point $\boldsymbol{r}$ at the surface changes when ice at that point melts (Fig. 1a). Here, the rate of change of topography at a point depends only on local characteristics of that particular point. For this reason, we will call this type of motion "local." Points on the surface that melt locally move "downwards," i.e. to lower elevations above sea level.

2. Second, in order to maintain hydrostatic balance, the entire ice surface can shift up or down in response to mass being removed above or below sea level. Since we are assuming that the entire ice floe is in hydrostatic balance, melting any region of ice moves the entire floe as a rigid body (Fig. 1b). For this reason, we will call this type of motion the "rigid body" motion. Melting above sea level induces an upward rigid body motion, whereas melting below sea level induces a downward rigid body motion. An ice floe is not a rigid body, but up to its
flexural wavelength (roughly 30 m on 1.5 m thick ice) we can approximate it as such. As the flexural wavelength is larger than the typical scale of melt ponds (roughly 10m), the rigid body approximation is likely good.

At each point on the ice surface the change in elevation above sea level can be calculated as the sum of these two contributions.

In our model, ponds grow in two ways, "freeboard sinking" and "enhanced melting":

   1. Freeboard sinking represents the average change in freeboard height (average height above sea level of bare ice). In this way the topography of ice above sea level remains unchanged. Freeboard sinking should not be confused with rigid body motion: the average freeboard height always decreases as a response to ice thinning, whereas the rigid body motion can point both
upward and downward depending on whether mass is lost above or below sea level. Both rigid body motion and average local melting contribute to freeboard sinking.

   2. Enhanced melting represents the change in the shape of the topography without changing its average height. Ponds can grow in this way if some regions melt faster than average. Therefore, a positive deviation in the local melt rate can grow ponds. Conversely, a negative deviation in
the local melt rate can slow down or even reverse pond growth. Pond growth only occurs due to topography changes near sea level. Therefore, deviations from the mean melt rate for points high above the sea level do not influence pond evolution since these points are correlated with points close to sea level only through hydrostatic adjustment, which is determined by the average melt rates rather than the deviations from the average.

## 2.2   Equation for the evolution of topography

We now proceed to build the quantitative model of pond evolution. Following the above ideas, we divide the total rate of change of vertical position of the point $\boldsymbol{r}$ on the surface of the ice, $\frac{ds}{dt}(\boldsymbol{r})$, into

a contribution from rigid body motion, $\frac{ds_{\text{rigid body}}}{dt}$, and a contribution from local melting, $\frac{ds_{\text{loc}}}{dt}(\boldsymbol{r})$,

$$\frac{ds}{dt}(\boldsymbol{r}) = \frac{ds_{\text{rigid body}}}{dt} + \frac{ds_{\text{loc}}}{dt}(\boldsymbol{r}) \,. \tag{1}$$

Ice above sea level must hydrostatically balance ice below sea level. We can write this hydrostatic balance as

$$m_{\text{above s. l.}} = \frac{\rho_{\text{w}} - \rho_{\text{i}}}{\rho_{\text{i}}} m_{\text{below s. l.}} \,, \tag{2}$$

where $m_{\text{above s. l.}}$, and $m_{\text{below s. l.}}$ represent the mass of ice above and below sea level, and $\rho_{\text{w}}$, and $\rho_{\text{i}}$ represent the densities of sea water and pure ice. Throughout the paper we use $\rho_{\text{w}} = 1025 \text{ kg m}^{-3}$

and $\rho_{\text{i}} = 916 \text{ kg m}^{-3}$.

The mass above and below sea level can change either because the ice melts or because the floe moves as a rigid body, changing the proportion of ice above and below sea level. Therefore, differentiating Eq. (2) and splitting into melt and rigid body contributions, we find

$$dm_{\text{above s. l.}}^{\text{melt}} + dm_{\text{above s. l.}}^{\text{rigid body}} = \frac{\rho_{\text{w}} - \rho_{\text{i}}}{\rho_{\text{w}}} \left[ dm_{\text{below s. l.}}^{\text{melt}} + dm_{\text{below s. l.}}^{\text{rigid body}} \right] \,, \tag{3}$$

where $dm_{\text{above/below s. l.}}^{\text{melt/rigid body}}$ represent changes in mass above and below sea level due to either ice melting or the entire floe floating up or down.

The mass melted above and below sea level after some time $dt$ is

$$dm_{\text{above s. l.}}^{\text{melt}} = -A_{\text{bi}} \frac{\overline{F}_{\text{bi}}}{l} dt \,,$$

$$dm_{\text{below s. l.}}^{\text{melt}} = -A_{\text{mp}} \frac{\overline{F}_{\text{mp}}}{l} dt - A \frac{\overline{F}_{\text{bot}}}{l} dt \,, \tag{4}$$

where $l = 334 \text{ kJ kg}^{-1}$ is the latent heat of melting, $\overline{F}_{\text{bi}}$ is the total energy flux used for melting bare

ice averaged over all bare ice, $\overline{F}_{\text{mp}}$ is the total energy flux used for melting ponded ice averaged over ponded ice, $\overline{F}_{\text{bot}}$ is the total energy flux used for melting the ice bottom averaged over the ice bottom, $A_{\text{bi}}$, $A_{\text{mp}}$, and $A$ are the area of bare ice, the area of melt ponds, and the area of the entire floe.

Since floating up or down does not change the total mass of the ice, mass changes above and below

sea level due to rigid body motion are equal with an opposite sign, $dm_{\text{above s. l.}}^{\text{rigid body}} = -dm_{\text{below s. l.}}^{\text{rigid body}}$. We can express $dm^{\text{rigid body}}$ in terms of rigid body displacement of the floe as

$$dm_{\text{above s. l.}}^{\text{rigid body}} = \rho_b A_{\text{bi}} ds_{\text{rigid body}} \,,$$

$$dm_{\text{below s. l.}}^{\text{rigid body}} = -\rho_b A_{\text{bi}} ds_{\text{rigid body}} \,, \tag{5}$$

where $\rho_b$ is the bulk ice density. This is the density of sea ice once all the brine has drained and is always less than $\rho_i$. We assume it to be uniform throughout the vertical ice column, but discuss the

effects of vertical variations in $\rho_b$ in section 7.2.

Substituting Eqs. (4) and (5) into Eq. (3), solving for $ds_{\text{rigid body}}$, and differentiating with respect to time, we find the rate of change of surface topography due to rigid body motion to be

$$\frac{ds_{\text{rigid body}}}{dt} = \left[\frac{\rho_{\text{i}}}{\rho_{\text{w}}}\frac{\overline{F}_{\text{bi}}}{l\rho_b}\right] - \left[\frac{\rho_{\text{w}} - \rho_{\text{i}}}{\rho_{\text{w}}}\frac{A_{mp}}{A_{bi}}\frac{\overline{F}_{\text{mp}}}{l\rho_b}\right] - \left[\frac{\rho_{\text{w}} - \rho_{\text{i}}}{\rho_{\text{w}}}\frac{A}{A_{bi}}\frac{\overline{F}_{\text{bot}}}{l\rho_b}\right] . \tag{6}$$

The three terms in large square brackets correspond to topography change due to bare ice melting,
ponded ice melting, and ice bottom melting. Rigid body motion depends only on spatially averaged energy fluxes, which in turn depend on parameters such as the average insolation on the floe, the average albedo, and the average longwave, sensible, latent and bottom heat fluxes. If bare and ponded ice melt only from energy absorbed by the upper surface of the ice, the fluxes $\overline{F}_{\text{bi}}$, and $\overline{F}_{\text{mp}}$ can also be written in terms of albedo as:

$$\overline{F}_{\text{bi}} = (1 - \alpha_{\text{bi}})F_{\text{sol}} + F_{\text{r}} ,$$
$$\overline{F}_{\text{mp}} = (1 - \alpha_{\text{mp}})F_{\text{sol}} + F_{\text{r}} , \tag{7}$$

where $\alpha_{\text{bi}}$ and $\alpha_{\text{mp}}$ are the average albedos of bare and ponded ice, $F_{\text{sol}}$ is the solar flux, and $F_{\text{r}}$ is equal to the sum of net longwave, net sensible, and net latent heat fluxes. This parameterization neglects light transmission, and assumes that all of the energy is deposited in the surface. Much of the variation in albedo of ponded ice is due to the fact that the pond bottom is partially transparent,
and energy is deposited in the ocean instead of directly in the ice. However, this does not make much difference in our model since the energy deposited in the ocean is likely used for melting ice below sea level anyway.

Local displacement, $ds_{\text{loc}}$, quantifies how much the ice surface topography changes as a result of local melt. We can determine the local melt rate from $F_{\text{surf}}(\boldsymbol{r})$, the flux of energy used for melting
the ice surface at a point $\boldsymbol{r}$

$$\frac{ds_{\text{loc}}}{dt}(\boldsymbol{r}) = -\frac{F_{\text{surf}}(\boldsymbol{r})}{l\rho_b} , \tag{8}$$

where the positive direction is defined as upwards. The local flux depends on parameters such as the local albedo, the local insolation, the local longwave, sensible and latent heat fluxes, and the angle between ice and incoming radiation at that point.
The flux $F_{\text{surf}}(\boldsymbol{r})$ averaged over all the points on the surface of the ice above sea level equals $\overline{F}_{\text{bi}}$

$$< F_{\text{surf}}(\boldsymbol{r}) > = \overline{F}_{\text{bi}} , \tag{9}$$

where $< ... >$ represents averaging over all the points on bare ice. For this reason, we will parameterize the rate of local melting as

$$\frac{ds_{\text{loc}}}{dt}(\boldsymbol{r}) = -k(\boldsymbol{r})\frac{\overline{F}_{\text{bi}}}{l\rho_b} , \tag{10}$$

where $k(\boldsymbol{r})$ is a non-dimensional number that quantifies the deviation of the melt rate at the point $\boldsymbol{r}$ from the mean melt rate of the bare ice surface, which depends on the detailed conditions of ice and

its environment. The parameter $k$ could be either greater than or less than one. Here we will take $k$ to be constant in time, but in reality it need not be. Finally, according to Eq. (1) we add Eq. (6) and Eq. (10) to get the equation for the evolution of the bare ice topography. We express this in terms of melt pond fraction, $x \equiv \frac{A_{mp}}{A}$

$$
\frac{ds}{dt}(\boldsymbol{r}) = -\left[(k(\boldsymbol{r})-1)\frac{\overline{F}_{bi}}{l\rho_b}\right] -
$$
$$
- \left[\frac{\rho_w - \rho_i}{\rho_w}\frac{1}{l\rho_b}\left(\overline{F}_{bi} + \frac{x}{1-x}\overline{F}_{mp} + \frac{1}{1-x}\overline{F}_{bot}\right)\right] . \tag{11}
$$

Here, we split the equation into two terms, enclosed by the square brackets. The first term represents the local deviation from the average surface melt rate, which changes the general shape of the to-pography while preserving its average height above sea level. We identify this term with enhanced melting. The second term represents a global shift of the average elevation above sea level due to freeboard sinking.

In this way, the topographic evolution equation can be split into two terms: enhanced melting, and freeboard sinking:

$$
\frac{ds}{dt} = \frac{ds_{em}}{dt} + \frac{ds_{fs}}{dt} , \tag{12}
$$

where $\frac{ds_{em}}{dt}$, and $\frac{ds_{fs}}{dt}$ are contributions from enhanced melting, and freeboard sinking, and correspond to the first and second term of Eq. (11).

### 2.3 Model for the evolution of pond coverage

We now need to relate the vertical displacements near the sea level to the change in area of the melt ponds. To this end we define the hypsographic curve, $s(x_h)$, which relates the elevation above sea level, $s$, to the percent of ice surface below that elevation, $x_h$ (Fig. 2). Such curves have been measured and reported on several occasions (e.g. Fig. 8 of Eicken et. al. (2004), or Fig. 8 of Landy et. al. (2014)). If the ice is highly permeable, the melt pond fraction, $x$, can be inferred from a hypsographic curve as the intersection of sea level with the curve. Since ponds are hydraulically connected with the ocean, the average freeboard height of bare ice, $h$, depends on the pond fraction. The average freeboard height, $h$, can be expressed in terms of the ice thickness $H$ and the pond fraction as

$$
h = \frac{\rho_w - \rho_i}{\rho_w}\frac{H}{1-x} . \tag{13}
$$

Here, the average freeboard height is defined as the elevation of the ice surface above sea level averaged over bare ice. For two ice floes of the same thickness, the one with higher pond coverage will also need to have a higher average freeboard in order to maintain hydrostatic balance.

The above sea level part of every measured hypsographic curve we tested can be fit relatively well with a tangent function (Fig. 2a, red line). We will assume that this fit holds for a wide range

of different sea ice floes, and use it to initialize our model with different physical parameters. We give the exact form of this function in appendix A (Eq. 35). To get a hypsographic curve for a particular initial pond fraction, $x_i$, and ice thickness, $H$, we set it to zero at the initial pond coverage, $s(x_h = x_i) = 0$, and rescale it vertically to get a freeboard that hydrostatically balances the floe. The topography below sea level is not important for the evolution of pond coverage if the pond coverage grows, and we replace it with a straight line.

We show several curves for different initial ice thickness and initial pond coverage in Fig. 2b and Fig. 2c. We note that the initial pond fraction, $x_i$, corresponds to the pond fraction when ice first becomes permeable. Once we choose $x_i$ and $H$, the tangent function Eq. (35) has only two unconstrained parameters, $p_1$ and $p_2$, that determine the exact shape of the curve. Knowing additional physical parameters, such as ice roughness, we can constrain additional parameters of this curve. Throughout this paper we will mostly use $p_1$ and $p_2$ that fit the measurements of the hypsographic curve made by Landy et. al. (2014) for June 25th of 2011 or the measurements made during the SHEBA mission along the topography profile "1" on July 10th 1998. However, when examining the effects of sea ice roughness, we will vary these parameters to get curves of different shape. Several examples of hypsographic with different $p_1$ and $p_2$ are shown in Fig. 2d.

In the case of pure freeboard sinking the overall shape of the hypsographic curve does not change as the ice melts. Instead the whole curve is shifted following a displacement of $ds_{\text{fs}}$ (Fig. 3a). We can calculate the resulting change in pond coverage as

$$\frac{dx}{dt} = \frac{dx_h}{ds}(x)\frac{ds_{\text{fs}}}{dt} \, , \tag{14}$$

where $ds_{\text{fs}}$ is the vertical displacement of the bare ice topography due to freeboard sinking (as determined by the second term in Eq. (11)), and $\frac{dx_h}{ds}(x)$ is the change in pond fraction for a vertical shift of the ice surface of $ds_{\text{fs}}$ when the pond fraction is equal to $x$. It is equal to the reciprocal of the derivative of the hypsographic curve, $s(x_h)$, evaluated at $x_h = x$. Substituting $\frac{ds_{\text{fs}}}{dt}$ from Eq. (11) we find

$$\frac{dx}{dt} = \frac{d\hat{x}_h}{d\hat{s}}(x)\Big[S_{\text{bi}} + S_{\text{mp}}\frac{\widehat{x}}{1-x} + S_{\text{bot}}\frac{1}{\widehat{1-x}}\Big] \, , \tag{15}$$

where $\widehat{x} \equiv \frac{x}{x_i}$, and $\widehat{1-x} \equiv \frac{1-x}{1-x_i}$ are the pond and bare ice fractions normalized by the initial pond and bare ice fractions, $\frac{d\hat{x}_h}{d\hat{s}}(x) \equiv \frac{h}{1-x_i}\frac{dx_h}{ds}(x)$ is the non-dimensional slope of the hypsographic curve, and we have defined the strengths of pond growth by freeboard sinking due to melting bare, ponded, and ice bottom, $S_{\text{bi}}$, $S_{\text{mp}}$, and $S_{\text{bot}}$ as

$$S_{\text{bi}} \equiv \frac{(1-x_i)^2\overline{F}_{\text{bi}}}{Hl\rho_b} \, ,$$
$$S_{\text{mp}} \equiv \frac{(1-x_i)x_i\overline{F}_{\text{mp}}}{Hl\rho_b} \, ,$$
$$S_{\text{bot}} \equiv \frac{(1-x_i)\overline{F}_{\text{bot}}}{Hl\rho_b} \, . \tag{16}$$

The non-dimensional factors $\widehat{x}$, $\widehat{1-x}$, and $\frac{d\hat{x}_h}{d\hat{s}}(x)$ are chosen to be of the order unity, so that $S_{\mathrm{bi}}$, $S_{\mathrm{mp}}$, and $S_{\mathrm{bot}}$ control the strengths of pond growth by melting bare ice, melting ponded ice, and melting ice bottom. The reciprocals of the strengths represent the timescales of the growth modes.

The set of parameters needed to describe pure freeboard sinking can be further reduced by rewriting Eq. (15) as

$$\frac{dx}{dt} = \frac{d\hat{x}_h}{d\hat{s}}(x)\Big[S_1 \frac{\widehat{x}}{\widehat{1-x}} + S_2 \frac{1}{\widehat{1-x}}\Big] , \tag{17}$$

where $S_1 \equiv S_{\mathrm{mp}} - x_i S_{\mathrm{bi}}/(1-x_i)$ and $S_1 \equiv S_{\mathrm{bot}} + S_{\mathrm{bi}}/(1-x_i)$ represent a minimal set of parameters needed to describe pure freeboard sinking. However, these parameters do not have a clear physical interpretation, and we will henceforth focus only on $S_{\mathrm{bi}}$, $S_{\mathrm{mp}}$, and $S_{\mathrm{bot}}$.

Next we need to consider the contribution from enhanced melting. Before doing so we need to make some assumptions about the nature of enhanced melt. There are multiple physical processes that can cause the melt rate to deviate from the mean. One process that stands out as being particularly important is albedo decrease due to ice wetting: ice close to sea level will likely be wet and therefore have a lower albedo compared to ice higher up. The deviation from the mean melt rate in this case depends primarily on the height above sea level. Another potential contribution to height-dependent enhanced melt may effectively come from random fluctuations in the melt rate around the average: ice near the sea level has a higher probability of falling below sea level due to random fluctuations than ice higher up. After falling below sea level, ice becomes ponded, melts faster, and is unable to return to its previous position. Other processes, such as lateral melt, may not depend on height above sea level, but for now we neglect this possibility (see section 7.1 for discussion).

Because of the processes described above, we will assume that the deviation from the mean melt rate, $k(\boldsymbol{r}) - 1$, depends only on height above sea level, $s$. In this scenario, we need to consider enhanced melting together with freeboard sinking, as freeboard sinking constantly supplies new ice to low elevations to be affected by enhanced melting. Effects of enhanced melting and freeboard sinking can be approximately separated if, instead of height-dependence, enhanced melting is constrained to act on a fixed fraction of bare ice. In this case, a constant fraction of bare ice that would experience enhanced melting would evolve, at least approximately, independently of freeboard sinking.

Therefore, we will consider two cases of enhanced melting. Firstly, we will consider a height-dependent enhanced melting. In particular, we will assume that $k(0 < s < \Delta s) \equiv k$ and $k(s > \Delta s) \equiv 1$, where $\Delta s$ is a height above which there is no enhanced melting and below which enhanced melting is constant $k > 1$. This is the case we ultimately wish to describe. We describe a potential model for pond growth under this assumption in appendix B and Fig. 3d. However, from a practical viewpoint, it is simpler to consider enhanced melting which acts upon a fixed fraction of bare ice. In this case, we will assume that $k(x < x_h < x + \delta) \equiv k$ and $k(x_h > x + \delta) \equiv 1$, where $\delta$ is a fraction of ice affected by enhanced melting (Fig. 3b). In appendix B, we show that, if $\delta$ is appropriately chosen, a height-dependent model and a fixed fraction model become equivalent. Therefore, we will first

solve a model assuming a fixed $\delta$ and no freeboard sinking, and then relate it to a fixed $\Delta s$ model by choosing the appropriate $\delta$.

We note that the assumption that $k(\boldsymbol{r}) = 1$ high above the sea level and $k(\boldsymbol{r}) > 1$ near the sea level is strictly not true since averaged over all of bare ice $k(\boldsymbol{r})$ needs to equal one. However, it is approximately true if $\Delta s$ or $\delta$ are small, such that the area where $k(\boldsymbol{r}) \neq 1$ is small compared to

295 the total area of bare ice. Also, we have assumed $k(\boldsymbol{r}) = 1$ high above the sea level without loss of generality, since deviations from the mean melt rate high above the sea level are not important, as only ice close to sea level may become ponded.

Now we proceed to consider the case of "pure enhanced melting" that assumes a fixed fraction of the ice, $\delta$, melts, and there is no freeboard sinking (Fig. 3b). If there is no topographic variation above

300 sea level, and the entire ice floe above sea level has the same height, $h$, the pond coverage would grow by $\delta$ after a time $\Delta t = \frac{h}{ds_{\text{em}}/dt}$, where $ds_{\text{em}}/dt$ is the rate of change of topography due to enhanced melting as determined by the first term of Eq. (11). Therefore, the pond growth rate in this case would be $\frac{\Delta x}{\Delta t} = \frac{\delta}{h} \frac{ds_{\text{em}}}{dt}$. If there is non-negligible topography above sea level described by the hypsographic curve, the time $\Delta t$ it takes for pond coverage to grow by $\delta$, would be $\Delta t = \frac{s(x_h = x + \delta)}{ds_{\text{em}}/dt}$.

Here, $s(x_h = x + \delta)$ is the original hypsographic curve evaluated at $x_h = x + \delta$. We will assume this expression generally holds for enhanced melting. Thus, we arrive at the expression for pond growth due to pure enhanced melting with fixed $\delta$

$$\frac{dx}{dt} = \frac{\delta}{s(x+\delta)} \frac{ds_{\text{em}}}{dt} \ . \tag{18}$$

If $\delta$ is small compared to the variation in the hypsographic curve, we can substitute $s(x + \delta)$ with

310 $s(x)$. This is only not justified near the beginning of the melt, when $s(x) \approx 0$. Substituting $\frac{ds_{\text{em}}}{dt}$ from Eq. (11) we find

$$\frac{dx}{dt} = S_{\text{em}} \frac{1}{\hat{s}(x+\delta)} \ , \tag{19}$$

where $\hat{s}(x) \equiv \frac{s(x)}{h}$ is the non-dimensional hypsographic curve, and the strength of the enhanced melting, $S_{\text{em}}$, is defined as

$$S_{\text{em}} \equiv \frac{\rho_{\text{w}}}{\rho_{\text{w}} - \rho_{\text{i}}} \frac{(1 - x_i)\delta(k-1)\overline{F}_{\text{bi}}}{Hl\rho_b} \ . \tag{20}$$

Ultimately, however, our goal was to describe the height-dependent enhanced melting. In appendix B, we showed that such a model can be approximated with a fixed fraction model, if we appropriately relate $\delta$ and $\Delta s$. Here we simply state the result

$$\delta = \frac{\rho_{\text{w}}}{\rho_{\text{w}} - \rho_{\text{i}}} \frac{2\Delta s(1 - x_i)^2}{3H(1 + \frac{ds_{\text{em}}}{ds_{\text{fs}}})} \ . \tag{21}$$

Here, $\frac{ds_{\text{em}}}{ds_{\text{fs}}}$ represents the ratio of the topographic rate of change due to enhanced melting to freeboard sinking and is given by

$$\frac{ds_{\text{em}}}{ds_{\text{fs}}} = \frac{\rho_w}{\rho_w - \rho_i} \frac{|\overline{F}_{\text{bi}}|(k-1)}{|\overline{F}_{\text{bi}}| + \frac{x_i}{1-x_i}|\overline{F}_{\text{mp}}| + \frac{1}{1-x_i}|\overline{F}_{\text{bot}}|} \ , \tag{22}$$

where $|\overline{F}|$ are the representative values of energy fluxes, e.g. their time-averages. Therefore, the strength of height-dependent enhanced melting becomes

$$S_{\text{em}} = \left(\frac{\rho_{\text{w}}}{\rho_{\text{w}} - \rho_{\text{i}}}\right)^2 \frac{2\Delta s(1-x_i)^3(k-1)\overline{F}_{\text{bi}}}{3H^2 l \rho_b (1 + \frac{ds_{\text{em}}}{ds_{\text{fs}}})} \ . \tag{23}$$

We have made a number of assumptions in deriving the expression for enhanced melting. Below we compare this model to a more complicated "1D" model and show that all these assumptions are justified. We also show that if the function describing the local melt rate, $k(s)$, has a non-trivial dependence on height above sea level, parameter $S_{\text{em}}$ is better replaced with a parameter

$$< S_{\text{em}} > \equiv \left(\frac{\rho_{\text{w}}}{\rho_{\text{w}} - \rho_{\text{i}}}\right)^2 \frac{2(1-x_i)^3\overline{F}_{\text{bi}}}{3H^2 l \rho_b} \int\limits_0^\infty \frac{k(s)-1}{1 + \frac{ds_{\text{em}}}{ds_{\text{fs}}}(s)} ds \tag{24}$$

In this way, we have separated the effects of freeboard sinking and enhanced melting. Finally, we will assume that contributions from freeboard sinking and enhanced melting can be added independently. Therefore, we solve Eq. (15) for pure freeboard sinking, and Eq. (19) for enhanced melting independently, and add them together to get the full evolution of pond coverage, $x(t)$:

$$x(t) = x_{\text{fs}}(t) + x_{\text{em}}(t) - x_i \ , \tag{25}$$

where $x_{\text{fs}}(t)$, and $x_{\text{em}}(t)$ are solutions to Eq. (15), and Eq. (19), both forced using the same parameters, and initialized with the same initial pond fraction $x_i$. This concludes the 0D model.

Equation (25) represents a sum of solutions to two simple ordinary differential equations, in which the rate of change of pond fraction depends on the pond fraction. Here, we have reduced the number of parameters from the original ten ($H$, $x_i$, $\rho_b$, $\overline{F}_{\text{bot}}$, $F_{\text{sol}}$, $F_{\text{r}}$, $\alpha_{\text{bi}}$, $\alpha_{\text{mp}}$, $k$, and $\Delta s$) to four ($S_{\text{bi}}$, $S_{\text{mp}}$, $S_{\text{bot}}$, and $S_{\text{em}}$). The strengths of freeboard sinking, $S_{\text{bi}}$, $S_{\text{mp}}$, and $S_{\text{bot}}$, depend only on the parameters that are available in GCM simulations, and are relatively easily measured in observational studies. The enhanced melting strength, $S_{\text{em}}$, however, also depends on the difficult-to-measure parameters $k$ and $\Delta s$ that describe the melt rate near the sea level, and may also have contributions from processes that are not height dependent. Furthermore, as we discuss below, ice roughness can also play an important role in pond evolution. With reliable constraints on these parameters, our model would be a useful parameterization in GCMs for pond growth after ice becomes permeable.

## 2.4 Testing the model

In order to test the assumptions we made to simplify the model, we have developed a "1D" model in which we explicitly determine pond evolution when both freeboard sinking and enhanced melting are happening simultaneously. Apart from resolving the melt rates in one dimension, the underlying assumptions for the 1D model are essentially the same as for the simple model. For this reason, we simply give an outline for this model, without discussing it in much detail.

In the 1D model, we evolve the hypsographic curve by prescribing a melt rate, $ds_{\text{loc}}$, to each point on the hypsographic curve depending on the height above sea level (Fig. 3c). The hypsographic curve

high above sea level melts at a uniform rate, whereas the hypsographic curve slightly above sea level melts at an enhanced rate. Parts of the curve below sea level melt at a uniform rate determined by the flux used for melting ponded ice, $\overline{F}_{\mathrm{mp}}$. Finally, hydrostatic adjustment is calculated by finding the ice thickness directly at each time step, and placing the floe in hydrostatic balance. The evolution of pond coverage obtained from this model is shown in Fig. 4a. The comparison with the simple 0D model is excellent.

The 1D model allows us some freedom to test the detailed assumptions of the 0D model. First, we can test how the functional form of $k(s)$ affects the pond evolution (Fig. 4b). The functions $k(s)$ were chosen such that they all have the same integral parameter $< S_{\mathrm{em}} >$ defined in Eq. (24). Figure 4a shows that in each of these cases the evolution of pond coverage proceeds nearly identically. Second, we can test the difference between an assumption that enhanced melting acts below a constant height $\Delta s$ and an assumption that enhanced melting acts on a constant fraction of ice, $\delta$. The yellow line in Fig. 4a shows that if $\delta$ and $\Delta s$ are chosen according to Eq. (21), both assumptions yield very similar results. Finally, we can test the effects of varying pond albedo. In reality pond albedo decreases as the ponds deepen. We assume a dependence of pond albedo on pond depth reported in Table VII of Morassutti and Ledrew (1996) for mean broadband albedo. The magenta line in Fig. 4a shows that allowing for pond albedo to vary has a negligible effect on pond evolution.

We should note that, when both freeboard sinking and enhanced melting occur simultaneously, the agreement between the 0D model and the 1D model becomes poor if the hypsographic curve is convex (e.g. Fig. 2d, blue curve), and the 0D model should be used with care. Happily, the measured hypsographic curves are mostly concave, in which case the agreement between the two models is excellent.

## 3  A 0D model can approximate observations well using realistic parameters

In Fig. 6, we compare the results from our model to observations made on a 200m long albedo line during SHEBA (red line). Ice along the albedo line was level multiyear ice, but the ponds drained to sea level after some time which makes them amenable to our model (Perovich et. al., 2003). The pond coverage along the albedo line dropped to a minimum around the end of June. Therefore, we choose to model only the period after July 1st. In order to keep the albedo line pristine, no thickness measurements were made. However, relatively close to the albedo line, topography measurements were made along a level multiyear ice profile roughly every ten days. After approximately July 10th, ponds along the topography profile also drained to sea level. We show the topography profile pond coverage in blue dots (we have artificially subtracted 0.05 from the pond coverage to facilitate comparison with the pond coverage along the albedo line). The pond coverage along the topography profile and along the albedo line follow roughly the same trend, suggesting that the physical parameters driving the pond evolution in the two places are likely similar. Based on the average freeboard

height, we estimate the ice thickness on July 10th to be roughly $1.4$m along the topography profile, meaning that on July 1st, ice thickness was around $1.6$m. Therefore, we assume the same thickness for the ice along the albedo line, and use a hypsographic curve corresponding to the one measured along the topography profile on July 10th (Fig. 2a, dashed line). In order to run our model, we use the melt rates of bare ice, ponded ice, and ice bottom measured directly using ablation stakes during SHEBA (Perovich et. al., 2003). We choose a realistic $\rho_b = 850 \, \text{kg m}^{-3}$ (Timco and Frederking, 1996). We have no way of directly constraining the parameters $\Delta s$ and $k$ that control the strength of enhanced melting. Therefore, we treat $S_{\text{em}}$ as a fitting parameter. Choosing $S_{\text{em}} = 0.22 \, \text{month}^{-1}$ fits the observations well by eye. This value can be obtained using $\Delta s = 15$ cm and $k = 1.7$ which likely fall at the upper end of the range of reasonable values for these constants (see section 4 for a discussion on $\Delta s$ and $k$). Such a high value of $S_{\text{em}}$ can be explained by a significant contribution from lateral melting.

The full black line in Fig. 6 represents a solution to the full Eq. (25). The agreement between model and observation is excellent, with a maximum discrepancy of $3\%$ pond coverage at the end of the melt season. The dashed black line represents the contribution to pond growth due to freeboard sinking, whereas the dotted line corresponds to enhanced melting. Almost all pond growth in this case is due to enhanced melting. This is due to ice topography. On multiyear ice, meltwater typically collects in depressions formed by ponds in previous years. The topography created in this way is highly bimodal, and, after drainage, ponds typically have steep walls. Bare ice topography, on the other hand, is relatively smooth, preventing new pond formation. This is apparent in the hypsographic curve we used. Such a topography inhibits freeboard sinking, and pond coverage grows mostly by enhanced melting acting near the pond sidewalls, growing the existing ponds. In addition to height-dependent enhanced melting we introduced in the previous section, in this case there is likely a significant contribution from lateral melting as well. This contribution helps explain the high value of $S_{\text{em}}$ we had to choose to get a close agreement between our model and observations. First year ice topography, on the other hand, permits ample pond growth through freeboard sinking. Observations suggest that on first year ice ponds grow primarily due to freeboard sinking (Polashenski et. al., 2012; Landy et. al., 2014).

## 4 Numerical solutions

We now solve Eq. (25) numerically to gain intuition about the behavior of our model. We use a set of realistic parameters we will henceforth refer to as the "default parameters."

For shortwave, longwave, latent, and sensible heat fluxes, we use values inferred by Skyllingstad et. al. (2009) using hourly measurements from the SHEBA mission. We use the bottom heat flux inferred from measurements of ice bottom ablation during the SHEBA mission (Perovich et. al., 2003). The albedo of bare ice can vary between 0.5 and 0.7 (Hanesiak et. al., 2001), while the albedo of

melt ponds can vary between 0.1 and 0.6, depending on pond depth and conditions of ice at the pond bottom (Morassutti and Ledrew, 1996; Perovich et. al., 1998; Perovich, 1996). Here we prescribe a default bare ice albedo of 0.55, and a default pond albedo of 0.2. We use a realistic bulk ice density of $\rho_b = 850$ kg m$^{-3}$ (Timco and Frederking, 1996). We use an initial ice thickness of 1.5 m, and use the first year ice topography measured by Landy et. al. (2014) adjusted for the prescribed ice thickness and initial pond fraction (usually $x_i = 0.2$). We will assume enhanced melting is entirely due the albedo dependence on height above sea level. Some preliminary results based on field measurements of bare ice albedo on first year ice suggest that albedo changes from around 0.3 near sea level to around 0.55 at a height of around 10 cm above sea level, after which the correlation between albedo and surface elevation tapers off (Chris Polashenski, pers. comm.). Using such an albedo and the average values of shortwave, longwave, latent, and sensible heat fluxes, we can estimate the rate of melt as a function of height above sea level, $k(s) = \frac{\overline{F}(s)}{\overline{F}_{\mathrm{bi}}}$. Using Eq. (24), we can then find the integral parameter $< S_{\mathrm{em}} >$. We choose $\Delta s = 6$ cm and $k = 1.7$ to correspond to the same integral parameter. We should note that there is significant scatter in the data, and measurements correspond to only one study. Therefore, this is a rough estimate of enhanced melting, but it is likely of the correct order of magnitude.

Figure 5a shows the solution to Eq. (25) for different initial conditions. We can see that ponds grow more rapidly when the initial pond coverage is lower, and the pond evolution curves cluster together as time progresses. This is because lower initial pond coverage corresponds to lower initial freeboard height, making the pond growth more rapid. The dashed line corresponds to the solution using the fluxes time-averaged over the 30 day run. The solutions using the averaged fluxes are very similar to the ones using time-varying fluxes, meaning that daily, and even monthly variations in the forcing have little effect on pond growth. This insensitivity to short time scale variations in the forcing means that pond coverage evolution may be faithfully represented in the large scale models, as it would not be affected by the coarse time scales of those models. Henceforth, we will use the time-averaged fluxes.

A larger ice thickness means a higher freeboard. For this reason, ponds grow more slowly on thicker ice. Because the pond growth rate is inversely proportional to ice thickness, pond coverage is more sensitive to variations in ice thickness when the ice is thin (Fig. 5b). In Fig. 5b we see that a 0.5 m difference in the initial ice thickness (between a floe 1.5m and a floe 2m thick) can mean a 20% difference in pond coverage at the end of the melt season.

Figure 5c shows the dependence of pond coverage on albedo. A variation of 0.1 in bare ice albedo has a much larger effect on pond evolution than the same change in pond albedo. The reason is that melting ponded ice only affects pond coverage through downward rigid body motion of the floe, whereas melting bare ice grows the ponds through both enhanced melting and freeboard sinking. Furthermore, when pond coverage is low, rigid body motion due to ponded ice melting is less efficient than that due to bare ice melting because it is proportional to melt pond fraction.

The parameters controlling the strength of enhanced melting are the least constrained parameters in our model. In Fig. 5d we show the dependence of pond evolution on the height below which enhanced melting is active, $\Delta s$. Exploring a range of realistic values for $\Delta s$, $0 < \Delta s < 15$ cm, we find that the pond fraction at the end of the melt season can vary by about $30\%$. This difference would be larger if we chose a smaller ice thickness. The effects of changing $k$ are relatively small, so long as $k$ is large enough (not shown). For example, using current parameters, pond coverage evolution becomes fairly insensitive to $k$ when $k > 1.5$. Smaller values of $k$, however, can significantly impact pond evolution. If $k$ is enough smaller than 1, $S_{\text{em}}$ can become negative, and the pond coverage can stop growing. In this case, ice near the sea level melts slowly enough such that an upward rigid body movement due to melting ice high above sea level pushes the ice near sea level upwards, preventing pond coverage growth. The evolution of such a pond coverage cannot be represented well in our model since the equation for enhanced melting becomes invalid in this case, and the blue curve in Fig. 5d serves therefore simply as an illustration.

## 5 Pond evolution is slower on smoother ice

The evolution of pond coverage in our model depends on the detailed shape of the hypsographic curve which is not captured by the strengths of freeboard sinking and enhanced melting. As we show below, pond coverage is sensitive to such details, and in particular to ice roughness. Below we will introduce the "effective strengths", $S^*$, which approximately capture the effects of roughness and allow us to estimate mean pond coverage after a period of time. Using effective strengths, we will demonstrate how multiyear ice topography suppresses pond growth by freeboard sinking, while first year ice topography permits it.

In the tangent function parameterization, Eq. (35), the exact shape of the hypsographic curve is determined by parameters $p_1$ and $p_2$. Here, we will not discuss these parameters individually, but will rather focus on often measured bare ice roughness, $\sigma$, defined as the standard deviation of surface elevation of ice above sea level:

$$\sigma \equiv \left( \frac{\int_{x_i}^{1} s^2(x_h) dx_h}{1 - x_i} - h^2 \right)^{\frac{1}{2}}. \tag{26}$$

We will use the non-dimensional form of bare ice roughness, defined as $\hat{\sigma} \equiv \frac{\sigma}{h}$. Typically, a concave hypsographic curve (e.g. Fig. 2d, red curve) will have a small $\hat{\sigma}$, whereas a convex hypsographic curve (e.g. Fig. 2d, blue curve) will have a high $\hat{\sigma}$.

During the permeable stage, all else equal, ponds will grow more rapidly on rougher ice, since a larger fraction of ice is close to sea level. This is not true on impermeable ice, as meltwater filling deep topographic lows on rough ice will cover a smaller area relative to the same amount of meltwater filling shallow topographic lows on smooth ice. For this reason, the initial pond coverage will likely be smaller on rougher ice due to a smaller pond coverage during the impermeable stage.

Figures 7 show the pond coverage evolution due to pure freeboard sinking (Fig. 7a) and pure enhanced melting (Fig. 7c) for hypsographic curves with different parameters $p_1$ and $p_2$ and all other parameters kept constant. For each choice of $p_1$ and $p_2$, we find the normalized bare ice roughness, $\hat{\sigma}$, represented by the color of the curves. Blue colors correspond to low roughness and red colors to high roughness. Pond evolution on measured topographies (Fig. 2a) is also shown. We can see that although roughness does not fully determine the pond evolution, it is a viable proxy for how pond coverage will evolve, with high roughness curves typically having a higher average pond coverage.

We wish to quantify the effect of roughness by its impact on the mean pond coverage. In particular, we hope to find the "effective strengths", $S^*(\hat{\sigma})$, which include the roughness effects and allow us to easily estimate the average pond coverage after some time $t$

$$< x(t) > \approx \frac{1}{2} S^* t + x_i \,, \tag{27}$$

where $< x(t) > \equiv \frac{\int_0^t x(t) dt}{t}$. Effective strengths are proportional to strengths of freeboard sinking and enhanced melting we derived in section 2.3. In general they themselves may depend on time, and are time-independent only if pond coverage evolution is linear, $x(t) = St + x_i$, in which case $S^* = S$, where $S$ is either $S_{fs} \equiv (S_{bi} + S_{mp} + S_{bot})$ in the case of freeboard sinking or $S_{em}$ in case of enhanced melting.

In appendix C, we describe the procedure to estimate the effective strengths as functions of non-dimensional roughness and time. Here, we only state the result

$$S_{fs}^* \approx \left[ 1.3\hat{\sigma}^2 \right] \left( S_{bi} + S_{mp} + S_{bot} \right) \,,$$
$$S_{em}^* \approx \left[ 1 + \left( \frac{2}{\sqrt{\hat{t}_{em}}} - \frac{3}{2} \right) \hat{\sigma} \right] S_{em} \,, \tag{28}$$

where $S_{fs}^*$ is the effective strength of freeboard sinking, $S_{em}^*$ is the effective strength of enhanced melting, and $\hat{t}_{em} \equiv \frac{S_{em}t}{1 - x_i}$ is the non-dimensional time of pond evolution due to enhanced melting. The terms in square brackets represent the corrections due to roughness. If both freeboard sinking and enhanced melting occur simultaneously the total effective strength is the sum of these two, $S^* = S_{fs}^* + S_{em}^*$. Knowing the effective strengths, allows us estimate the mean pond coverage after a period of time without having to run the model.

Roughness has a different effect on freeboard sinking and enhanced melting. Freeboard sinking is roughly independent of time and proportional to the square of non-dimensional roughness. Therefore, it is very sensitive to variations in roughness: doubling the ice roughness roughly quadruples the mean pond coverage due to freeboard sinking after some time. Enhanced melting depends roughly linearly on roughness. However, as roughness tends to zero, the effective strength remains non-zero, $S_{em}^*(\hat{\sigma} \to 0) \to S_{em}$. Therefore, ponds on smooth ice grow primarily due to enhanced melting. Effective strength also depends on the non-dimensional time, $\hat{t}$, and is higher and more sensitive to variations in roughness early in the melt season.

Multiyear ice topography shown in Fig. 2a, dashed line, has $\hat{\sigma} \approx 0.25$ and is significantly smoother than first year ice topography shown in Fig. 2a, solid line, which has $\hat{\sigma} \approx 0.55$. From Eq. (28) it follows that freeboard sinking on multiyear ice is roughly 5 times less efficient in growing the ponds than on first year ice.

## 6 Analyzing the 0D model yields useful insight into factors influencing the pond evolution

Extracting the dependence of a desired property on physical parameters and understanding its scaling is the main strength of our model. These types of relationships would be difficult to obtain in a more complex model.

The parameters $S_{\mathrm{bi}}^*$, $S_{\mathrm{mp}}^*$, $S_{\mathrm{bot}}^*$, and $S_{\mathrm{em}}^*$ control the mean rates of pond growth by melting different regions of ice. Roughly, they represent the amount of pond growth per unit time by freeboard sinking

due to melting bare ice; freeboard sinking due to melting ponded ice; freeboard sinking due to melting ice bottom; and enhanced melting. Knowing these parameters allows us to estimate mean pond coverage after a period of time with significant accuracy without having to run the numerical model. Moreover, analyzing them can yield useful insight into the behavior of melt ponds under general circumstances.

We can estimate the change in magnitude of the strength of each of the growth modes when a physical parameters $p$ changes by $\Delta p$ as

$$\Delta S_i^* = \frac{\partial S_i^*}{\partial p} \Delta p \,, \tag{29}$$

where $\Delta S_i^*$ is the change in magnitude of the effective strength of the $i$th growth mode. This equation holds so long as the change in the physical parameter is not too large. A change in pond growth rate

can then be estimated as $\Delta S^* = \sum_i \Delta S_i^*$. Then, using Eq. (27), we can roughly estimate a change in mean pond fraction, $\Delta < x >$, after some time, $\Delta t$, following a change in physical parameter, $p$, as $\Delta < x > \approx \frac{1}{2} \Delta S^* \Delta t$. This provides a means to estimate changes in mean pond coverage under different environmental conditions.

### 6.1 Ponds are more sensitive to changes in bare ice albedo than changes in pond albedo

We will illustrate the use of effective strengths using an example where we vary the ice and pond albedos. If the bare ice albedo changes by $\Delta \alpha_{\mathrm{bi}}$, the change in growth rate would be roughly

$$\Delta S^* = -\Big[S_{\mathrm{bi}}^* + \frac{\frac{\rho_w - \rho_i}{\rho_w}(ds_{\mathrm{em}}/ds_{\mathrm{fs}})^2 + (k-1)}{(1+\frac{ds_{\mathrm{em}}}{ds_{\mathrm{fs}}})(k-1)} S_{\mathrm{em}}^*\Big] \frac{F_{\mathrm{sol}}}{\overline{F}_{\mathrm{bi}}} \Delta \alpha_{\mathrm{bi}} \approx -0.9 \frac{1}{\mathrm{month}} \Delta \alpha_{\mathrm{bi}} \,. \tag{30}$$

On the other hand, if the melt pond albedo changes by $\Delta \alpha_{\mathrm{mp}}$, the change in growth rate would be roughly

$$\Delta S^* = -\Big[S_{\mathrm{mp}}^* + \frac{(\rho_w - \rho_i)x_i(ds_{\mathrm{em}}/ds_{\mathrm{fs}})^2 \overline{F}_{\mathrm{mp}}}{\rho_w(1+\frac{ds_{\mathrm{em}}}{ds_{\mathrm{fs}}})(k-1)(1-x_i)\overline{F}_{\mathrm{bi}}} S_{\mathrm{em}}^*\Big] \frac{F_{\mathrm{sol}}}{\overline{F}_{\mathrm{mp}}} \Delta \alpha_{\mathrm{mp}} \approx -0.2 \frac{1}{\mathrm{month}} \Delta \alpha_{\mathrm{mp}} \,. \tag{31}$$

It follows from these estimates that after a month the mean pond fraction would differ by roughly 4.5% for a bare ice albedo difference of 0.1, and by around 1% for a pond albedo difference of 0.1. Therefore, variation in pond albedo affects pond evolution roughly five times less than variation in bare ice albedo. This explains our observation from Fig. 5c that pond evolution is much more sensitive to variations in bare ice albedo than to variations in pond albedo. In this way, we also extract the dependence of sensitivity on physical parameters. A major difference between the two sensitivities is their dependence on the initial pond coverage: the sensitivity to pond albedo is proportional to $x_i$, whereas the sensitivity to bare ice albedo is proportional to $1 - x_i$. In the above example we used $x_i = 0.2$, which explains most of the large difference between the two sensitivities. If the pond coverage were higher, variations in the pond albedo could become more important than variations in bare ice albedo. For example, assuming no enhanced melting, the sensitivity to pond albedo would become greater than the sensitivity to bare ice albedo at 50% pond coverage ($\frac{\Delta S_{mp}^*}{\Delta S_{bi}^*} = \frac{x_i}{1-x_i}\frac{\Delta \alpha_{mp}}{\Delta \alpha_{bi}}$).

## 6.2 Under global warming, pond feedback could lead to significant ice thinning

We now use the effective strengths to roughly estimate the impact of global warming on the pond coverage. At high latitudes, feedbacks due to changes in albedo, the atmospheric lapse rate, and clouds can amplify the forcing due to global warming (Holland and Bitz, 2006). For this reason forcing at high latitudes is generally larger than direct radiative forcing due to an increase in $CO_2$ concentration. In a global warming scenario, the pond growth rate would increase because the ice melts faster, but also because ice at the beginning of the melt would be thinner. We can emulate a global warming scenario by increasing the flux $F_r$ by a certain amount, $\Delta F_r$, and by assuming that the initial ice thickness decreases by $\Delta H \equiv \frac{\partial H}{\partial F_r}\Delta F_r$, where $\frac{\partial H}{\partial F_r}$ is the ice thinning per $1 \text{ Wm}^{-2}$ of warming. Therefore, we split the change in pond growth rate due to global warming, $\Delta S^*$, into a contribution from direct forcing, $\Delta S_F^*$, and a contribution from ice thinning, $\Delta S_H^*$. Using the above formalism, we find

$$\Delta S_F^* \equiv \sum_i \frac{\partial S_i^*}{\partial F_r}\Delta F_r = \left[\frac{S_{bi}^*}{\overline{F}_{bi}} + \frac{S_{mp}^*}{\overline{F}_{mp}} + \frac{\frac{\rho_w - \rho_i}{\rho_w}(ds_{em}/ds_{fs})^2 + (k-1)(1-x_i)}{(1 + \frac{ds_{em}}{ds_{fs}})(k-1)(1-x_i)}\frac{S_{em}^*}{\overline{F}_{bi}}\right]\Delta F_r \approx \frac{0.5\%}{\text{W/m}^2 \times \text{month}}\Delta F_r \, ,$$

$$\Delta S_H^* \equiv \sum_i \frac{\partial S_i^*}{\partial H}\frac{\partial H}{\partial F_r}\Delta F_r = -\left(S_{bi}^* + S_{mp}^* + S_{bot}^* + 2S_{em}^*\right)\frac{1}{H}\frac{\partial H}{\partial F_r}\Delta F_r \approx \frac{1.9\%}{\text{W/m}^2 \times \text{month}}\Delta F_r \, ,$$

$$\Delta S^* \equiv \Delta S_F^* + \Delta S_H^* \approx \frac{2.4\%}{\text{W/m}^2 \times \text{month}}\Delta F_r \, .$$

$$(32)$$

The numbers in Eq. (32) were obtained using the default values of the parameters, and $\frac{\partial H}{\partial F_r} = -0.05 \text{ m}^3 \text{ W}^{-1}$ roughly estimated using the Eisenman and Wettlaufer (2008) model. This means that after a month's growth global warming would increase mean pond coverage by roughly 1.2% per $1 \text{ Wm}^{-2}$ of warming. Nearly half of this increase in the mean pond coverage comes from an increase in the strength of enhanced melting due to ice thinning. Simulating a 30 day melt numeri-

cally using our model predicts an increase in mean pond coverage with forcing at a rate of $1.5\%$ per $1\,\mathrm{Wm}^{-2}$ of warming for small forcing ($\Delta F_r \approx 0$), which confirms the approximate validity of our linearization. For larger forcing, the sensitivity of pond coverage to forcing increases because the ice thins. Our linearized estimate, Eq. (32), also gives the dependence of the sensitivity on physical parameters. In a likely scenario where the forcing is around $10\,\mathrm{Wm}^{-2}$, our estimate predicts that after a month mean pond coverage would increase by around $15\%$, which corresponds to around 12 cm of ice thinning solely due to the pond feedback. Ice thinning after a month directly due to forcing would be only around 9 cm, meaning that the pond feedback must be taken into account to understand ice thinning under global warming. Increased forcing could also lead to changes in initial pond coverage, changes in ice roughness or changes in $\Delta s$ or $k$. We ignored these feedbacks, as we have no way of reliably estimating $\frac{\partial p}{\partial F_r}$ for these parameters.

### 6.3   Different growth modes yield different pond evolution

Each of the four growth modes has different effects on the pond coverage. We will now look in detail at each of the growth modes, their effect on the pond evolution, and their scaling with physical parameters. Figure 8 shows the dependence of growth rate on pond fraction and solutions to Eq. (25) when only one of the strengths is non-zero, assuming a first year ice topography. Figure 9 shows the evolution of pond coverage distribution when only one of the strengths is non-zero.

All modes of growth depend in the same way on the bulk ice density, $\rho_b$. Each of the strengths is inversely proportional to $\rho_b$, meaning that ponds grow faster on ice with a lower bulk density. The effect is, however, modest: within a reasonable range of $916\,\mathrm{kg\,m}^{-3} > \rho_b > 750\,\mathrm{kg\,m}^{-3}$, pond growth rate can vary by at most $20\%$.

We will first discuss freeboard sinking. Common to all modes of freeboard sinking is the dependence on ice thickness. Each freeboard sinking growth mode is inversely proportional to the ice thickness, $S_{\mathrm{fs}}^* \propto \frac{1}{H}$, meaning that, all else equal, ponds grow proportionally slower on thicker ice.

Although ice roughness may have a different effect on each of the individual modes of freeboard sinking, for simplicity we will assume that they are all affected by roughness in the same way, as parameterized in Eq. (28). In that case, each of these strengths is roughly proportional to the square of the non-dimensional ice roughness, $S_{\mathrm{fs}}^* \propto \hat{\sigma}^2$, meaning that pond growth due to freeboard sinking is suppressed on smooth ice.

We will now focus on individual components of freeboard sinking. The parameter $S_{\mathrm{bi}}^*$ controls pond growth by freeboard sinking due to melting bare ice. On first year ice, owing to the shape of the hypsographic curve, the pond growth rate by bare ice melting increases up to a certain pond coverage and decreases afterwards (Fig. 8, blue line). $S_{\mathrm{bi}}^*$ is proportional to the flux $\overline{F}_{\mathrm{bi}}$, and depends on the initial pond coverage as $S_{\mathrm{bi}}^* \propto (1-x_i)^2$. The quadratic dependence on initial bare ice fraction means that ponds on floes with less initial pond coverage grow faster. It also means that floes that start off less ponded can at some point become more ponded than floes that start off more heavily

ponded. We can see this in Fig. 9a, where the pond coverage distribution narrows up to a certain point, after which it starts to widen again because floes with lower $x_i$ overtake the floes with higher $x_i$. Using the default values of physical parameters of $\overline{F}_{\text{bi}} = 85\ \text{Wm}^{-2}$, $H = 1.5$ m, $x_i = 0.2$, and $\hat{\sigma} = 0.55$, we get $S_{\text{bi}}^* \approx 0.13\ \text{month}^{-1}$.

The parameter $S_{\text{mp}}^*$ controls pond growth by freeboard sinking due to melting ponded ice. The pond growth rate increases with pond fraction from 0 at $x = 0$ to very high values at high pond coverage, and can be the dominant mode of pond growth if the pond coverage is high enough (Fig. 8, green line). For this reason, giving a representative number to pond growth rate, such as $S_{\text{mp}}$, is only meaningful if the melt season is short enough such that pond coverage during that period does not change substantially. The dependence on initial pond coverage is $S_{\text{mp}}^* \propto x_i(1 - x_i)$. For this reason the pond coverage distribution widens over time when $S_{\text{mp}}^*$ is dominant (Fig. 9b). Using $\overline{F}_{\text{mp}} = 171\ \text{Wm}^{-2}$ and other parameters the same as above, we get $S_{\text{mp}}^* \approx 0.07\ \text{month}^{-1}$. Although in this case, melting ponded ice affects pond evolution less than bare ice melting, it can become stronger if the pond coverage is higher. For example, $S_{\text{mp}}^*$ and $S_{\text{bi}}^*$ are roughly the same at $x = 0.35$, while at $x = 0.5$, $S_{\text{mp}}^*$ is roughly twice as large as $S_{\text{bi}}^*$.

The parameter $S_{\text{bot}}^*$ controls pond growth by freeboard sinking due to melting of the ice bottom. The pond growth rate due to bottom melting increases with increasing melt pond fraction, although more gradually than in the ponded ice melting case (Fig. 8, red line). Since the growth rate is proportional to the bare ice fraction, $S_{\text{bot}}^* \propto (1 - x_i)$, the pond coverage distribution gets concentrated over time (Fig. 9c). Using $\overline{F}_{\text{bot}} = 20\ \text{Wm}^{-2}$ and other parameters the same as above, we get $S_{\text{bot}}^* \approx 0.04\ \text{month}^{-1}$. The contribution from ice bottom melting becomes larger than the contribution from bare ice melting only at high $x$.

Now, we will turn to enhanced melting. The parameter $S_{\text{em}}^*$ controls pond growth by enhanced melting, and is the least constrained in our model due to the many poorly-constrained physical processes that potentially contribute to it. Here we will only consider enhanced melting due to height-dependent processes (Eq. (23)) and leave lateral melting for the discussion (subsection 7.1).

Because the growth rate by enhanced melting is inversely proportional to the hypsographic curve, pond growth by enhanced melting is very fast at the beginning of the melt, and decelerates afterwards (Fig. 8, cyan line). The enhanced melting strength is inversely proportional to the square of the ice thickness, $S_{\text{em}}^* \propto \frac{1}{H^2}$, meaning that it is significantly more sensitive to variations in thickness than freeboard sinking. On the other hand it is significantly less sensitive to variations in ice roughness, Eq. (28). Even on perfectly smooth ice, $\hat{\sigma} = 0$, ponds will grow due to enhanced melting. In that case, however, lateral melt, rather than height-dependent enhanced melting may dominate.

The strength of enhanced melting is proportional to the height below which enhanced melting is operational, $S_{\text{em}}^* \propto \Delta s$. If we take ice wetting as a physical example, this means that enhanced melting is sensitive to microphysical processes that determine how high above sea level the ice will be wet. The dependence on the parameter $k$ depends on its magnitude. It appears in $S_{\text{em}}^*$ in the

term $\frac{k-1}{ds_{em}/ds_{fs}+1}$. The term $ds_{em}/ds_{fs}$ is proportional to $k-1$. Therefore, if $ds_{em}/ds_{fs} \ll 1$, enhanced

melting is proportional to $k-1$. On the other hand, if $ds_{em}/ds_{fs} \gg 1$, enhanced melting becomes independent of $k$. Using default parameters, we find this transition happens at around $k \approx 1.2$. In the example of ice wetting, this means that enhanced melting is sensitive to albedo variations near sea level when ice near sea level has a similar albedo to the rest of the floe. On the other hand, if the albedo near sea level is significantly lower than the average, pond growth is insensitive to variations

in properties of ice near sea level.

Enhanced melting is proportional to the cube of the bare ice fraction, $S_{em}^* \propto (1-x_i)^3$, making it very sensitive to variations in initial pond coverage. For this reason, the pond coverage distribution gets quickly concentrated (Fig. 9d), and it is possible for initially less ponded floes to overtake initially more ponded floes. If we assume ice wetting is the only physical process responsible for

enhanced melting, we can place a rough estimate on $S_{sm}^*$. Taking $k = 1.7$, $\Delta s = 0.06$ m, and $t = 30$ days, we get for default parameters $S_{em}^* \approx 0.31$ month$^{-1}$. This suggests that the contribution to mean pond coverage from enhanced melting is slightly larger than the contribution from freeboard sinking after 30 days of melt.

The black line in Fig. 8 shows the total pond evolution using the default physical parameters.

The pond growth rate when both freeboard sinking and enhanced melting occur is not simply a sum of the growth rates of the four modes since the equations for freeboard sinking and enhanced melting are solved separate of each other. Therefore, the dependence of growth rate on pond coverage (Fig. 8a, black line) was obtained by finding the derivative of the pond evolution curve. The pond growth rate first decreases with pond fraction indicating that enhanced melting dominates early in the

season and then increases indicating that freeboard sinking dominates later in the season. The pond coverage distribution using realistic parameters narrows with time (Fig. 9e). Since each growth mode affects the pond coverage distribution in a distinct way, fitting both the evolution of the mean and the standard deviation of the pond coverage distribution in observational data could add constraints on the relevant strengths. Using the above values of strengths, we find that after a month of growth

bare ice melting contributes to roughly 25% of mean pond coverage, ponded ice melting contributes to around 13%, ice bottom melting contributes to around 7%, and enhanced melting contributes to roughly 55%.

## 7    Discussion

### 7.1    Lateral melting of pond walls by pond water

In our model, we focused on vertical changes in topography, and neglected pond growth by lateral melting of pond sidewalls by pond water. We will now briefly discuss this possibility.

This type of melt was the main focus of Skyllingstad et. al. (2009), who carefully calculated the lateral melt rates of pond sidewalls by pond water. The red line in Fig. 10 shows their results. The

rate of change of pond fraction due to a lateral melt flux $\overline{F}_{\mathrm{lat}}$ is

$$\frac{dx_{\mathrm{lat}}}{dt} = \frac{P}{A} \frac{\overline{F}_{\mathrm{lat}}}{l \rho_b} \,, \tag{33}$$

where $P$ is the total perimeter of the ponds and $A$ is the area of the floe. If $\overline{F}_{\mathrm{lat}}$ is constant and the dependence of $P$ on pond fraction is weak, pond growth is linear, which explains the roughly linear pond coverage evolution in Skyllingstad et. al. (2009). In Fig. 10, black line, we solve Eq. (33) assuming a lateral melt flux proportional to the ponded ice melting flux, $\overline{F}_{\mathrm{lat}} = K_{\mathrm{lat}} \overline{F}_{\mathrm{mp}}$, where $K_{\mathrm{lat}}$ is a constant. We use the same energy fluxes used by Skyllingstad et. al. (2009), and estimate $\frac{P}{A} \approx 0.1 \ \mathrm{m}^{-1}$ from the aerial photographs taken during SHEBA. A nearly perfect match is obtained with $K_{\mathrm{lat}} = 1.5$. Therefore, a single constant that relates the rate of melt of ponded ice to the rate of melt of pond walls, $K_{\mathrm{lat}}$, is enough to capture the effects of lateral melting on pond growth. This suggests that the complicated physics of lateral melting can, to a large extent, be ignored. More work would, however, be needed to determine to what degree $K_{\mathrm{lat}}$ varies under different circumstances.

If we ignore the topographic variation above sea level, pond growth due to enhanced melting also becomes linear (Eq. (19)). Therefore, lateral melting can approximately be considered a contribution to enhanced melting, $S_{\mathrm{em}}$, although it scales differently with physical parameters than the height-dependent enhanced melting, Eq. (23). It is important to note that in this model lateral melt does not depend on ice thickness, $H$, or on initial pond coverage, $x_i$, although, in reality, it may depend on these to some degree. For this reason, the pond coverage distribution width does not change in time, while the mean increases linearly (Fig. 9f).

It is not simple to understand the contribution of lateral melting to pond growth when both lateral and vertical melting occur simultaneously. Each point along the pond boundary can either expand by lateral melting or by vertical melting, but not by both. This is because when a point along the pond boundary melts laterally, it creates a completely vertical slope at that point. Therefore a small vertical shift will not grow the ponds, and a large vertical shift will outgrow the lateral expansion. Therefore, if pond growth due to vertical melting is strong, the contribution from lateral melting will be small. This is consistent with observations of Polashenski et. al. (2012) and Landy et. al. (2014) who found that on first year ice the contribution from lateral melting is small. On the other hand, steep topography on level multiyear ice inhibits pond expansion through vertical motion and could lead to lateral melting being the dominant mode of growth. This is consistent with our findings of a large contribution from enhanced melting to pond growth on multiyear ice during SHEBA (Fig. 6).

## 7.2 Effects of density variations and internal melt

So far, we have assumed that all the melt occurs either on the top or the bottom surface of the ice. However, some of the melt can happen internally, in the bulk of the ice. Internal melt occurs when trapped brine pockets with high salt content expand and dilute in order to reach a thermo-dynamic equilibrium with the surrounding ice. This phenomenon has been reported to occur both

above and below sea level. Internal melt leads to a reduction in bulk ice density, $\rho_b$, which in turn
affects pond evolution. Accounting for internal melt correctly can be quite challenging as it requires
detailed knowledge of the vertical structure of internal melt and bulk density. Nevertheless, we find
that although the effects of internal melt and density variation may be significant when considered
individually, if considered together, they are likely small.

If internal melt is uniform throughout the vertical ice column, the only effect is a gradual reduction
in $\rho_b$ over the course of the melt season, slightly increasing the pond growth rate. If, on the other
hand, internal melt has a vertical structure, it will create a vertically non-uniform bulk ice density
which can have more complicated effects on pond evolution. Variations in bulk density and internal
melt affect pond evolution in the following ways: 1) mass transported across sea level due to rigid
body movement depends on the bulk density at sea level, 2) the volume of ice removed by local melt
depends on the bulk ice density at the surface, 3) freeboard height depends on average bulk densities
above and below sea level, and 4) internal melt induces rigid body motion by melting mass above and
below sea level, without changing the ice surface. We outline the procedure to include these effects
in the pond evolution model in appendix D. The resulting equation for pond coverage evolution has
the same form as Eq. (25), with only the strengths modified. Here, we only qualitatively discuss our
findings. Pond evolution is most sensitive to

1. The difference between the internal melt rate above and below sea level, $\overline{e}_{\mathrm{asl}} - \overline{e}_{\mathrm{bsl}}$, creating a
   rigid body motion. Here, $\overline{e}_{\mathrm{asl/bsl}}$ is the energy density used for internal melting, averaged over
   all ice above or below sea level. More internal melt above (below) the sea level will create an
   upward (downward) rigid body motion of the floe, slowing down (speeding up) pond growth.

2. The difference between the bulk ice density at the surface and the bulk ice density at sea level,
   $\rho_b(h) - \rho_b(0)$, changing the ratio of topographic change due to local melt to rigid body motion.
   Using default parameters, rigid body motion is upwards, slowing down pond growth. There-
   fore, a lower (higher) bulk ice density at the surface relative to sea level increases (decreases)
   the rate of local melt relative to rigid body motion, speeding up (slowing down) pond growth.

If considered as independent processes, vertical variations in bulk ice density and internal melt can
significantly alter the rate of pond growth. For example, assuming $\rho_b(0) = 850 \, \mathrm{kg \, m^{-3}}$, $\rho_b(h) = 750 \, \mathrm{kg \, m^{-3}}$, and no internal melt, leads to a roughly $60\%$ increase in the pond growth rate. However,
these processes depend on each other and have the opposite effects on pond evolution. For example a
high rate of internal melt above sea level, slowing down pond growth, will lower the bulk ice density
above sea level, speeding up pond growth.

Density and internal melt can be related via a differential equation, $\frac{\partial \rho_b(z)}{\partial t} = -\frac{e(z)}{l} - \frac{\partial \rho_b(z)}{\partial z} \frac{ds_{\mathrm{rigid\,body}}}{dt}$,
where $z$ is a vertical coordinate within the ice column. Assuming vertically uniform rates of internal
melt above and below sea level, an approximate long-time solution to this equation yields a verti-
cally uniform bulk density below sea level, and a linearly decreasing bulk density above sea level.

This also defines a long-time relationship between the vertical profiles of internal melt and bulk ice density, $\overline{e}_{\mathrm{asl}} - \overline{e}_{\mathrm{bsl}} = \frac{l}{h} \frac{ds_{\mathrm{rigid\,body}}}{dt} (\rho_b(0) - \rho_b(h))$. Using densities from the example in the paragraph above, and the rate of internal melt obtained in this way, leads to a roughly $10\%$ increase in pond growth rate, significantly less than $60\%$ we found when considering only the effects of vertical density structure.

A long-time effect of vertically non-uniform internal melt and density is always a significant compensation between the two, although there may be transient effects. For this reason, we believe that including a vertical structure of density or internal melt in the simple model of pond evolution model is most likely unnecessary.

### 7.3 Under certain conditions, ponds can stop growing

Here, we will entertain the possibility of pond growth by vertical motion of the topography stopping entirely for a period of time. This is an example of a possible transient effect of internal melting, which, although interesting, seems unlikely.

If there is enough mass removed above sea level to induce an upward rigid body motion that is able to compensate for the effects of local melting near the sea level, points near the sea level would move upwards, $\frac{ds}{dt} > 0$, and pond growth would stop. This could, for example, occur if there is strong internal melting above sea level. After a time, however, high internal melt above sea level would lower the bulk ice density at the surface thereby increasing the rate of local melt, and reinitializing pond growth.

We will use an equation for $\frac{ds}{dt}$ that includes the effects of vertically non-uniform internal melt and bulk ice density we derive in appendix D, Eq.(42). Requiring that $\frac{ds}{dt}(x) > 0$ for any $x$, we find the condition for pond growth stopping as

$$k < \frac{\rho_b(h)}{\rho_b(0)} \frac{\rho_i}{\rho_w} \left( 1 + \frac{h}{\overline{F}_{\mathrm{bi}}} (\overline{e}_{\mathrm{asl}} - \overline{e}_{\mathrm{bsl}} \frac{\overline{\rho}_{\mathrm{asl}}}{\overline{\rho}_{\mathrm{bsl}}}) \right) - \frac{\rho_w - \rho_i}{\rho_w} \frac{\overline{F}_{\mathrm{bot}}}{\overline{F}_{\mathrm{bi}}} , \tag{34}$$

where $\overline{\rho}_{\mathrm{asl/bsl}}$ is the average bulk density above and below sea level. Using the values of internal melt and bulk densities from the previous chapter and taking $\frac{\overline{\rho}_{\mathrm{asl}}}{\overline{\rho}_{\mathrm{bsl}}} \approx 1$, we find that in order for ponds to stop growing, $k$ has to be less than 0.85. This is unlikely as ice near the sea level likely melts faster than ice higher up. Nevertheless, if internal melt has not had enough time to adjust densities above and below sea level, it is possible that pond growth could be stopped for a time by the action of internal melt above sea level. For example, assuming the same internal melt as in the previous example but a uniform bulk ice density ($\rho_b(h) = \rho_b(0)$), pond growth would be stopped at $k = 1$. In this case it is likely that growth by lateral melt would take over, as Eq. (34) ensures only that pond growth by vertical motions is prevented.

## 8  Conclusions

We presented a simple analytical model for melt pond evolution on permeable Arctic sea ice. The model is represented by two ordinary differential equations in which the rate of change of pond coverage depends on pond coverage. The model is governed by four parameters, $S_{bi}$, $S_{mp}$, $S_{bot}$, and $S_{em}$, that control the rate of pond growth by bare ice melting, ponded ice melting, ice bottom melting, and enhanced melting. Using this model we are able to reproduce observations well.

Our main finding is that we can estimate the mean pond coverage as a function of time without running the model by using "effective strengths:" $S_{bi}^*$, $S_{mp}^*$, $S_{bot}^*$, and $S_{em}^*$. Here all the physical parameters combine in a known way which permits understanding of the behavior of pond coverage under general conditions. The most important conclusions we draw from analyzing the effective strengths are:

1. Ponds grow slower on smoother ice, with freeboard sinking roughly proportional to the square of the bare ice roughness and enhanced melting increasing roughly linearly with roughness.

2. Ponds respond to both freeboard sinking and enhanced melting on first year ice and almost entirely to enhanced melting on multiyear ice.

3. The pond growth rate is more sensitive to changes in bare sea ice albedo than changes in pond albedo unless the ice is already mostly covered in ponds.

4. Under a global warming scenario, the pond feedback could lead to ice thinning comparable to thinning due to direct forcing.

5. The dependence of ice albedo on height above sea level is likely a significant control on pond evolution.

6. The pond coverage distribution over an ensemble of floes likely narrows over time.

7. Pond evolution is insensitive to small time scale variations in the forcing.

8. If freeboard sinking is suppressed by topography, lateral melting likely plays an important role, making it a significant factor on multiyear ice.

9. The complicated physics of lateral melting can be summarized by a single non-dimensional constant $K_{lat}$ that relates the lateral melt flux to the flux used for melting the pond bottom.

10. The vertical structure of density and internal melt can likely be ignored.

As melt pond coverage is one of the key controls on summer Arctic sea ice albedo, some representation of it in GCMs is necessary for predicting the future of sea ice and its impact on global climate. With the exception of enhanced melting, our model depends only on parameters that are

either available in large scale models or that can be reasonably estimated. Therefore, if stricter constraints can be placed on the strength of enhanced melting, our model may present an accurate and
835 computationally low-cost representation of sea level melt ponds that could be used in GCMs.

## 9 Appendix A

A good fit to measured hypsographic curves is a tangent function (Fig. 2):

$$s(x_h) = a\left[\tan\left(\frac{\pi}{2m}p_1\big((x_h - x_i) - p_2(1 - x_i)\big)\right) + \tan\left(\frac{\pi}{2m}p_1p_2(1 - x_i)\right)\right]$$
$$m \equiv \max\left(p_2(1 - x_i), (1 - x_i) - p_2(1 - x_i)\right) \tag{35}$$

Although this function has a cumbersome form, the parameters involved have a clear interpretation.
The requirement that the initial pond fraction is at $x_h = x_i$ is automatically satisfied as this is a zero
of the function Eq. (35). The parameter $a$ is determined by the requirement of hydrostatic balance,
$< s(x_h) >= h$. Therefore, after specifying the initial pond fraction, $x_i$, and the initial ice thickness
$H$, the only two unconstrained parameters are $p_1$ and $p_2$. Parameter $0 < p_1 < 1$ determines the level
of "variability" of the curve: if $p_1$ is close to 0, $s(x_h)$ is roughly linear, whereas if $p_1$ is close to 1,
$s(x_h)$ is highly curved. Parameter $p_2$ determines the position of the inflection point of the tangent
function relative to $x_i$. Therefore $p_2 < 0$ means that the inflection point is to the left of $x_i$, and $s(x_h)$
is fully convex. For $p_2 > 1$, the inflection point is to the right of $x_h = 1$, and $s(x_h)$ is fully concave.
If $0 < p_2 < 1$, $s(x_h)$ transitions from concave to convex at $x_h = x_i + p_2(1 - x_i)$. We note that the
non-dimensional bare ice roughness, $\hat{\sigma}$, for a hypsographic curve defined in this way does not depend
on ice thickness or initial pond coverage, but only on parameters $p_1$ and $p_2$. For the hypsographic
curve measured by Landy et. al. (2014) for June 25th of 2011, the values of the shape parameters
are $p_1 \approx 0.8$ and $p_2 \approx 0.4$, whereas for the hypsographic curve measured during SHEBA (Fig 2a,
dashed line) the parameters are $p_1 \approx 0.9$ and $p_2 \approx 0.5$.

## 10 Appendix B

In order to make a connection between a model where a constant fraction of bare ice, $\delta$, is affected
by enhanced melting, and a model where ice below a fixed elevation, $\Delta s$, is affected, we need
to estimate how $\delta$ scales with $\Delta s$. It is important to make this connection since several physical
mechanisms that significantly affect the melt rate depend on the elevation of ice above sea level. To
do this, we will use an alternative model where we assume both freeboard sinking and enhanced
melting occur simultaneously, and enhanced melting only affects ice below $\Delta s$ (Fig. 3d). We define
$x_s$ to be the fraction of ice below $\Delta s$, $x$ to be the fraction of the ice below sea level, and $\delta \equiv x_s - x$
to be the difference between the two. $x_s$ evolves only due to freeboard sinking, whereas $x$ evolves
due to both freeboard sinking and enhanced melting. The equations for the evolution of $x_s$ and $x$ are

$$\frac{dx_s}{dt} = \frac{dx_h}{ds}\Big|_{x_s} \frac{ds_{\text{fs}}}{dt}$$

$$\frac{dx}{dt} = \frac{dx}{ds}\left[\frac{ds_{\text{fs}}}{dt} + \frac{ds_{\text{em}}}{dt}\right]. \tag{36}$$

Here, $\frac{ds_{\text{fs}}}{dt}$ and $\frac{ds_{\text{em}}}{dt}$ are determined by Eq. (11). Since freeboard sinking does not change the shape of the topography and $x_s$ evolves only due to freeboard sinking, $\frac{dx_h}{ds}\big|_{x_s}$ is simply the inverse slope of the original hypsographic curve evaluated at $x_s$. On the other hand, the hypsographic curve near sea level is affected by enhanced melting, and therefore changes shape over time. For this reason, $\frac{dx}{ds}$, which relates the change in pond fraction, $dx$, to the vertical change in the hypsographic curve at sea level, $ds$, changes with time. Nevertheless, if $\Delta s$ is small enough, we can approximate the hypsographic curve between $x$ and $x_s$ to be a straight line, meaning that $\frac{dx}{ds} \approx \frac{x_s - x}{\Delta s} = \frac{\delta}{\Delta s}$. This approximation closes our alternative model. This model provides a similar level of agreement with the 1D model as the 0D model Eq. (25), but is more complicated to analyze. For this reason, we focus on Eq. (25) to analyze pond evolution, and use Eq. (36) only in what follows. We note that if the hypsographic curve is convex, Eq. (36) agrees better with the 1D model than Eq. (25). This configuration is, however, unrealistic.

Using $\frac{dx}{ds} = \frac{\delta}{\Delta s}$, and subtracting $\frac{dx}{dt}$ from $\frac{dx_s}{dt}$ in Eq. (36), we get an equation for evolution of $\delta$:

$$\frac{d\delta}{dt} = \frac{dx_h}{ds}\Big|_{(x+\delta)} \frac{ds_{\text{fs}}}{dt} - \frac{\delta}{\Delta s}\left[\frac{ds_{\text{fs}}}{dt} + \frac{ds_{\text{em}}}{dt}\right]. \tag{37}$$

Since $\frac{ds_{\text{fs}}}{dt} + \frac{ds_{\text{em}}}{dt}$ is larger than $\frac{ds_{\text{fs}}}{dt}$, $\delta$ decreases until it reaches a constant value after some time. Therefore, a constant $\Delta s$ model and a constant $\delta$ model become equivalent after some time. Therefore, finding the value of $\delta$ for which $\frac{d\delta}{dt} = 0$, represents a natural way to relate the two models.

The values of $\frac{ds_{\text{fs}}}{dt}$, $\frac{ds_{\text{em}}}{dt}$, and $\frac{dx_h}{ds}\big|_{(x+\delta)}$ themselves depend on pond fraction, $x$ (Eq. (11)). Furthermore, $\frac{ds_{\text{fs}}}{dt}$ and $\frac{ds_{\text{em}}}{dt}$ depend on the energy fluxes used for melting the ice, which may fluctuate in time. For these reasons, $\delta$ is never fully constant. To deal with this this, we estimate the magnitudes of $\frac{ds_{\text{fs}}}{dt}$, $\frac{ds_{\text{em}}}{dt}$, and $\frac{dx_h}{ds}\big|_{(x+\delta)}$ by substituting $x \to x_i$, $\frac{dx_h}{ds}\big|_{(x+\delta)} \to \frac{1-x_i}{h}$, and energy fluxes, $\overline{F}$, with their representative values, $|\overline{F}|$, e.g. their time-averages. We then find the magnitude of $\delta$ as

$$\delta = C\frac{\rho_{\text{w}}}{\rho_{\text{w}} - \rho_{\text{i}}} \frac{\Delta s(1 - x_i)^2}{H} \frac{1}{\frac{ds_{\text{em}}}{ds_{\text{fs}}} + 1}, \tag{38}$$

where $C$ is a non-dimensional number that does not depend on physical parameters, there to compensate for the crude approximations of using only the initial pond fraction and the average slope of the hypsographic curve. Comparing to 1D model, we find $C \approx \frac{2}{3}$. The term $\frac{ds_{\text{em}}}{ds_{\text{fs}}}$ is the ratio of magnitudes of $\frac{ds_{\text{em}}}{dt}$ and $\frac{ds_{\text{fs}}}{dt}$, and is given by

$$\frac{ds_{\text{em}}}{ds_{\text{fs}}} = \frac{\rho_w}{\rho_w - \rho_i} \frac{|\overline{F}_{\text{bi}}|(k-1)}{|\overline{F}_{\text{bi}}| + \frac{x_i}{1-x_i}|\overline{F}_{\text{mp}}| + \frac{1}{1-x_i}|\overline{F}_{\text{bot}}|}. \tag{39}$$

Using $\delta$ defined in this way in the 0D model, Eq. (25), provides excellent agreement with Eq. (36) and the 1D model run with constant $\Delta s$. We note that this agreement is reached in the long-time

limit, and for times shorter than roughly $\frac{\Delta s}{(ds_{em}/dt + ds_{fs}/dt)}$ some disagreement can persist. Although the magnitude of the disagreement depends on the shape of the hypsographic curve, it is typically not very large, and the 0D model provides a reasonable estimate of pond evolution even for short times.

## 11 Appendix C

Here we describe the procedure we used to estimate the effective strengths, Eq. (28). We write the effective strengths as

$$S^* = f(\hat{\sigma}, \hat{t})S \,, \tag{40}$$

where $f(\hat{\sigma}, \hat{t})$ is a non-dimensional function of non-dimensional roughness $\hat{\sigma}$ and non-dimensional time $\hat{t} \equiv \frac{St}{1-x_i}$, and $S$ is either $S_{fs} \equiv (S_{bi} + S_{mp} + S_{bot})$ in the case of freeboard sinking or $S_{em}$ in case of enhanced melting. The non-dimensional time, $\hat{t}$, defined in the above way measures how far the melt season has progressed, with $\hat{t} = 0$ corresponding to the beginning of pond growth and $\hat{t} = 1$ roughly corresponding to the end of pond growth with entire floe flooded. The function $f(\hat{\sigma}, \hat{t})$ measures how much the mean pond coverage deviates from a mean coverage of linearly evolving ponds. For a linear pond evolution, $x(t) = St + x_i$, the function $f(\hat{\sigma}, \hat{t}) = 1$.

We separately consider freeboard sinking and enhanced melting. For all the curves in Figs. 7a and b, we find $f(\hat{\sigma}, \hat{t})$ at several different times $\hat{t}$ as $f(\hat{\sigma}, \hat{t}) = 2\frac{<x(t)> - x_i}{St}$. We show the results in Figs. 11a and b, where $f$ are plotted as functions of roughness and different colors correspond to different times $\hat{t}$. For any given time, the scatter comes from the fact that the hypsographic curve is not fully determined by roughness.

In the case of freeboard sinking, $f_{fs}$ does not depend much on $\hat{t}$. A quadratic $f_{fs}(\hat{\sigma}, \hat{t}) = c\hat{\sigma}^2$ fits the scatter data well. Based on best fit estimates, we find $c \approx 1.3$ (Fig. 11a, red dashed line).

In the case of enhanced melting, $f_{em}$ depends strongly on time $\hat{t}$. We choose to parameterize $f_{em}$ with a linear function of the form $f_{em}(\hat{\sigma}, \hat{t}) = 1 + c(\hat{t})\hat{\sigma}$. We can approximate $c(\hat{t})$ by exactly solving the equation for enhanced melting, Eq. (19), for a linear hypsographic curve, $s(x_h) \propto (x_h - x_i)$. Finding the roughness and $< x(t) >$ in this case, we find $c(\hat{t}) \approx \left(\frac{2}{\sqrt{\hat{t}}} - \frac{3}{2}\right)$. Red dashed lines in Fig. 7d show $f_{em}$ parameterized in this way.

## 12 Appendix D

Here, we outline the procedure to include the effects of vertically non-uniform internal melt and bulk ice density. We assume that the bulk ice density, $\rho_b$, and the energy density used for melting the ice internally, $e$, have a vertical structure, $\rho_b(z)$ and $e(z)$, where $z$ is positive upwards, $z = 0$ corresponds to sea level, and $z = h$ corresponds to ice surface.

Mass transported across sea level depends on the bulk density at the sea level, the rate of local melting depends on the bulk ice density at the surface, and the freeboard height depends on the average densities above and below sea level, $\overline{\rho}_{\text{asl/bsl}}$. Internal melt above and below sea level creates a rigid body motion. This is summarized as

$$dm^{\text{rigid body}} = \rho_b(0) A_{\text{bi}} ds_{\text{rigid body}} \,,$$

$$dm^{\text{melt}}_{\text{above s. l.}} = -A_{\text{bi}} \frac{\overline{F}_{\text{bi}}}{l} dt - A_{\text{bi}} h \frac{\overline{e}_{\text{asl}}}{l} dt \,,$$

$$dm^{\text{melt}}_{\text{below s. l.}} = -A_{\text{mp}} \frac{\overline{F}_{\text{mp}}}{l} dt - A \frac{\overline{F}_{\text{bot}}}{l} dt - A H_d \frac{\overline{e}_{\text{bsl}}}{l} dt \,,$$

$$h = \frac{\rho_w - \rho_i}{\rho_w} \frac{H}{1-x} \frac{1}{1 - \Delta\rho_b \frac{\rho_i}{\rho_w}} \,,$$

$$\frac{ds_{\text{loc}}}{dt}(\boldsymbol{r}) = -k(\boldsymbol{r}) \frac{\overline{F}_{\text{bi}}}{l\rho_b(h)} \,, \tag{41}$$

where $H_d$ is the ice draft depth defined as the volume of ice below sea level divided by the area of the ice floe, $\overline{e}_{\text{asl/bsl}}$ is the energy density used for internal melting averaged over all ice above or below sea level, and $\Delta\rho_b \equiv \frac{\overline{\rho}_{\text{bsl}} - \overline{\rho}_{\text{asl}}}{\overline{\rho}_{\text{bsl}}}$ is the relative difference in mean bulk density above and below sea level.

With these changes, we can find the equation for pond coverage evolution straightforwardly, by repeating all of the steps from section 2. We first derive the equation for the vertical motion of points near the sea level

$$\frac{ds}{dt} = -\left[ (k-1) \frac{\overline{F}_{\text{bi}}}{l\rho_b(h)} \right] -$$

$$- \left[ \frac{1}{l\rho_b(0)} \left( \overline{F}_{\text{bi}} \left( \frac{\rho_b(0)}{\rho_b(h)} - \frac{\rho_i}{\rho_w} \right) + \frac{(\rho_w - \rho_i)x}{\rho_w(1-x)} \overline{F}_{\text{mp}} + \frac{\rho_w - \rho_i}{\rho_w(1-x)} \overline{F}_{\text{bot}} + \frac{\rho_i}{\rho_w} h \overline{e}_{\text{bsl}} (\Delta e - \Delta\rho_b) \right) \right] \,, \tag{42}$$

where $\Delta e \equiv \frac{\overline{e}_{\text{bsl}} - \overline{e}_{\text{asl}}}{\overline{e}_{\text{bsl}}}$ is the relative difference in average energy density used for internal melting below and above sea level. The two terms in square brackets correspond to enhanced melting and freeboard sinking. Then we repeat the procedure to relate Eq. (42) to the change in pond coverage. The resulting equation has the same form as Eq. (25), with only the strengths modified

$$S_{\text{int}} = \frac{(1-x_i) \frac{\rho_i}{\rho_w} \overline{e}_{\text{bsl}} (\Delta e - \Delta\rho_b)}{l\rho_b(0)} \,,$$

$$S_{\text{bi}} = \frac{(1-x_i)^2 (1 - \Delta\rho_b \frac{\rho_i}{\rho_w})(1 + \frac{\rho_w(\rho_b(0) - \rho_b(h))}{\rho_b(h)(\rho_w - \rho_i)})}{H l \rho_b(0)} \overline{F}_{\text{bi}} \,,$$

$$S_{\text{mp}} = \frac{(1-x_i) x_i (1 - \Delta\rho_b \frac{\rho_i}{\rho_w})}{H l \rho_b(0)} \overline{F}_{\text{mp}} \,,$$

$$S_{\text{bot}} = \frac{(1-x_i)(1 - \Delta\rho_b \frac{\rho_i}{\rho_w})}{H l \rho_b(0)} \overline{F}_{\text{bot}} \,,$$

$$S_{\text{em}} = \left( \frac{\rho_w}{\rho_w - \rho_i} \right)^2 \frac{2\Delta s (1-x_i)^3 (k-1)(1 - \Delta\rho_b \frac{\rho_i}{\rho_w})^2}{3H^2 l \rho_b(0)(1 + \frac{ds_{\text{em}}}{ds_{\text{fs}}})} \overline{F}_{\text{bi}} \,. \tag{43}$$

Here, the strength of internal melting, $S_{\text{int}}$ should be included in the equation for freeboard sinking. The term $\frac{ds_{\text{em}}}{ds_{\text{fs}}}$ is given by the ratio of the two terms in Eq. 42. The equation for pond growth, Eq.

(25), using the above strengths, Eq. (43), should also be supplemented with an equation for evolution of bulk density

$$950 \quad \frac{\partial \rho_b(z)}{\partial t} = -\frac{e(z)}{l} - \frac{\partial \rho_b(z)}{\partial z}\frac{ds_{\text{rigid body}}}{dt} \ . \tag{44}$$

*Acknowledgements.* We thank B. Cael Barry, Daniel Koll, Edwin Kite, and Mary Silber for reading the paper and giving useful comments. We also thank Douglas MacAyeal, and Edwin Kite for discussions and ideas about the physical mechanisms involved. We thank Chris Polashenski for an extensive review and useful discussions that greatly improved this manuscript and for providing data on the dependence of the albedo on surface elevation. We also thank an anonymous reviewer. We thank Don Perovich for providing data from the SHEBA mission. Predrag Popović was supported by a NASA Earth and Space Science Fellowship. This work was partially supported by the National Science Foundation under NSF award number 1623064 and under NSF award number 0940261, which is part of the Mathematics and Climate Research Network.

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

| Parameter | Meaning |
|---|---|
| $t, \hat{t}$ | Time and non-dimensional time, $\hat{t} = \frac{St}{1-x_i}$. |
| $s(\boldsymbol{r})$ | Surface elevation above sea level at point $\boldsymbol{r}$. |
| $s(x_h), \widehat{s}(x_h), \frac{d\widehat{s}}{d\widehat{x}_h}$ | Hypsographic curve, non-dimensional hypsographic curve, $\widehat{s}(x_h) = \frac{s(x_h)}{h}$, and its non-dimensional derivative, $\frac{d\widehat{s}}{d\widehat{x}_h} = \frac{1-x_i}{h}\frac{ds}{dx_h}$. |
| $ds_{\text{rigid body}}, ds_{\text{loc}}(\boldsymbol{r})$ | Change in surface elevation due to rigid body motion and due to local melting at point $\boldsymbol{r}$. |
| $ds_{\text{fs}}, ds_{\text{em}}, ds_{\text{em}}/ds_{\text{fs}}$ | Change in surface elevation due to freeboard sinking, due to enhanced melting, and the magnitude of their ratio. |
| $dm_{\text{above/below s. l.}}^{\text{melt/rigid body}}$ | Change in mass above and below sea level due to ice melting or rigid body motion. |
| $x, \widehat{x}, \widehat{1-x}$ | Pond fraction, normalized pond fraction $\widehat{x} = \frac{x}{x_i}$, and normalized bare ice fraction, $\widehat{1-x} = \frac{1-x}{1-x_i}$ |
| $x_i$ | Initial pond fraction. |
| $x_h$ | Fraction of ice below an elevation given by the hypsographic curve. |
| $x_s$ | Fraction of ice below $\Delta s$. |
| $x_{\text{fs}}(t), x_{\text{em}}(t), x_{\text{lat}}(t)$ | Pond coverage evolution due to freeboard sinking, enhanced melting, and lateral melting. |
| $A, A_{\text{bi}}, A_{\text{mp}}$ | Areas of the floe, bare ice, and melt ponds |
| $P$ | Total perimeter of the ponds. |
| $\rho_w, \rho_i, \rho_b$ | Densities of salt water, pure ice, and bulk ice once all the brine has drained. |
| $l$ | Latent heat of melting. |
| $H, h$ | Initial thickness of the ice and average initial freeboard height. |
| $\sigma, \hat{\sigma}$ | Bare ice roughness and non-dimensional bare ice roughness, $\hat{\sigma} = \frac{\sigma}{h}$. |
| $p_1, p_2$ | Shape parameters of the hypsographic curve that control the "amount of variability" of the curve and the location of the inflection point. |
| $k(\boldsymbol{r})$ | Ratio of the melt rate at point $\boldsymbol{r}$ to the average rate of bare ice melting. |
| $\Delta s$ | Height above sea level below which there is enhanced melting. |
| $\delta$ | Fraction of the ice affected by enhanced melting. |
| $\alpha_{\text{bi}}, \alpha_{\text{mp}}$ | Albedos of bare ice and melt ponds. |
| $F_{\text{sol}}, F_{\text{r}}$ | Solar energy flux and the sum of longwave, latent, and sensible heat fluxes. |
| $\overline{F}_{\text{bi}}, \overline{F}_{\text{mp}}, \overline{F}_{\text{bot}}, \overline{F}_{\text{lat}}$ | Fluxes of energy used for melting bare ice, ponded ice, ice bottom, and lateral melting averaged over bare ice, ponded ice, ice bottom, and the pond perimeter. |
| $|\overline{F}|$ | Representative values of fluxes, e.g. their time-averages. |
| $K_{\text{lat}}$ | Constant relating the flux of energy used for melting ponded ice to the flux of energy used for lateral melting. |
| $S_{\text{bi}}, S_{\text{mp}}, S_{\text{bot}}, S_{\text{em}}$ | Strengths of bare ice melting, ponded ice melting, ice bottom melting, and enhanced melting. |
| $S_{\text{bi}}^*, S_{\text{mp}}^*, S_{\text{bot}}^*, S_{\text{em}}^*$ | Effective strengths of bare ice melting, ponded ice melting, ice bottom melting, and enhanced melting, that take into account the effects of bare ice roughness. |
| $S_{\text{fs}}^*$ | Effective strength of freeboard sinking, $S_{\text{fs}}^* = S_{\text{bi}}^* + S_{\text{mp}}^* + S_{\text{bot}}^*$. |
| $S^*$ | Total effective strength, $S^* = S_{\text{bi}}^* + S_{\text{mp}}^* + S_{\text{bot}}^* + S_{\text{em}}^*$. |

a) Local Displacement

$ds_{loc}/dt$

b) Rigid body displacement

$ds_{rigid\ body}/dt$

**Figure 1.** a) Local displacement represents the movement of a point on the ice surface as a result of ice melting at that particular point. It is a function only of local ice characteristics at that point. For both local and hydrostatic displacements the positive direction is defined as upwards. b) Rigid body displacement represents the motion of a floe as a whole in an effort to maintain hydrostatic balance because melting removes mass above or below sea level. Melting above sea level induces an upward motion of the floe, whereas melting below sea level induces a downward motion.

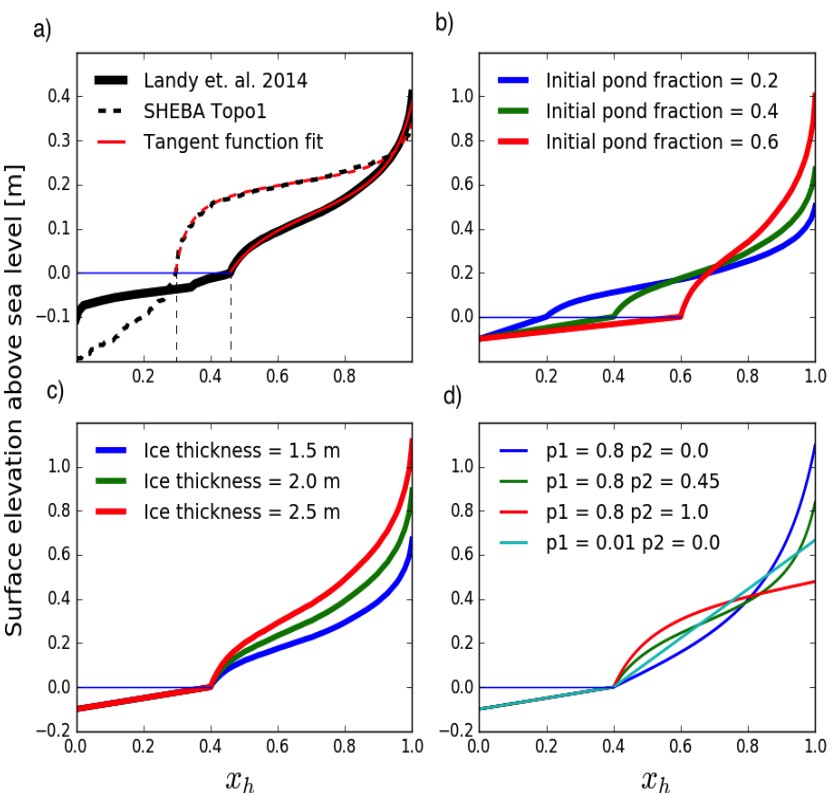

**Figure 2.** Hypsographic curves showing the percentage of the sea ice surface that is lower than a particular elevation. Pond coverage on highly permeable sea ice can be inferred from here as the intersection of sea level (horizontal blue line) with the hypsographic curve. a) A hypsographic curve measured by Landy et. al. (2014) on June 25th of 2011 (solid black line), and a hypsographic curve measured during SHEBA along a 100m long "topography profile 1" on July 10th 1998 (black dashed line). The vertical dashed lines represent the pond coverage, assuming that ice is permeable. The red line represents a fit to the part of the hypsographic curve above sea level with a tangent function, Eq. (35). b) Adjusted hypsographic curves for different initial pond coverage, and the same ice thickness. c) Adjusted hypsographic curves for the same initial pond coverage and different ice thickness. d) Hypsographic curves for different shape parameters, $p_1$ and $p_2$, defined and discussed in appendix A, Eq. (35). Parameter $p_1$ controls the amount of curvature, while $p_2$ controls the position of the inflection point of the tangent function.

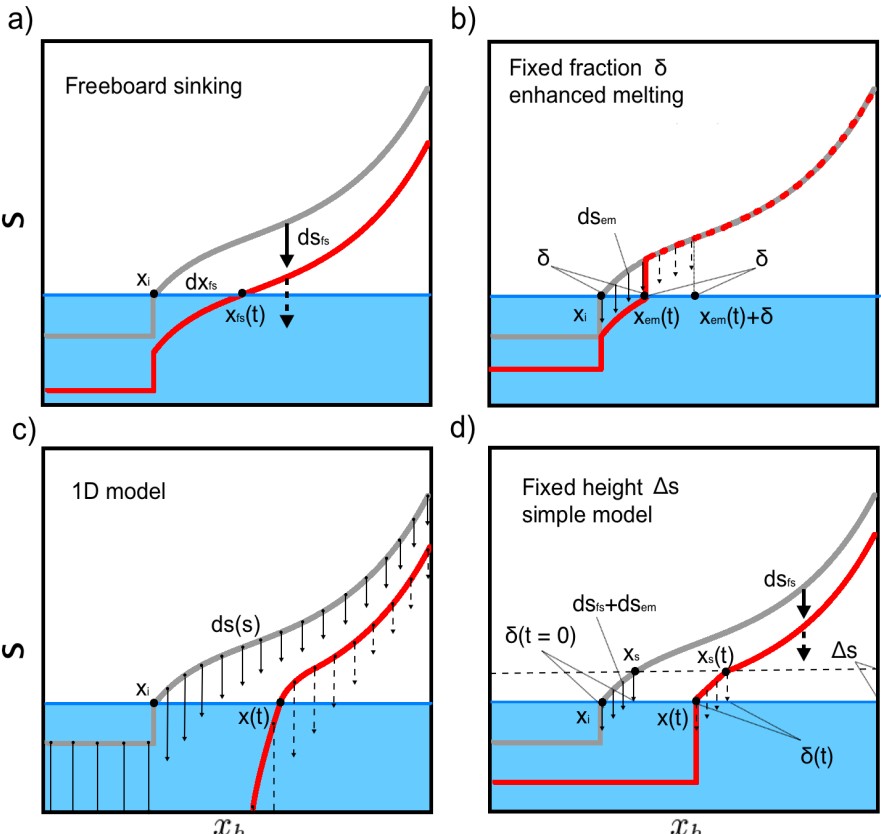

**Figure 3.** Explanation of different models of pond growth. Models evolve a hypsographic curve, $s(x_h)$, above sea level to find the pond coverage evolution. Evolution of the hypsographic curve below sea level is not relevant for pond growth and, apart from the 1D model, is not captured well in these models. a) Freeboard sinking shifts the entire hypsographic curve downward following a displacement of $ds_{\text{fs}}$. b) Enhanced melting acts on a constant ice fraction, $\delta$, and there is no freeboard sinking. The hypsographic curve changes only between $x_h = x$ and $x_h = x + \delta$, and remains unchanged otherwise. After a time $\Delta t = \frac{s(x+\delta)}{ds_{\text{em}}/dt}$ pond coverage grows by $\delta$. The 0D model, Eq. (25), assumes that the total pond evolution is the sum of pond evolution due to such enhanced melting and freeboard sinking (panel a). c) The 1D model prescribes a melt rate at each point on the hypsographic curve as a function of height above sea level, $\frac{ds}{dt}(s)$. d) A simplified model that assumes both freeboard sinking and enhanced melting (appendix B). Enhanced melting occurs only below height $\Delta s$. After some time, the fraction of ice affected by enhanced melting, $\delta$, becomes constant, meaning that a constant fraction model (panel b) and a constant height model are equivalent if $\delta$ and $\Delta s$ are related appropriately.

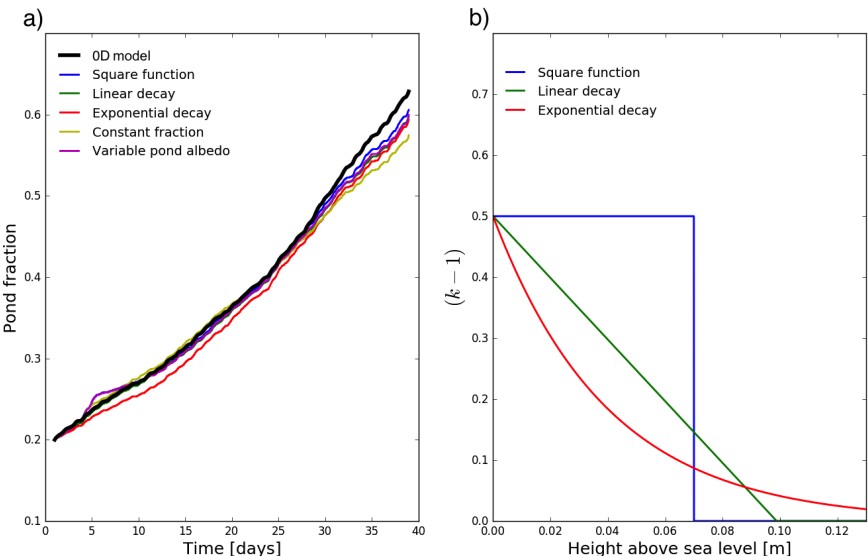

**Figure 4.** a) A comparison between pond evolution in the 0D model and the 1D model. The black curve represents the 0D model. The blue, green, and red curves represent the 1D model for different functions $k(s)$ shown in panel b). These different functions were chosen such that the integral parameter $< S_{\mathrm{em}} >$ (Eq. (24)) is the same as for the 0D model. The yellow curve represents the 1D model where enhanced melting acts on a constant fraction of bare ice, $\delta$, chosen according to Eq. (21). The magenta curve represents the 1D model with pond albedo varying with depth. There is significant agreement between all of the curves, suggesting that the simplifications made in the simple model were justified. Since including variable pond albedo does not change the pond evolution significantly, this detail can be neglected when estimating the pond coverage on permeable ice. b) The blue, green, and red lines represent functions $k(s) - 1$ used to run the 1D model.

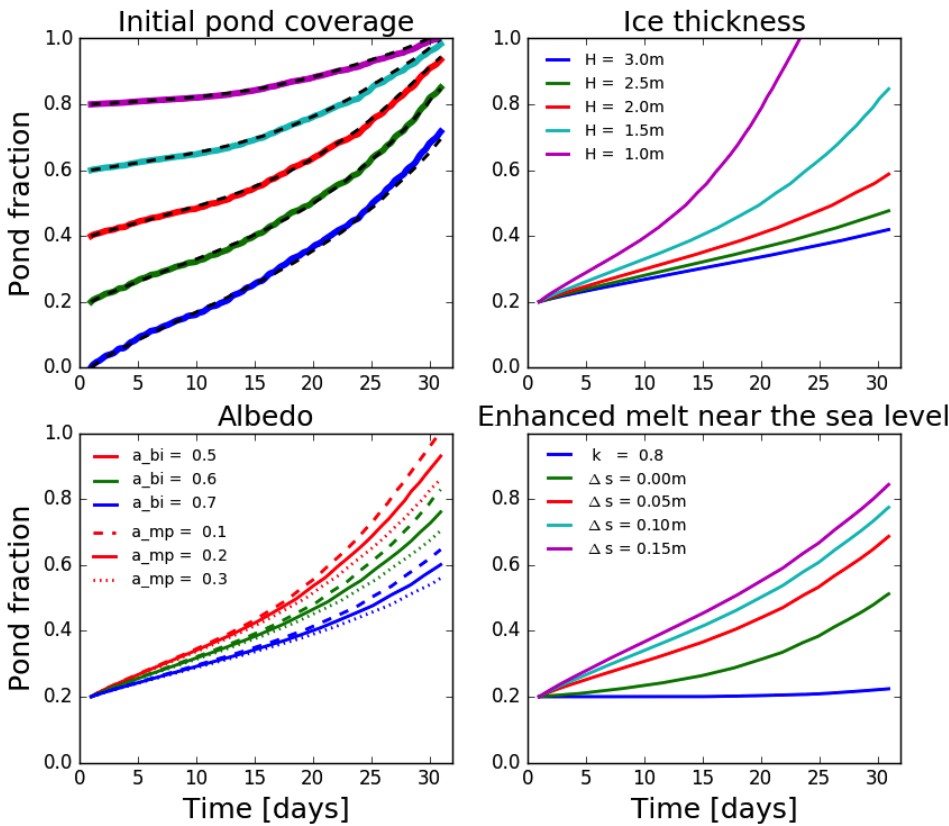

**Figure 5.** Numerical solutions to Eq. (25) with parameters varied around the defaults described in the text. a) Varying initial pond coverage. Solid lines represent solutions using full time-varying fluxes, while dashed lines represent solutions using time-averaged fluxes. The two solutions are very similar, so we subsequently use only the time-averaged fluxes. b) Varying ice thickness. Ponds grow slower on thicker floes. c) Varying pond and bare ice albedo. Different colors represent different bare ice albedos, and full, dotted, and dashed lines represent different pond albedos. A change in bare ice albedo has a much larger effect on pond fraction than the same change in pond albedo. d) Varying the $\Delta s$ and $k$. For $k = 0.8$, the ponds shrink. However, pond evolution for $k < 1$ is not represented well in our model, so this curve serves only as an illustration.

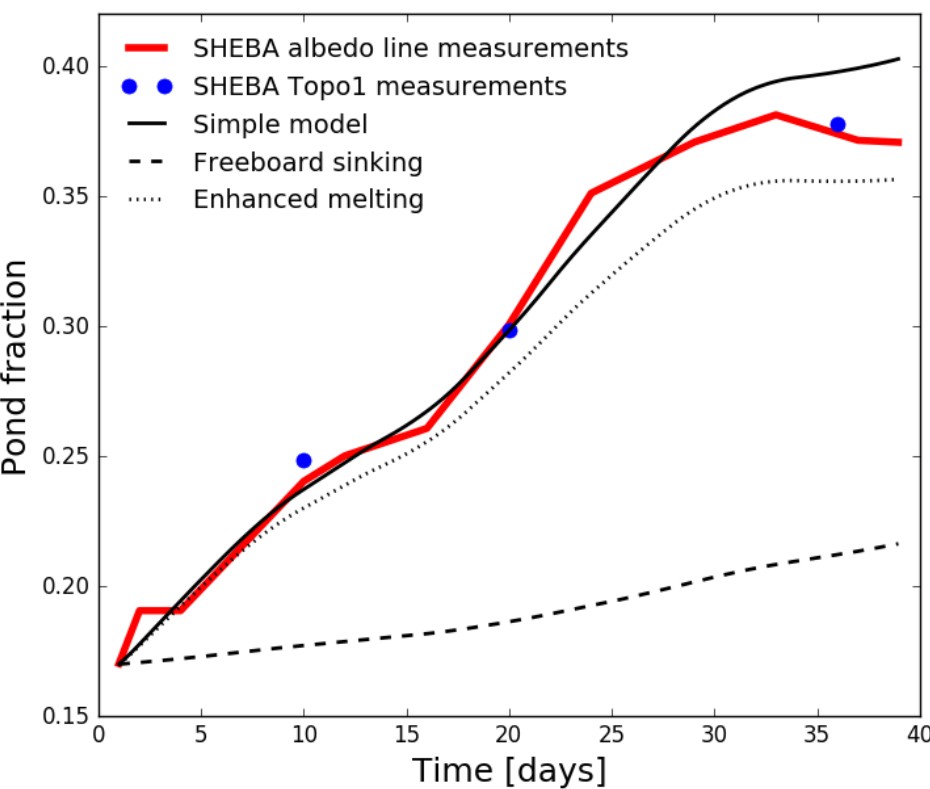

**Figure 6.** A comparison between measurements of pond fraction made during SHEBA along the albedo line (red line), along a topography profile (blue dots), and our model (black line). The blue dots have been shifted downward by 0.05 to make a more obvious comparison between albedo line and topography profile trends. The black dashed line is the contribution to our model from freeboard sinking and the black dotted line is the contribution from enhanced melting. Ponds grow almost entirely due to enhanced melting as a result of the steep topography of multiyear ice.

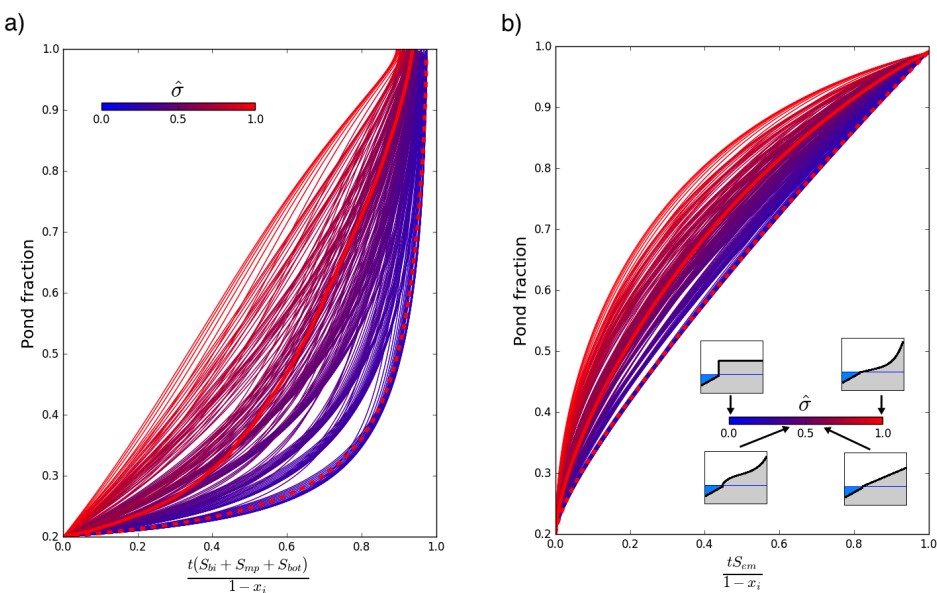

**Figure 7.** Exploring the effects of sea ice roughness. a) Pond evolution due to pure freeboard sinking for hypsographic curves with different shape parameters $p_1$ and $p_2$. The x-axis shows non-dimensional time $\hat{t} = \frac{t(S_{bi} + S_{mp} + S_{bot})}{1 - x_i}$. Color represents normalized roughness, $\hat{\sigma}$, with blue colors corresponding to small $\hat{\sigma}$ and red colors corresponding to large $\hat{\sigma}$. Thick red solid line represents pond evolution on the measured first year ice hypsographic curve, and the thick red dashed line represents pond evolution on the measured multiyear ice hypsographic curve. All else equal, rougher ice has a larger pond fraction. b) Pond evolution due to pure enhanced melting for hypsographic curves with different shapes. The x-axis shows non-dimensional time $\hat{t} = \frac{t S_{em}}{1 - x_i}$. Cartoon examples of hypsographic curves and their approximate positions along the $\hat{\sigma}$-axis are also shown.

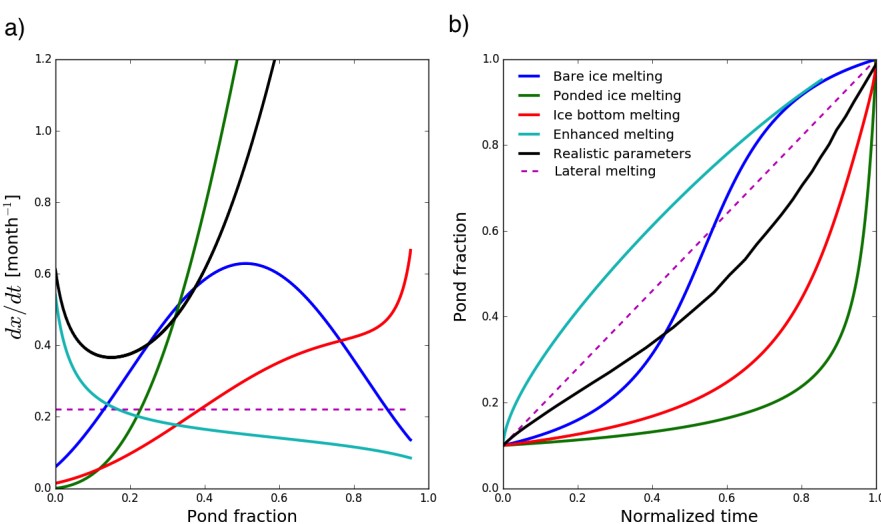

**Figure 8.** a) Dependence of growth rate on pond coverage for different modes of pond growth. The y-axis shows the growth rate, $\frac{dx}{dt}$, for each of the growth modes calculated using the default parameters and $x_i = 0$. Pond growth rate for bare ice melting (blue line) first increases up to a certain pond coverage and then decreases. Ponded ice melting (green line) increases with pond coverage from $\frac{dx}{dt} = 0$ at $x = 0$ to very high values at high pond coverage. The ice bottom melting rate (red line) gradually increases with pond coverage. The vertical enhanced melting rate (cyan line) decreases with pond coverage. The black line represents a realistic combination of the four growth modes, and shows that pond growth is dominated by enhanced melting early in the season, and by freeboard sinking late in the season. The dashed magenta line represents lateral melting estimated using parameters described in section 7.1. b) Solutions to Eq. (25) when only one of the growth modes is active. The x-axis shows the normalized time, where 0 corresponds to the beginning of the melt and 1 to entire floe being flooded.

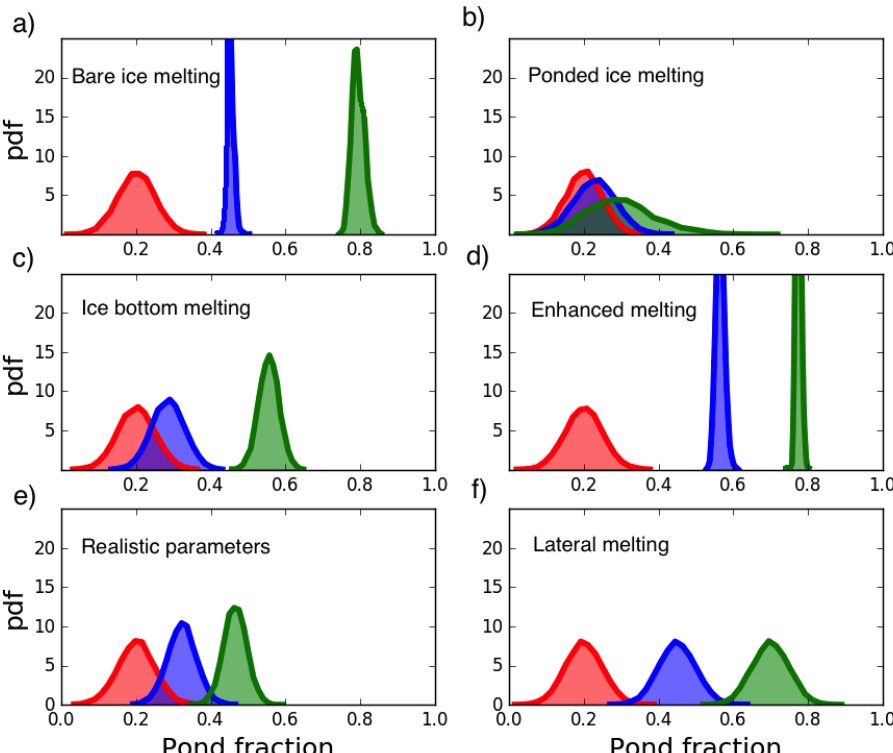

**Figure 9.** In this figure we have evolved an ensemble of $10^5$ floes with varying initial pond coverage according to Eq. (25) when only one of the growth modes is active. Red curves represent the initial pond fraction distribution, blue curves represent the pond fraction distribution after a time, $t$, while the green curves represent the pond fraction distribution after $2t$. A time used in panel a is $t = \frac{1}{2}\frac{1-x_i}{S_{\mathrm{bi}}}$, in panel b it is $t = \frac{1}{6}\frac{1-x_i}{S_{\mathrm{mp}}}$, and in panels c through f it is $t = \frac{1}{4}\frac{1-x_i}{S}$, where $x_i$ is the mean pond fraction of the initial distribution and $S$ is an appropriate strength. We show how different growth modes have different effects on the pond fraction distribution. a) Bare ice melting first narrows the distribution, and then widens it. b) Ponded ice melting widens the distribution. c) Bottom ice melting narrows the distribution, while the mean of the distribution increases at an increasing rate. d) Enhanced melting narrows the distribution, while the mean of the distribution increases at a decreasing rate. e) Using realistic parameters, the pond distribution slowly narrows and accelerates. f) Due to lateral melting, pond coverage distribution does not change width, and the growth is linear.

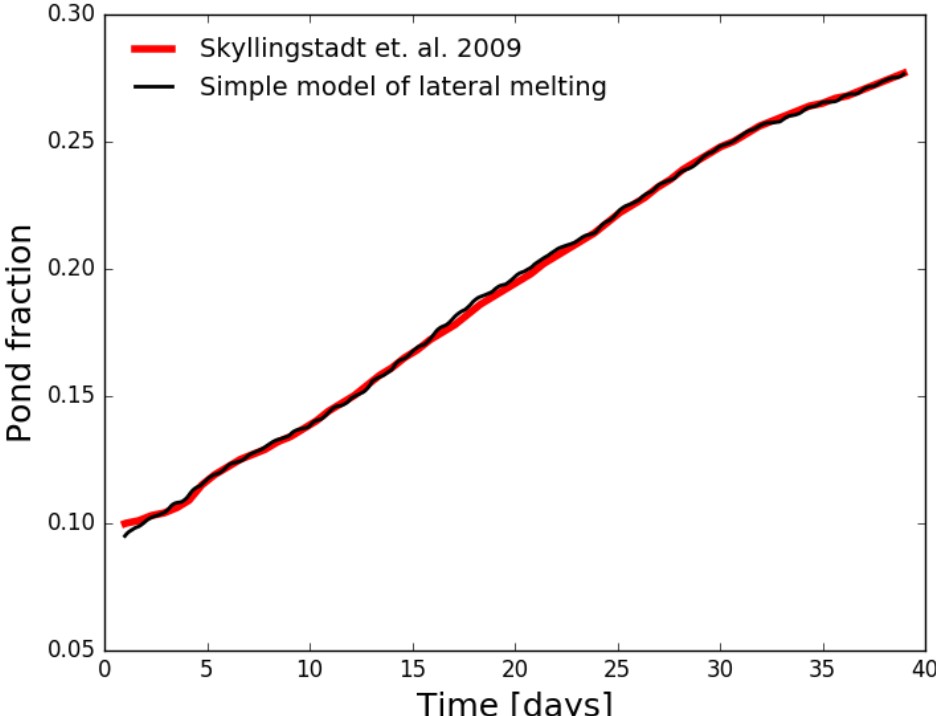

**Figure 10.** The red curve is the results of Skyllingstad et. al. (2009). The black curve is the solution to Eq. (33) with $\overline{F}_{\mathrm{lat}} = K_{\mathrm{lat}}\overline{F}_{\mathrm{mp}}$. The pond albedo and the shortwave, longwave, sensible, and latent heat fluxes used to find $\overline{F}_{\mathrm{mp}}$ are the same as used in Skyllingstad et. al. (2009) and $K_{\mathrm{lat}} = 1.5$. A nearly perfect agreement between the two curves suggests that a single non-dimensional constant, $K_{\mathrm{lat}}$, is enough to describe pond growth by lateral melting, and the complicated physics of lateral melting are important only in determining the value of $K_{\mathrm{lat}}$.

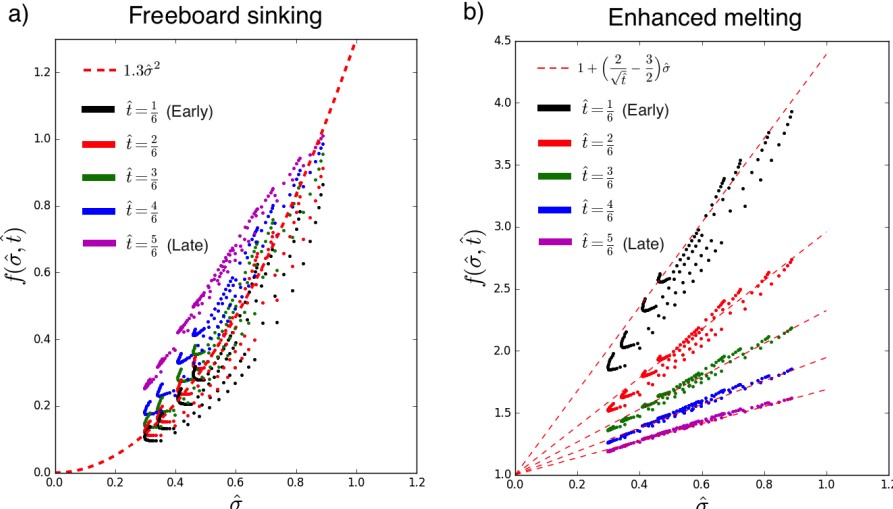

**Figure 11.** Determining the effective strengths, $S^* \equiv f(\hat{\sigma}, \hat{t})S$. Points represent estimates of the correction $f(\hat{\sigma}, \hat{t})$ for each of the curves in Fig. 7 evaluated at different times $\hat{t} \equiv \frac{St}{1-x_i}$. The function $f(\hat{\sigma}, \hat{t})$ is evaluated as $f(\hat{\sigma}, \hat{t}) \equiv 2(<x(t)> - x_i)/(St)$. Different colors correspond to different times with black corresponding to early in the season and magenta to late in the season. Non-dimensional roughness, $\hat{\sigma}$, is shown on the x-axis. a) $f_{\text{fs}}(\hat{\sigma}, \hat{t})$ evaluated for the freeboard sinking curves in Fig. 7a. There is no obvious dependence on $\hat{t}$. Freeboard sinking becomes completely suppressed as roughness tends to zero. The dashed red line represents the fit to these estimates of the form $f_{\text{fs}}(\hat{\sigma}, \hat{t}) = a\hat{\sigma}^2$. b) $f_{\text{em}}(\hat{\sigma}, \hat{t})$ evaluated for the enhanced melting curves in Fig. 7b. There is a clear dependence on $\hat{t}$. Enhanced melting proceeds even as roughness tends to zero. Red dashed lines are fits to these data of the form $f_{\text{em}}(\hat{\sigma}, \hat{t}) = 1 + c(\hat{t})\hat{\sigma}$, where $c(\hat{t}) \equiv \frac{2}{\sqrt{\hat{t}}} - \frac{3}{2}$.