# Peer review of "A simple model for the evolution of melt pond coverage on permeable Arctic sea ice"

_The Cryosphere, 2016_

## Referee Comment (RC1) · Anonymous Referee #1 · 12 Apr 2016

Review of Popovic and Abbot 2016 - for The Cryosphere.

Considerable attention is currently being devoted to understanding the evolution of melt ponds on sea ice surface. The ponds are highly important to summer energy balance of the sea ice cover. Popovic and Abbot provide an intriguing discussion of how freeboard loss due to ice melt at different interfaces can influence the growth or decline of melt pond areal coverage over the duration of the melt season. Within the paper the authors discuss the impact of melt at different locations on the buoyancy of the ice and the ramifications of these impacts on pond coverage, assuming ponds have already drained to sea level. The authors create a mathematical representation of the processes discussed and use this to explore several possible scenarios for melt, demonstrating the possibility that this mechanism could leaded to a dichotomy of melt ponding behavior, with ponds above some critical coverage growing, and below some coverage either shrinking or, more likely, stagnating in areal coverage.

The reviewer notes and appreciates that the paper is well written; both of high prose quality and clearly organized to convey the concepts postulated and mechanisms discovered. The reviewer finds, however, that there are several very serious issues with the evidence presented and its ability to explain the dichotomy of pond coverage evolution. The reviewer feels there is more than ample evidence in the literature to disprove this proposed mechanism and notes that the authors were aware of this evidence, at several points discuss this evidence, and then follow the discussion by ignoring the contradicting evidence. As such, this reviewer feels the authors have presented a mechanism that does not explain the pond dichotomy because of a mismatch in timing and because the mechanism relies on several assumptions very likely to be false. The reviewer highlights these several major scientific flaws here and then discusses line by line details below. The authors do, however, have the framework for evaluating and discussing the impacts of freeboard on pond coverage. The reviewer advises the authors re-write the paper entirely to focus on what these relationships appear to be, without trying to coax them into showing a particular result, such as explanation of dichotomous behavior.

Major scientific issues:
1. **The necessary assumption required for hydrostatics to produce a dichotomous behavior is that ice immediately surrounding ponds melts more slowly than other ice. This is extremely unlikely to be true.** Several potential reasons for slower melt at pond perimeter are presented by the authors. Some of these are logical (e.g. a.) while others (e.g. c.) ignore that the mechanism is just as likely to cause enhanced melt as slowed melt. (see detailed comments on lines 318-355) The mechanisms discussed are very likely overridden by much stronger mechanisms for accelerated melt at pond perimeters. In particular, lower albedo is observed in low-freeboard situations where a full surface scattering layer cannot emerge above freeboard (common near freeboard level at pond perimeters) and lateral heat transfer from pond water lapping against the wall of the pond has been shown to encourage lateral melt at the pond boundaries (see SHEBA pond related publications by Freitag, Eicken, and Perovich). Extensive lidar studies by Polashenski et al. and Landy et al. also indicate relatively little melt rate preference, and decidedly do not support a hypothesis of slower melt rates on the perimeter of ponds (as the author notes and then dismisses without cause in lines 395-397). These studies show freeboard change overall does dominating later season pond growth.
2. **Topography of the ice is neglected throughout.** Topography of the ice is intimately connected to bare ice area by hydrostatics. Given two floes of the same mean thickness, both ponded at sea level, the one with the lower bare ice area must necessarily have higher mean bare ice height to maintain the same volume of ice above sea level. This means the bare ice in the high pond coverage case is more resistant to flooding or melt. This is a powerful stabilizing feature that resists pond growth when pond area is high. The neglect of this inherent feedback leads to the dichotomy of behavior.

3. **The dichotomy in system state between high pond coverage and almost no pond coverage presented by Perovich et al., and further discussed by Webster et al., 2015 does not match the proposed mechanism in timing.** Some observations indicate the dichotomy occurs very early, in the melt season, (i.e. ponds never form in the first place) while others suggest the dichotomy is initiated at the time ponds drain to sea level. The mechanisms proposed here operate after the drainage of the ponds to sea level and take considerable time to produce effects. By timing, therefore, it appears the mechanisms proposed here are not well situated to create the observed ponding dichotomy.

4. **The reviewer is not aware of any observations that indicate pond area decreasing after ponds have drained to sea level.** The available observations appear to indicate that any area ponded and at sea level will remain ponded for the rest of the year (e.g. Polashenski et al., 2012, Landy et al., 2014). All observations of pond area reduction that this reviewer is aware of are due to drainage of above sea level ponds. This mechanism, as described by the 1d model, creates declines in pond coverage after ponds have drained to sea level – a behavior that does not appear to be documented anywhere in the literature. The mechanism as described by the 2d model does not create declines in pond coverage, but if this is the case, it does not explain the creation of low pond coverage ice, just the its maintenance.

The reviewer also finds that a few other processes are important enough in controlling hydrostatics that they probably cannot be neglected and should at least be discussed even if they cannot be incorporated into the model.

a. Density change both above and below sea level caused by internal melt – a very common mode of melt accounting for more than 1/3 of total ice melt in most cases.
b. Sub-ice freshwater lenses which commonly form in the Arctic reducing ice buoyancy.
c. Timing of surface melt and bottom melt is not coincident. Surface melt is often occurring for a considerable time prior to initiation of bottom melt.

The reviewer feels that correcting these assumptions will show that the ability of this mechanism to produce dichotomous behavior is removed. The paper will have to be re-written to reflect this reality. It appears to this reviewer that the dichotomy is established prior to ponds draining to sea level anyhow, so perhaps the authors are trying to make this mechanism explain something it should not explain. All is not lost though! An honest evaluation of what freeboard interactions should do to pond coverage would likely still be useful to the modeling community. It may be that freeboard control of pond coverage can explain some pond evolution after ponds have drained to freeboard even if it cannot explain the dichotomy.

**Line by Line comments:**
Line 28: Landy et al., 2015 did not provide any evidence to demonstrate that pond coverage should increase in a warming climate. If anything the colder year had more pond coverage. This and several other instances throughout the paper seem to indicate a style of citation this reviewer does not support. Specifically, the authors are citing an offhand statement within the prior work's introduction, which is not specifically supported by the prior work's data or analysis. Proper citation should only reference the content of a work where that work has actually demonstrated evidence that supports the assertion being referenced. If the prior work has only conjectured that a trend may exist, it would be proper to discuss the expected future state of melt ponds as such.

Line 30: "This should cause… ecosystem." This is only true where light, rather than nutrients, limit productivity. Nutrient limitation is the norm over most of the Arctic basin, so at very least a caveat clause should be inserted to this effect.

Line 47: The reviewer notes that Perovich et al date the photo JULY 15 – not June 15. This is an important correction. If June is actually correct, the mechanisms being discussed would not have had time

(weeks to months as discussed later) to operate at this stage in the melt season. The dichotomy in system state would have occurred very early, and potentially prior to freeboard impacts. Though the reviewer believes this is a simply typo, the timing of when the dichotomy first emerges is important for evaluating whether this mechanism is an appropriate explanation for the behavior. Webster et al., 2015 suggest the dichotomy occurs much sooner, perhaps pointing to other causes.

Line 59 – consider changing 'relevance' to 'existence'.

Line 64, 67,  and throughout: consider changing 'pondedness' to 'ponding' which would be more consistent with terminology in prior works.

Line 117 – The reviewer finds this first condition implausible and unsupported by existing evidence. See lidar studies by Polashenski et al, and or Landy et al. Data appears to be available from both authors if the figures are inadequate for fully resolving whether this is likely.

Line 128 -135 – the authors consider only melt at the interfaces. This ignores that sea ice experiences considerable internal melt. Internal melt causes substantial changes in ice density both above and below freeboard. This is likely a very important mechanism to add to consideration. Maybe conditions which favor internal melt above sea level (e.g. dirty ice) dis-favor ponding?

Line 128 – consider adding a caveat that all this discussion is only true if the ponds are hydraulically connected to the ocean. The current statement makes it seem like this is ALWAYS the case.

Line 132-133 –"The larger the pond fraction is, the larger their tendency to grow will be and the smaller ther tendency to shrink will be." This simplistic description of the implications of pond area on the ease of additional flooding (due to hydrostatics) ignores the necessary other implication of large pond area. Assume two floes have the same thickness, but one has greater pond area than the other. The bare ice on the floe with greater pond area must be topographically steeper and/or higher as required to keep the floe in hydrostatic equilibrium. Thus it is more resistant to encroachment, providing a very strong stabilizing effect not included in the model. The one with greater pond area is here asserted to have much easier potential for flooding additional area, but considering the topography of the remaining bare ice, it must necessarily become extremely resistant to pond growth at low bare ice area.

Line 195-6: It would be helpful if the authors consider the practical constraints on this number – while it is mathematically true that it could be greater than or less than 1, it is likely that thought about the system could rule out some values.

Line 194-203 – the assumption of constant slope used to turn the model into a 'one dimensional' approximation is invalid for a full range of pond coverage. In order to satisfy hydrostatics, the remaining bare ice in high pond coverage scenarios must necessarily have higher height and likely greater slope at the ice-pond interfaces. This invalid assumption results in the dichotomy of behavior.

Line 243 –This is not an entirely accurate statement, and an odd citation. The major thesis of these and other recent papers by Perovich et al is that bottom melt can be much greater than surface melt under certain circumstances. These Perovich papers also include considerable information about the time offset between surface and bottom melt which may be quite useful in thinking about how top vs. bottom melt impacts the hydrostatics at different times in the year.

Line 258 – these fluxes are somewhat outdated and based on heavy multiyear ice. Consider updating with albedo information more recently published, such as Perovich and Polashenski 2012 or papers by Nicolaus.

Line 265 – the slope of the pond boundaries is shown and briefly discussed by Landy et al, 2014 and Polashenski et al. 2012. These slope values appear to be steeper than most of the FYI boundaries they observe.

Line 270-272 – "a potential…" Discarding contrary evidence in this way is not supportable. A disagreement between the model and established evidence is not necessarily due to just the crudeness of the model – it could be evidence that the model is wrong and that the mechanisms are not the predominate ones! A more appropriate statement would be "Flooding due to freeboard loss is observed by Landy and Polashenski in a manner comparable to what we model. The timescale of the flooding observed, however, is much shorter than our model suggests (days vs. weeks to months). This discrepancy could be due to the crudeness of our model or may indicate that other mechanisms not considered are also important.

Line 280 – A 'considerable' number may be too strong a descriptor. The evidence for the non-ponding floes in the literature is sporadic.

Line 270-300 – this discussion appears to ignore that 'initial' pond coverage as discussed in this paper is actually the pond coverage at the time that ponds drain to sea level (a later stage of pond development) This discussion would be strengthened by considering the impacts of large or small pond coverage during the time period when ponds are above sea level as well. The ponding during the above-sea-level stage exerts considerable influence over later pond coverage by creating the below sea level depressions which will remain ponded later in the melt season (see Polashenski and Landy ). It seems likely that low coverage during this time would create few below-sea-level depressions, moderate coverage would create many, and very very high coverage would again create few because the ice surface would melt evenly, creating no pond basins once drainage occurred. Considering the impacts of this early season ponding behavior might suggest a stabilizing behavior on ponding? Maybe this would explain why most floes actually show ponding in a relatively narrow range.

Line 306 – Albedo set as a function of pond depth. This is OK, but the authors should be aware that this is a correlative description, not based on the mechanism controlling the radiative transfer. In other words, there is a modest correlation between pond depth and albedo, but physically speaking there is no relationship. Albedo is controlled by the properties of the ice underlying the pond, and it just so happens that thinner, clearer ice has a tendency to be under deeper ponds. Also, bare ice albedo does show considerable variation when less than about 0.5 m thick, or when it has limited freeboard (i.e. near a pond edge).

Line 306-307 – "A narrow lip around the pond edge had a slightly higher albedo than bare ice." The reviewer is not aware of any supporting evidence for this, and is aware of many observations that suggest albedo a low freeboard pond-edge sites is lower than elsewhere on the bare ice. The authors must acknowledge that this assumption runs contrary to observation or present evidence that it does not. In acknowledging that this assumption is implausible, the authors will find the ability of this mechanism to produce dichotomous behavior is removed. The paper will have to be re-written to reflect this reality. It may be that freeboard control of pond coverage can explain some pond evolution after ponds have drained to freeboard even if it cannot explain the dichotomy. It appears to this reviewer that the dichotomy is established prior to ponds draining to sea level anyhow, so perhaps the authors are trying to make this mechanism explain something it should not explain. An honest evaluation of what freeboard interactions should do to pond coverage would likely still be useful to the modeling community.

312-317 – the change in behavior for the 2d model discussed here is very important. Once ponds reach sea level, the reviewer is not aware of any observations that show ponds shrinking in area. As stated, "the points on the bottom of the ponds melt faster than bare ice, and therefore cannot move upwards" This

should mostly preclude pond shrinkage after ponds reach sea level. If this is the case, the dichotomy in pond coverage is produced by some other mechanism operating prior to the time ponds reach sea level. The mechanism operating after ponds reach sea level does not have the opportunity to reduce a moderate pond coverage to a low or zero pond coverage, at best it can maintain the status quo.

Lines 318 – 355 The reviewer appreciates these efforts to support the assumption that pond edges melt more slowly than other bare ice but finds them unconvincing. A. This mechanism may be valid. B. In practice this reviewer has observed the opposite, with bands of dark algae being washed up on the perimeter of many advanced ponds. Concentrations of light absorbing aerosols (ie. Black carbon) tend to be too small to make much difference. Sediment laden ice may behave as discussed here. C. The 'shadowing' effect of a slope would also be a 'higher incident radiation' effect on a slope with the opposite aspect (i.e. facing the sun). Such a slope would melt more rapidly. With random slope aspects, the effects of this are a wash.

The authors also fail to consider low albedo at pond edges due to limited surface scattering layer above freeboard, lateral heat transfer from pond water, mechanical action of pond waves, etc. The reviewer feels this one-sided examination results in an erroneous conclusion that slower than normal melt rates at the pond edge are plausible, even likely. The reviewer feels they are plausible only under very narrow circumstances and very unlikely to be the norm.

Line 355 – considering only hydrostatics, the slope/topography must be a function of pond area.

Line 360 – 363 – " where the effect of increased light absorption by the water is minimal" The authors provide a perfectly valid reference which establishes that this conjecture is false. It is not true and must therefore be removed.

Line 365 – "However, we believe that even stopping pond growth might be enough to explain the observations". This statement strongly suggests that the mechanism discussed here does not cause the dichotomy in pond coverage. At best it maintains the dichotomy introduced by some mechanism operating earlier in the melt season, prior to pond drainage to sea level.

Line 371 – more recent publications focus on the importance of light transmitted through ponds in areas of high pond coverage. Frey et al., 2011; Perovich and Polashenski 2012.

Line 395-397 – The authors repeatedly ignore evidence that contradicts their theory to the considerable frustration and exasperation of this reviewer. This is poor scientific practice. Polashenski et al., 2012 finds similar behavior to that of Landy et al., meaning that both available case studies do not support a critical aspect of the author's mechanism. Perhaps all this contrary evidence means the mechanism needs to be reconsidered?!

Lines 398 – 404 : This is entirely speculative and insufficiently addressed. Recommend removing entirely.

---

## Referee Comment (RC2) · Anonymous Referee #2 · 10 May 2016

This paper presents a simplified, single column model of melt pond coverage for permeable ice where the ice is in buoyant equilibrium. The objective of the model is to characterize how pond fraction changes in response to assumed melting rates of the ice in contact with the atmosphere and ice in contact with the ocean. The primary equation for the model examines the vertical rate of change of a point on the surface of the ice. Vertical change is partitioned into motion representing the rigid body of the ice plus local melting along the perimeter of ponds. Along with equations for the local melting, the authors present a combined linear equation for bare ice fractional coverage that they use to diagnose pond fraction behavior.

I liked the idea of the simple model and the authors do a nice job introducing the concepts that they are trying to simulate. Things get more confusing when the equations are presented in section 3. I had a hard time figuring out what equations 2 and 3 were

really representing and I think this section needs to be thoroughly rewritten to better explain the basis of the model. In particular, the relationship between bottom and top melting seems quite confusing when dividing between volumes of ice above or below sea level. For example, the authors state that removing a volume of ice below sea level reduces the volume above sea level by about 0.1. They then say that reducing a set volume from above sea level reduces the volume below sea level by about 0.9. These are saying the same thing in terms of total volume, and both will have the same effect on the freeboard. What I think is buried in describing the ice behavior is the effect that surface melting has relative to pond melting or relative to the old location of sea level; this is hard to extract from their description.

Particularly confusing is the statement that melting above sea level has a much greater effect on rigid body displacement in comparison with bottom melting. I understand that they are referring to a given point in the ice, but it sounds like surface melting is more "effective" in removing ice volume. So for example, when surface ice melts, the location of sea level relative to a fixed location in the ice changes much more than when the bottom melts. This is what I think the authors are trying to describe, but it is hard to follow in their presentation.

After presenting the basic equations relating ice surface motion to heat fluxes, the paper then jumps to a linear solution relating these rates to pond fraction. I think this derivation needs to be presented in the paper more explicitly; it is not obvious how they get from equation (4) to 9-10, and this is the core of their solution. Equation 11 is also not fully explained; where does the upper limit for x_o come from?

Because of the difficulty I had in following the model development, I am reluctant to accept this paper without a major revision specifically aimed at more description of the model and the basic concepts. Some of the model outcomes, for example, the need for less melting around pond edges, may have to do with simplifications that are not presented in the manuscript.

Minor Comments

Line 50-55. It is also quite possible that the overall pond distribution is tied to sea ice topography and snow depth distributions. So flat ice never develops ponds because there are never deep enough depressions that extend below sea level. There would be no need for a dichotomy if the pond distribution is determined mostly by ice and snow variability at the start of the melt season.

Line 110-125. It is also possible for ponds to grow through increased pond edge melting from "warm" pond water (from solar heating) that is in contact with the pond edge.

Line 130. Isn't this just the definition of ice-albedo feedback with ponds?

Line 135. Wouldn't smaller pond fractions require a smoother ice surface, e.g. perfectly flat ice will not have any ponds, whereas ice with high ice sails could potentially be much more ponded. Ice surface topography is not a model parameter, but it seems like it would be very important. Can you comment on this?

Line 320. It seems unlikely that all ponds would have less melting along the pond edge. What assumptions in your model formulation are forcing this condition?

Conclusions. Observations are one way to test your model, especially looking at the albedo near pond edges. I did not understand the comment on line 396-97 about observations from pond-free ice; did you mean in addition to the observations from ponded ice?

Another way to test the model might be by applying an explicit meltpond/sea ice model as proposed by Scott and Feltham. This would allow altering of the pond edge melting rates to see if this does indeed trigger dichotomy.

---

## Author Comment (AC1) · 9 Sep 2016

**Response to reviewers**

Predrag Popovic[1] and Dorian Abbot[1]

[1]University of Chicago, Dept. of Geophysical sciences

*Correspondence to:* Predrag Popovic (ppopovic@uchicago.edu)

Dear reviewers and editor,

We found the reviewer's comments extremely useful and have added text to address each of them.

As a result of the reviewer's comments we have thoroughly rewritten the paper, changing the focus from trying to explain a dichotomy in melt pond coverage to developing a simple model for the evolution of melt pond coverage on permeable sea ice that produces good quantitative agreement with the observations. We kept the main ideas and assumptions from our previous model, but also made major changes. Major changes include adding topography to our model, and representing melting near the pond perimeter more accurately. In our simple model, the relationships between physical parameters are analytically tractable, which allows us to address the behavior of melt ponds under general environmental conditions. We retain the notion of a fixed point, but only as a possibility and a discussion point, and leave out the dichotomy altogether.

In response to comments made by both reviewers, we have:

1. Included the topography in the form of the hypsographic curve.

2. Added a discussion on lateral melt and mechanisms that increase the rate of melt near the pond perimeter.

In addition, in response to reviewer 1, we have added a discussion on internal melt and changed focus to cases with increased melting near the pond perimeter, $k > 1$. In response to reviewer 2, we have described the model development more carefully.

Below we respond in detail to each of the reviewer's comments, and attach the revised manuscript. Thank you for your time and help improving this manuscript.

**Reviewer 1**

Thank you for your review. We found your comments extremely useful, and have added text to the paper to address each of them.

We have thoroughly rewritten the paper, and changed the focus from trying to explain a dichotomy

30 in melt pond coverage to developing a simple model for the evolution of melt pond coverage on permeable sea ice that produces good quantitative agreement with the observations. We kept the main ideas and assumptions from our previous model, but also made major changes. Major changes include adding topography to our model, and representing melting near the pond perimeter more accurately. We use our model to extract analytical relationships between physical parameters, and

35 explore pond behavior under general circumstances. We retain the notion of a fixed point, but only as a possibility and a discussion point, and leave out the dichotomy altogether.

In response to your comments, we have:

1. Changed the focus of the paper from trying to explain the dichotomy in melt pond coverage

40 to creating a simple model for pond coverage evolution. We now mainly consider cases with $k > 1$.

2. Included the topography in the form of the hypsographic curve.

3. Added a discussion on internal and lateral melt.

4. Discussed the mechanisms that increase the rate of melt near the pond perimeter.

45 We respond (in black) in detail to all of your comments (in blue) below.

1) The reviewer feels there is more than ample evidence in the literature to disprove this proposed mechanism and notes that the authors were aware of this evidence, at several points discuss this evi-

50 dence, and then follow the discussion by ignoring the contradicting evidence. As such, this reviewer feels the authors have presented a mechanism that does not explain the pond dichotomy because of a mismatch in timing and because the mechanism relies on several assumptions very likely to be false. The reviewer highlights these several major scientific flaws here and then discusses line by line details below. The authors do, however, have the framework for evaluating and discussing the impacts

55 of freeboard on pond coverage. The reviewer advises the authors re-write the paper entirely to focus on what these relationships appear to be, without trying to coax them into showing a particular result, such as explanation of dichotomous behavior.

**Response:** Although the nature of the presented mechanism was more a proof of concept, rather than a claim about explaining the dichotomy, we agree with reviewer's concerns about the evidence,

60 and accept their suggestion to change the focus of the paper. The paper has been thoroughly rewritten to reflect this. Our model is now presented to work as a simple model for evolution of pond coverage after pond drainage. We focus on determining the relationships between physical parameters, exploring the types of qualitative behavior of pond coverage, and on obtaining quantitative agreement

with observations. A fixed point is still mentioned, but as a discussion point, rather than the main

65   point of the paper. We omitted all mentions of dichotomy in the paper.

2) Topography of the ice is neglected throughout. Topography of the ice is intimately connected to bare ice area by hydrostatics. Given two floes of the same mean thickness, both ponded at sea level, the one with the lower bare ice area must necessarily have higher mean bare ice height to maintain the same volume of ice above sea level. This means the bare ice in the high pond coverage case

70   is more resistant to flooding or melt. This is a powerful stabilizing feature that resists pond growth when pond area is high. The neglect of this inherent feedback leads to the dichotomy of behavior.

**Response:** This was a very useful comment. To address this, we included the ice topography via the hypsographic curve, chosen in such a way that, for a given ice thickness $H$, and pond coverage $x$, the hypsographic curve above sea level hydrostatically balances the ice floe. If ponds are at sea

75   level, pond coverage can be found as an intersection between sea level and the curve. We relate the displacements of ice topography near the pond perimeters, $ds$, and a corresponding change in pond coverage, $dx$, using the hypsographic curve. We explain how we do this between lines 258 and 318. On line 230, we introduce the relationship between the average freeboard height, $h$, and the ice thickness and pond coverage

80   $$h = \frac{\rho_w - \rho_i}{\rho_w} \frac{H}{1-x} \ . \tag{1}$$

This means that for higher initial pond coverage, initial freeboard height will be higher, and ponds on ice floes with high pond coverage grow more slowly.

3) The reviewer also finds that a few other processes are important enough in controlling hydrostatics that they probably cannot be neglected and should at least be discussed even if they cannot be

85   incorporated into the model.

a) Density change both above and below sea level caused by internal melt - a very common mode of melt accounting for more than 1/3 of total ice melt in most cases.

b) Sub-ice freshwater lenses which commonly form in the Arctic reducing ice buoyancy.

**Response:** We agree that internal melt can be a significant process. As a result of this comment,

90   we added a discussion about internal melt (subsection 5.3). We did not include internal melt in our model but we discussed its effects on pond evolution. We concluded that if there is more internal melt above sea level than below sea level, a fixed point is more likely to occur.

We also added a sentence in the section about numerical solutions to our model on line 372: "*Although water and ice densities can vary due to changes in water salinity, internal ice melt, presence*

95   *of air bubbles and brine in the ice, or snowfall we will take these values to be fixed throughout the paper at $\rho_w = 1000 \frac{kg}{m^3}$, and $\rho_i = 900 \frac{kg}{m^3}$.*"

4) Landy et al., 2015 did not provide any evidence to demonstrate that pond coverage should increase in a warming climate. If anything the colder year had more pond coverage. This and several

other instances throughout the paper seem to indicate a style of citation this reviewer does not support. Specifically, the authors are citing an offhand statement within the prior work's introduction, which is not specifically supported by the prior work's data or analysis. Proper citation should only reference the content of a work where that work has actually demonstrated evidence that supports the assertion being referenced. If the prior work has only conjectured that a trend may exist, it would be proper to discuss the expected future state of melt ponds as such.

**Response:** We reconsidered all the citations, and removed or changed those that do not reference a main point in the cited paper. In doing this, we removed the reference to Landy et al., 2015 entirely.

5)"This should cause... ecosystem." This is only true where light, rather than nutrients, limit productivity. Nutrient limitation is the norm over most of the Arctic basin, so at very least a caveat clause should be inserted to this effect.

**Response:** We removed this line from the paper.

6) The reviewer notes that Perovich et al date the photo JULY 15 - not June 15. This is an important correction. If June is actually correct, the mechanisms being discussed would not have had time (weeks to months as discussed later) to operate at this stage in the melt season. The dichotomy in system state would have occurred very early, and potentially prior to freeboard impacts. Though the reviewer believes this is a simply typo, the timing of when the dichotomy first emerges is important for evaluating whether this mechanism is an appropriate explanation for the behavior. Webster et al., 2015 suggest the dichotomy occurs much sooner, perhaps pointing to other causes.

**Response:** Since there is no more mention of the dichotomy, we removed the reference to this photo.

7) consider changing 'pondedness' to 'ponding' which would be more consistent with terminology in prior works.

**Response:** In the revised paper, we do not use either of these two terms.

8) the authors consider only melt at the interfaces. This ignores that sea ice experiences considerable internal melt. Internal melt causes substantial changes in ice density both above and below freeboard. This is likely a very important mechanism to add to consideration. Maybe conditions which favor internal melt above sea level (e.g. dirty ice) dis-favor ponding?

**Response:** Thanks to this comment, we have added an entire section in the discussion about the effects of internal melt on pond evolution. We concluded that if there is more internal melt above sea level than below sea level, it could slow down, or even prevent pond growth, at least for a period of time.

9) consider adding a caveat that all this discussion is only true if the ponds are hydraulically connected to the ocean. The current statement makes it seem like this is ALWAYS the case.

**Response:** As a result of this comment, we added several sentences where we state that we assume the ponds are hydraulically connected to the ocean:

1. On line 50: "*We will assume that ice is permeable, ponds are at sea level and hydraulically connected to the ocean, the whole ice floe is in hydrostatic balance, and different points on the ice surface may melt at different rates.*"

2. On line 85: "*We will assume that from this point on, the ponds are hydraulically connected with the ocean, and the only way for pond coverage to increase is for the points on the ice surface which were above sea level to sink below sea level.*"

10) "The larger the pond fraction is, the larger their tendency to grow will be and the smaller their tendency to shrink will be." This simplistic description of the implications of pond area on the ease of additional flooding (due to hydrostatics) ignores the necessary other implication of large pond area. Assume two floes have the same thickness, but one has greater pond area than the other. The bare ice on the floe with greater pond area must be topographically steeper and/or higher as required to keep the floe in hydrostatic equilibrium. Thus it is more resistant to encroachment, providing a very strong stabilizing effect not included in the model. The one with greater pond area is here asserted to have much easier potential for flooding additional area, but considering the topography of the remaining bare ice, it must necessarily become extremely resistant to pond growth at low bare ice area.

**Response:** We removed this sentence from the paper. The stabilizing mechanism was addressed by including topography in the form of a hypsographic curve, and by requiring that the topography be in the state of hydrostatic balance. Using this approach, we extract the dependence of pond growth rate on initial pond fraction.

11) It would be helpful if the authors consider the practical constraints on this number - while it is mathematically true that it could be greater than or less than 1, it is likely that thought about the system could rule out some values.

**Response:** We placed rough constraints on $k$ by assuming that ice near the pond edge probably melts at a rate faster than bare ice, but slower than ponded ice. In this way, we found that $k$ probably lays between $1$, and $1.7$.

12) the assumption of constant slope used to turn the model into a 'one dimensional' approximation is invalid for a full range of pond coverage. In order to satisfy hydrostatics, the remaining bare ice in high pond coverage scenarios must necessarily have higher height and likely greater slope at the ice-pond interfaces. This invalid assumption results in the dichotomy of behavior.

**Response:** The assumption of constant slope was made unnecessary by introducing the hypsographic curve to relate displacements with changes in pond coverage.

13) This is not an entirely accurate statement, and an odd citation. The major thesis of these and other recent papers by Perovich et al is that bottom melt can be much greater than surface melt under certain circumstances. These Perovich papers also include considerable information about the time offset between surface and bottom melt which may be quite useful in thinking about how top vs. bottom melt impacts the hydrostatics at different times in the year.

**Response:** Instead of rough estimates, we now use the full time-resolved fluxes measured during SHEBA. For the shortwave, longwave, sensible, and latent heat fluxes, we use those derived by Skyllingstadt et al 2009 using SHEBA measurements. For bottom flux, we use the time resolved flux from obtained from measurements of bottom ablation rates during SHEBA, and described in Perovich et al 2003. We found that our model results using time-resolved fluxes do not differ significantly from the results using the time-averaged fluxes. When comparing our model result with SHEBA measurements of pond coverage (reported in Perovich et al 2003), we use the full time-resolved bare ice, ponded ice, and bottom ablation rates reported in Perovich et al 2003.

14) these fluxes are somewhat outdated and based on heavy multiyear ice. Consider updating with albedo information more recently published, such as Perovich and Polashenski 2012 or papers by Nicolaus.

**Response:** We compared our results to model results of Skyllingstadt et al 2009, and with measurements made along the albedo line during the SHEBA mission, as described in Perovich et al 2003. For this reason, we used the same fluxes reported in these two studies.

15) the slope of the pond boundaries is shown and briefly discussed by Landy et al, 2014 and Polashenski et al. 2012. These slope values appear to be steeper than most of the FYI boundaries they observe.

**Response:** Because we now use the hypsographic curve to relate displacements with changes in pond coverage, we removed all mentions of slope of the pond boundaries.

16) "a potential..." Discarding contrary evidence in this way is not supportable. A disagreement between the model and established evidence is not necessarily due to just the crudeness of the model - it could be evidence that the model is wrong and that the mechanisms are not the predominate ones! A more appropriate statement would be 'Flooding due to freeboard loss is observed by Landy and Polashenski in a manner comparable to what we model. The timescale of the flooding observed, however, is much shorter than our model suggests (days vs. weeks to months). This discrepancy could be due to the crudeness of our model or may indicate that other mechanisms not considered are also important.'

**Response:** Our model is now changed so that it provides quantitative agreement with the measurements.

17) Albedo set as a function of pond depth. This is OK, but the authors should be aware that this is a correlative description, not based on the mechanism controlling the radiative transfer. In other words, there is a modest correlation between pond depth and albedo, but physically speaking there is no relationship. Albedo is controlled by the properties of the ice underlying the pond, and it just so happens that thinner, clearer ice has a tendency to be under deeper ponds. Also, bare ice albedo does show considerable variation when less than about 0.5 m thick, or when it has limited freeboard (i.e. near a pond edge).

**Response:** In the revised paper, we generalized our model to a 1D model, instead of a 2D model (lines 319 - 341). In the 1D model, we mostly use a constant pond albedo. We also tested the effects of pond albedo varying with depth by implementing a pond albedo parameterization used in Skyllingstadt et al 2009, and concluded that varying pond albedo with depth has little effect on pond coverage evolution. This is probably due to the fact that the pond albedo itself is relatively less important for pond evolution in our model than other parameters, such as the bare sea ice albedo.

18) "A narrow lip around the pond edge had a slightly higher albedo than bare ice." The reviewer is not aware of any supporting evidence for this, and is aware of many observations that suggest albedo a low freeboard pond-edge sites is lower than elsewhere on the bare ice. The authors must acknowledge that this assumption runs contrary to observation or present evidence that it does not. In acknowledging that this assumption is implausible, the authors will find the ability of this mechanism to produce dichotomous behavior is removed. The paper will have to be re-written to reflect this reality. It may be that freeboard control of pond coverage can explain some pond evolution after ponds have drained to freeboard even if it cannot explain the dichotomy. It appears to this reviewer that the dichotomy is established prior to ponds draining to sea level anyhow, so perhaps the authors are trying to make this mechanism explain something it should not explain. An honest evaluation of what freeboard interactions should do to pond coverage would likely still be useful to the modeling community.

**Response:** Allowing a narrow lip around the pond edge to have a lower melting rate than bare ice is equivalent to allowing $k < 1$. We retained this only as a possibility, and do not include it in the 1D generalization of the simple model. For the most part we now estimate pond coverage evolution when a narrow lip around the pond edges melts faster, or at the same rate as bare ice on average, with $k$ varying between 1 and 1.67.

19) The reviewer appreciates these efforts to support the assumption that pond edges melt more slowly than other bare ice but finds them unconvincing. A. This mechanism may be valid. B. In practice this reviewer has observed the opposite, with bands of dark algae being washed up on the perimeter of many advanced ponds. Concentrations of light absorbing aerosols (i.e. Black carbon) tend to be too small to make much difference. Sediment laden ice may behave as discussed here. C. The 'shadowing' effect of a slope would also be a 'higher incident radiation' effect on a slope with the opposite aspect (i.e. facing the sun). Such a slope would melt more rapidly. With random slope aspects, the effects of this are a wash.

The authors also fail to consider low albedo at pond edges due to limited surface scattering layer above freeboard, lateral heat transfer from pond water, mechanical action of pond waves, etc. The reviewer feels this one-sided examination results in an erroneous conclusion that slower than normal melt rates at the pond edge are plausible, even likely. The reviewer feels they are plausible only under very narrow circumstances and very unlikely to be the norm.

**Response:** We removed the 'shadowing' effect of a slope. Instead, we added a mechanism to make the melt rate near the boundary lower than the rest of the bare ice, whereby existence of ridge sails can lead to pond boundary melting slower than bare ice on average. As ridge sails collect more sunlight, they melt faster than flat bare ice, thereby increasing the average bare ice melting rate. Flat ice near the pond boundaries would therefore melt slower than bare ice on average.

We added several mechanisms to make the melt rate near the boundary higher than the rest of the bare ice. For example, we wrote:

"*Even a very thin layer of water on top of ice can significantly decrease its albedo due to the effect of multiple light reflections (Makshtas and Podgorny, 1996). Since pond water could wet the surrounding ice, this mechanism could lead to ice near the pond boundaries melting faster than bare ice far away from the ponds.*"

and

"*Bands of dark algae are often washed up on the perimeter of advanced ponds, accelerating the melting.*"

We also devoted a page to discuss lateral melting by pond water.

20) more recent publications focus on the importance of light transmitted through ponds in areas of high pond coverage. Frey et al., 2011; Perovich and Polashenski 2012.

**Response:** This discussion on bottom melting has been removed entirely.

21) The authors repeatedly ignore evidence that contradicts their theory to the considerable frustration and exasperation of this reviewer. This is poor scientific practice. Polashenski et al., 2012 finds similar behavior to that of Landy et al., meaning that both available case studies do not support a critical aspect of the author's mechanism. Perhaps all this contrary evidence means the mechanism needs to be reconsidered?!

**Response:** The paper has been thoroughly rewritten, hopefully to considerable relief of the reviewer.

22) This is entirely speculative and insufficiently addressed. Recommend removing entirely.

**Response:** Done.

**Reviewer 2**

Thank you for your time, and your comments!

We have thoroughly rewritten the paper, and changed the focus from trying to explain a dichotomy in melt pond coverage to developing a simple model for the evolution of melt pond coverage on permeable sea ice that produces good quantitative agreement with the observations. We kept the main ideas and assumptions from our previous model, but also made major changes. Major changes include adding topography to our model, and representing melting near the pond perimeter more

accurately. We use our model to extract analytical relationships between physical parameters, and explore pond behavior under general circumstances. We retain the notion of a fixed point, but only as a possibility and a discussion point, and leave out the dichotomy altogether.

In response to your comments, we have:

1. Described the model development more carefully.

2. Included the topography in the form of the hypsographic curve.

3. Added a discussion about lateral melt.

4. Discussed the mechanisms that increase the rate of melt near the pond perimeter.

We respond (in black) in detail to all of your comments (in blue) below.

1) I liked the idea of the simple model and the authors do a nice job introducing the concepts that they are trying to simulate. Things get more confusing when the equations are presented in section 3. I had a hard time figuring out what equations 2 and 3 were really representing and I think this section needs to be thoroughly rewritten to better explain the basis of the model. In particular, the relationship between bottom and top melting seems quite confusing when dividing between volumes of ice above or below sea level. For example, the authors state that removing a volume of ice below sea level reduces the volume above sea level by about 0.1. They then say that reducing a set volume from above sea level reduces the volume below sea level by about 0.9. These are saying the same thing in terms of total volume, and both will have the same effect on the freeboard. What I think is buried in describing the ice behavior is the effect that surface melting has relative to pond melting or relative to the old location of sea level; this is hard to extract from their description.

**Response:** Thank you for this comment! We tried to make the derivation of the model equation more clear. We define $s(\mathbf{r})$ as the topography above sea level (that is elevation of ice surface above sea level), and split its change into a contribution from local and rigid body movement. We derived the rigid body movement in a slightly different way. We related the mass of ice above sea level to mass of ice below sea level through hydrostatics, and split the change in mass above and below sea level into contributions from melting and rigid body movement:

*We can write this hydrostatic balance as*

$$m_{\text{above s. l.}} = \frac{\rho_{\text{w}} - \rho_{\text{i}}}{\rho_{\text{w}}} m_{\text{below s. l.}} \, , \tag{2}$$

*where $m_{above\ s.\ l.}$ and $m_{below\ s.\ l.}$ represent the mass of ice above and below sea level, and $\rho_w$, and $\rho_i$ represent the densities of water and ice.*

*From Eq. (2), we write a relation for the change in mass above sea level relates to the change in mass below sea level. The mass above and below sea level can change either because the ice melts or because the floe moves as a rigid body, changing the proportion of ice above and below sea level. Therefore, differentiating Eq. (2) and splitting into melt and rigid body contributions, we find*

$$dm^{\text{melt}}_{\text{above s. l.}} + dm^{\text{rigid body}}_{\text{above s. l.}} = \frac{\rho_{\text{w}} - \rho_{\text{i}}}{\rho_{\text{w}}} \left[ dm^{\text{melt}}_{\text{below s. l.}} + dm^{\text{rigid body}}_{\text{below s. l.}} \right] , \tag{3}$$

*where $dm^{melt/rigid\ body}_{above/below\ s.\ l.}$ represent changes in mass above and below sea level due to either ice melting or the entire floe floating up or down.*

Then we related $dm^{\text{melt}}_{\text{above/below s. l.}}$ to fluxes used to melt bare ice, ponded ice, and ice bottom, and $dm^{\text{rigid body}}_{\text{above/below s. l.}}$ to the rigid body displacement, $ds_{\text{rigid body}}$. We also removed the discussion about melting above and below sea level moving the ice surface by different magnitudes, to avoid confusion.

2) Particularly confusing is the statement that melting above sea level has a much greater effect on rigid body displacement in comparison with bottom melting. I understand that they are referring to a given point in the ice, but it sounds like surface melting is more 'effective' in removing ice volume. So for example, when surface ice melts, the location of sea level relative to a fixed location in the ice changes much more than when the bottom melts. This is what I think the authors are trying to describe, but it is hard to follow in their presentation.

**Response:** We removed this discussion from the text.

3) After presenting the basic equations relating ice surface motion to heat fluxes, the paper then jumps to a linear solution relating these rates to pond fraction. I think this derivation needs to be presented in the paper more explicitly; it is not obvious how they get from equation (4) to 9-10, and this is the core of their solution. Equation 11 is also not fully explained; where does the upper limit for x_o come from?

**Response:** We added several more steps in deriving the final equation. We derive the displacement equation step by step, to obtain

$$\frac{ds}{dt}(\boldsymbol{r}) = -\left[ (k(\boldsymbol{r}) - 1) \frac{\overline{F}_{\text{bi}}}{l\rho_i} \right] - $$
$$- \left[ \frac{\rho_w - \rho_i}{\rho_w} \frac{1}{l\rho_i} \left( \overline{F}_{\text{bi}} + \frac{x}{1-x} \overline{F}_{\text{mp}} + \frac{1}{1-x} \overline{F}_{\text{bot}} \right) \right] . \tag{4}$$

From this point on, our model differs from the one described in the previous version of the paper. Instead of assuming a constant slope at the ice-pond interface, we relate the displacement near the pond edge to changes in pond coverage by using the hypsographic curve. If ponds are at sea level, pond coverage can be inferred from a hypsographic curve as an intersection between sea level and the curve. We first derive the relationship between $ds$ and $dx$ in the case of pure freeboard sinking, and then in the case when only vertical sidewall melting and no freeboard sinking is occurring. We

then add the two contributions together as if they were independent, and test this assumption on a
345  model that explicitly evolves the hypsographic curve in time to get pond coverage evolution.

4) Line 50-55. It is also quite possible that the overall pond distribution is tied to sea ice topography and snow depth distributions. So flat ice never develops ponds because there are never deep enough depressions that extend below sea level. There would be no need for a dichotomy if the pond distribution is determined mostly by ice and snow variability at the start of the melt season.

350  **Response:** This is quite plausible. For this reason, we do not try to explain the dichotomy anymore, but focus on building a simple model of pond evolution where the roles of physical parameters are tractable.

5) Line 110-125. It is also possible for ponds to grow through increased pond edge melting from "warm" pond water (from solar heating) that is in contact with the pond edge.

355  **Response:** In the paper, we stated that "*we focus on the vertical displacements of points on the surface of the ice in response to melt.*" (line 94), and we added a caveat that "*In reality, ponds can also grow through horizontal melting of their sidewalls.*" (line 86). Although we did not explicitly include lateral melting in our model, we added a discussion on it, and suggested how it might be included in the model. We derived the rate of change of pond coverage due to lateral melting, and
360  showed its effects on pond coverage distribution. We also compared our model to Skyllingstadt et al 2009 who consider only lateral melting (Fig. 5 a)).

6) Line 130. Isn't this just the definition of ice-albedo feedback with ponds?

**Response:** This is not the ice-albedo feedback in the typical sense. In the typical sense, ice-albedo feedback implies that due to more dark surfaces, more sunlight is absorbed, and therefore
365  more energy is available for melting additional ice. In this case the feedback is the following: since ponds are below sea level, and because melting below sea level induces a downward rigid body motion of the floe, melting ponded ice will increase the area of ice surface below sea level (i.e. pond area). In this sense melting ponded ice reinforces the pond growth. The feedback in this sense does not have to do with reflectivity of the ponds.

370  7) Line 135. Wouldn't smaller pond fractions require a smoother ice surface, e.g. perfectly flat ice will not have any ponds, whereas ice with high ice sails could potentially be much more ponded. Ice surface topography is not a model parameter, but it seems like it would be very important. Can you comment on this?

**Response:** It is true that ice surface topography is a very important factor in pond evolution,
375  and the other reviewer pointed this out several times (comments 2, 10, and 12). For this reason, we include the surface topography in the form of a hypsographic curve. We use the measurements of the hypsographic curve by Landy et al 2014, and modify it in order to initialize our model with an ice thickness and initial pond coverage of our choice.

8) Line 320. It seems unlikely that all ponds would have less melting along the pond edge. What
380  assumptions in your model formulation are forcing this condition?

**Response:** As was pointed out by the other reviewer as well, this is likely true (comment 19). This is one of the reasons why we decided to make the fixed point a minor point in our new paper. It seems more likely that ice near the pond edge melts faster than bare ice on average, and we mainly model this effect (which we call vertical sidewall melting). We also added a discussion about the physical mechanisms that could accelerate the melt near the pond boundary. For example, we stated

*" Wetting the ice surface can also decrease its albedo in another way. If the ice is rough and jagged on a small scale, it will have a high reflectance. Due to the effects of surface tension increasing the pressure, ice crystals with a small radius of curvature will be under higher pressure, and will therefore have a lower melting point than large flat crystals. If a collection of crystals of small but varying size is submerged in water that is at the crystals' mean melting temperature, small crystals will melt, and large crystals will grow at their expense through a process called Ostwald ripening (Raymond and Tusima, 1979). This process will smooth the ice, lowering its reflectiveness, and accelerating the rate of melt."*

9) Conclusions. Observations are one way to test your model, especially looking at the albedo near pond edges. I did not understand the comment on line 396-97 about observations from pond-free ice; did you mean in addition to the observations from ponded ice? Another way to test the model might be by applying an explicit meltpond/sea ice model as proposed by Scott and Feltham. This would allow altering of the pond edge melting rates to see if this does indeed trigger dichotomy.

**Response:** Although our model now does not attempt to explain the dichotomy, observations would still be useful to constrain $k$ which controls the vertical sidewall melting. On the line 344, we say

*"The vertical sidewall melting strength, $S_{sm}$, however, also depends on the difficult-to-measure parameters $k$ and $\delta$ that describe the melt rate near the pond boundary. If we could place some constraints on these parameters reliably enough, our model would be a useful parameterization in GCMs for pond growth after ice becomes permeable. Observations and detailed theoretical studies focusing on ice melt processes near the pond edge could constrain these parameters."*

Other parameters are fairly well constrained, but it would still be useful to test the relationships between physical parameters that we derived.

[revised manuscript text omitted]

---

## Author Comment (AC2) · 9 Sep 2016

Dear Referee #2,

please find attached our response and a revised manuscript.

Sincerely, Predrag Popovic

Please also note the supplement to this comment:
http://www.the-cryosphere-discuss.net/tc-2016-7/tc-2016-7-AC2-supplement.pdf

---

## Editor Comment (EC1) · D. L. Feltham (Editor) · 13 Feb 2017

Review posted by editor on behalf of anonymous reviewer

Much improved.

L 240- 250 the statements suggesting that hypsographic curves are general across sea ice in line 240-250 should be removed. It is likely that hypsographic curve of sea ice varies greatly due to many influences on topography. Related discussion could suggest that this model would be made stronger by input of accurate curves.

490 - vertical sidewall melting appears to be unimportant, the freeboard mechanism dominates. That is the conclusion here. This additional sentence trying to justify vertical sidewall melting is unsupported. By itself this sentence would be innocent, but

the first part of the discussion (5.1) which seems to be trying to focus on mechanisms of accelerated melt around pond edges really works this topic more. The extended discussion of vertical melt around pond edges doesn't really seem to have any place here given the behavior of the model. Suggest removing entirely. If not... see below.

Section 5.1 is problematic. It struck the reviewer as disingenuous to discuss all these mechanisms and then admit at the end that the available evidence suggests none are important, only as an aside. The model suggests changes in freeboard, and though the authors gloss over it, changes in hypsographic curve are the important features. The discussion should focus on these rather than relict bits from prior version. An analysis of sensitivity to ice topography/roughnes/hypsographic curve would be very highly valuable to the community here instead. There are also issues with the summary of the literature and applicability for several of the topics written as well.

line 545 - thin layer of water... due to multiple light reflections. I think you mean refractions and you should clarify that the refraction occurs more strongly (and hence more backscatter occurs) when the ice is not covered by liquid and air occupies pore/intercrystalline space.

paragraph around line 550. Yes, this generally describes grain ripening, but this process is not significant for large scale 'jagged' features and 'smoothing' ice. It is a microstructural process, and one unlikely to be much more prevalent near ponds than away from them, therefore not a good candidate as described.

c- 557 - needs a reference, or replace 'often' with something softer like 'sometimes' and a personal communication.

580 - the ridge accelerated melt is largely fictitious. For every sun-ward side, there is a shaded side.

-SHEBA ponds are not first year ice, nor were they all permeable during the times the authors use for comparison. This creates several instances in needed of adjustment or

revision.

-The conclusion of 1.3 percent per month in a warming climate is way to shaky to be allowed in the abstract or conclusion as a strong take home point. There are numerous processes not accounted for here. Trumpeting a pond coverage trend in a warming climate based only on an idealized model of one of several pond formation processes is just too bold to pass peer review. The strong conclusions of this paper are with regard to the relative roles of various processes implied. This point may be presented only in the discussion as a possible outcome given particular assumptions, not as a main take home of the paper.

---

## Editor Comment (EC2) · D. L. Feltham (Editor) · 13 Feb 2017

Full review from anonymous reviewer, posted by editor

This paper is much improved. The reviewer appreciates the authors' careful attention to the prior comments. The authors are tackling a very interesting topic and the paper should be a sound resource to the community as late season melt pond behavior begins to be addressed better in models. There are several additional areas where the reviewer feels the paper can be strengthened. If the authors address these, the paper should become suitable for publication.

Major comments 1. Reviewer feels that the authors use of Multiyear ice dataset for comparison of a model that only tracks FYI processes is a significant problem. Authors must either change out comparison data to a FYI dataset or place clear statements

throughout the text indicating the severe limitations of comparison to field observations that amount to a Frankenstein combination of FYI (e.g. Landy Hypsographic curve) and MYI (Perovich Albedos). 2. Reviewer feels the authors miss a substantial opportunity to comment on the impact that differing hypsographic curves would have on the ice. Reviewer suggests that vertical sidewall melting would be primarily controlled by height above sea level of the area. . . implying that 3. The conclusion of 1.3 percent per month in a warming climate is way to shaky to be allowed in the abstract or conclusion as a strong take home point. There are numerous processes not accounted for here. Trumpeting a pond coverage trend in a warming climate based only on an idealized model of one of several pond formation processes is just too bold to pass peer review. Authors have two options on this point. 1. Produce and include estimates of uncertainty with this number, 2. Drop the specific quantification and discuss the sign of the change. 4. Conclusions need to do a better job clarifying over what conditions the general interactions between melt location and pond coverage apply. Reviewer feels the model results only show how these are true for certain pond coverage and ice thickness pathways, dependent on the key initial assumptions, such as that ice all has the same hypsographic curve. As an example the conclusion that bottom melting's impact on growth rate increases with increasing pond coverage may not hold if comparing two floes with very different pond coverage and different topography. One floe may have very flat ice and low pond coverage at the time ponds drain to sea level, while another may have higher pond coverage and greater topography. Bottom melt will impact the level ice floe more than the rougher flow because of having more ice near sea level to flood, regardless of the initial pond coverage.

Specific comments: L1 Clarify that this is all about FY ice. . . As the melt season progresses FIRST YEAR sea ice (delete floes) in the Arctic often become(S) permeable...

L4 Clarify that it is the surface relation to sea level that is of concern . . .and correspond to regions of the sea ice WHOSE SURFACE IS (delete that are) below sea level. . .

L10 "relatively well" – relative to what? Can you replace this with a quantitative statement rather than this highly subjective one.

L13 Suggest replacing "demonstrates the somewhat surprising result" with simply "indicates". I'm not surprised by this! Also see comments later about some issues with the support for this statement.

L15. "Performing a similar... 1.3%... per month. This statement is too sketchy for the abstract. This quantitative conclusion is not something that a reader could use as a takeaway number from this paper. Reviewer could permit discussion of the <sign> of this impact here, but if the authors would like to quantify it specifically outside the context of the section where this comes from in the discussion, then error bars will need to be included. Reviewer expects these would be very large and challenging to quantify.

L40 "Typical... proceeds in (delete a) fairly consistent STAGES (delete manner)." The stages are predictable, and regular. Their duration is not.

L45 You can actually be clearer here. The stage when ice... is typically the longest, OFTEN LONGER THAN THE TWO FIRST STAGES COMBINED.

L72 "we analyze the model RESULTS analytically" (I think its more the results than the model you analyze)

L80 " ... and (delete all the meltwater created is) MELTWATER CAN BE." Need to be clear here that meltwater is not necessarily all removed, just that it can be removed.

L 82 "...when (delete: pond coverage drops to its summer minimum, meaning that all) the meltwater on the ice... " It is not necessarily true that the coverage is it its minimum at this point, though it is likely. Better to just leave this point defined by water level and sea level relationship as the remainder of the sentence does

L 86 " ...which were above sea level to sink OR MELT below sea level"... You allow for vertical melt to cause creation of below sea level surface.

L 89 " . . . at the surface of the ice". Need to clarify that this is at EITHER surface, top or bottom. Suggest ". . . at the surface or bottom of the ice" for clarity.

L90 " we will further assume. . . " You do not follow this. New ponds can form by flooding isolated areas by freeboard loss based on your use of the hypsographic curve. This clarifying constraint applies only to your vertical melt enhancement around pond perimeters. If you were to change the vertical melt enhancement to be based on surface height above freeboard, which is probably its strongest dependence, you could remove this weakness altogether and freely permit new ponds to form in low topography areas by both freeboard loss and accelerated vertical melt. Reviewr strongly suggests you consider doing this – it will make your model much more robust.

L 99 ". . . when the ice at that point melts AT THE SURFACE" – must clarify that this is only surface melt impacted.

L107 ". . . but up to its flexural wavelength. . . " please provide order of magnitude estimate of this and compare to melt pond size (e.g. 10 m) to show whether this assumption is supported.

L114 ". . . ponds can grow OR SHRINK as a result of freeboard CHANGES (delete sinking)." The freeboard changes work both ways, under certain circumstances L116 "Both rigid. . . and local. . . contribute to. . . sinking" This is inaccurate. Mass loss above sea level (local melting) contributes to freeboard rising in a relative sense.

L121 " assume no new ponds can form" You should clarify that this assumption only applies to this mode of growth and not to freeboard sinking as parameterized. As stated above strongly suggest making vertical sidewall melting actually based on a reduced albedo as a function of freeboard, which removes the need for this constraint altogether.

L128 ". . .not the horizontal EROSION (delete motion)"

L 132 "i) it is LIKELY near sea level"

L 139 ". . . pond perimeter melts slowly (delete enough) RELATIVE TO THAT FURTHER FROM THE PERIMETER" the rate is relative, not absolute and this wording was too qualitative anyhow.

L 146 How is mass above sea level determined? You have only height above sea level in the Hypsographic curve. You must use a density here as well. What is this density? (it should be lower than the density of ice below sea level)

L174-5 This parameterization entirely neglects light transmission, and assumes all the change in albedo is seen by the surface. In actuality much of the difference in energy ends up being in short wavelengths deposited deeper in the ice. This would be a substantial extra complication. Reviewer would be OK if you address this simply by acknowledging that the assumption all extra energy of a lower albedo is not actually deposited at the surface and that this is a first order approximation.

L 202 ". . . contribute to pond (delete growth) AREA CHANGE. (delete Since we assume new pons cannot form)" This constraint only applies to vertical sidewall melting, not to freeboard loss.

L211 "needs to equal one" This means that bare ice far away from the ponds must be greater than one if ice near ponds is less than one. The discussion here of some area being less than one, the rest = 1, and the overall average = 1 is not really sensible for any reasonably finite area. The reviewer feels the low albedo area subject to vertical sidewall melting is actually finite and of several to tens of percent of the surface. Therefore some adjustments are needed to this.

L225 You clarify this below on line 228, but suggest including here. . . " the average freeboard height OF THE BARE ICE FRACTION, h"

L240 " not important" This is only true so long s ponds GROW and k>1. Need to add some clarification of this.

L247 " we believe that such a statistical description. . . should be general. . . across

different ice types..., however" This statement is not supportable. At a minimum it must be deleted.

Reviewer thinks that the authors are sweeping what may be the single most important control over this process under the rug. Reviewer respects that authors many not want to tackle this now and leaves the option to authors as to whether to address the possibility of different hypsographic curves having impacts. Reviewer feels changing hypsographic curves would exert considerable influence over how ponds evolve late in the melt season as well as their overall coverage. Using this model to comment on this would be a very powerful expansion of this paper. Also, Hypsographic curves evolve during the melt season with ponded and unponded ice increasingly creating a bimodal distribution. This would change the relative impacts of the growth mechanisms... vertical sidewall growth may 'run out' of area to impact, for example. Since reviewer has revealed their identity to assist with review completion, reviewer is available to discuss this idea if authors choose to pursue it.

L 279 ".. only changes close to the pond boundary..." Actually this isn't quite true. Changes near the pond boundary will also impact overall freeboard, resulting in an average reduction of hypsographic curve height across all heights.

L283 " ... pond perimeter... is a function of pond fraction" Hmm. This is not the primary control. See Melt pond fractal dimension paper that came out of Ken Golden's group. If authors switch to treating vertical sidewall melting to be based on freeboard level as suggested above authors could avoid dealing with the significant complications in the relationship between pond area and perimeter.

L334 "decreases as the ponds deepen". This is only true by correlation, not causation. More accurately albedo decreases as the thickness of the underlying ice decreases (also as any snow ice on the surface is removed).

There are substantial issues with the next sentence. "We assume... " The difference in albedo of ponds caused by differences in radiative absorption in the water is negligible

compared to the impact of pond bottom albedo caused by changing ice thickness. This incorrect parameterization of pond albedo calls into question your conclusion about the small effect of pond albedo on pond evolution. Reviewer believes its likely this small effect will remain, but this parameterization of pond albedo is not supportable and must be revised.

L350 -360

Reviewer has some concerns about the use of this data, which is from multiyear ice, for a model exercise which is really only valid on FYI, where permeability becomes very large (MY ice ponds do not all drain to sea level). These ponds were not all permeable at the time they are discussed. The albedos here are high for first year ice and melt rates are low for FYI. In a way, the authors have created a sort of Frankenstein ice by combining FYI topography from Landy and MYI optical properties from Perovich. Would it be possible to do better using data from Polashenski et al 2012, Polashenski and Perovich 2012; and Landy et al 2014? There may also be some useful data from some of Marcel Nicolaus's papers.

L370 Suggest using Timco et al '96 to refine these densities. Seawater should be more like 1025 kg/m3 and sea ice more like 920 kg/m3 below sea level and 0.87 above (or lower if surface scattering layer is included).

L387 'grows ponds... melting, and freeboard sinking" Not true. The melting of bare ice grows ponds through vertical sidewall melting but SHRINKS ponds through freeboard rising.

L 389 Delete "... whereas freeboard sinking due to bare ice melting is independent of pond fraction" This is not accurate. The amount of freeboard sinking depends on how much area bare ice melting has to impact, which is dependent on pond fraction.

L 397 Might state why shrinking ponds can't be represented well. Unrealistic pond bottom topography in hypsographic curve. ' L 400 ice thickness of 2 m is very thick

[Figure]

FYI. These experiments would be better run on 1.5 m thick ice as a more typical FYI cover.

L403 Delete "in"

L 404 extra comma

L 410 ... be ignored, AS LONG AS THE HYPSOGRAPHIC CURVE INCLUDES SIGNIFICANT LOW TOPOGRAPHY AREAS"

L412 pond growth (ADD COMMA)

L419-425... this is Multiyear ice. SHEBA ponds are not first year ice, nor were they all permeable during the times the authors use for comparison. This poses considerable issues for this comparison. Suggest finding a FYI comparison in literature. If not, authors must acknowledge specifically and clearly that this is MYI, and not the same as what the model is set up for (FYI).

L428-432. This is all pretty optimistic. There is likely a significant disagreement between the hypsographic curve used by the authors and that of this MY ice. Must definitely state the possibility of such given the Frankenstein nature of this dataset.

Also " This difference in topography could explain the remaining discrepancy..." Yes, but its unlikely that it would completely explain it and many other factors are at play. Change to "This difference in topography could contribute to the remaining...

L445, add clarification... this is not a universal property of ice thickness, since changes in topography that may be related to ice thickness could impact. " meaning that... on thicker ice OF THE SAME SURFACE HYPSOGRAPHIC CURVE."

L 469 "Using ... we get..." This is also only true at a specific ice thickness and hypsographic curve. These should be specified as well to clarify that this is not a general result for all ice at this bottom melt flux.

L487-88 " contributes to roughly... both to... melting to..." Multiple split infinitive...

way hard to read.

490 – here is a great example of where the relative impact stated would be highly dependent on Hypsographic curve. In this particular case, it would seem also that vertical sidewall melting appears to be unimportant, the freeboard mechanism dominates. This additional sentence trying to justify vertical sidewall melting importance seems poorly supported. Reviewer thinks that vertical sidewall melting importance in some part of the regime would be clarified by use of differing hypsographic curbes.

L510 –this is an accurate conclusion and likely will stand after the revision, but for right now it is built on a very poor pond albedo parameterization which should be revised.

L525-540 This is an interesting model exercise. Reviewer will allow it here. The quantitative result, however, cannot be discussed outside this section (i.e. abstract and conclusion) unless the authors are able to produce reasonable estimates of the uncertainty. Reviewer feels the result is simply too sketchy for placing it in these areas without error estimates and risking a member of the modeling community pick it up as fact. Reviewer instead suggest discussion what sign these results indicate the melt pond change should have in a changing climate and highlighting this in the abstract and conclusion.

L 542 Start of section 5.1 is a non-sequitur from previous section, suggest a couple sentences to introduce why you are talking about this.

Section 5.1 is problematic in topic based on conclusions. Reviewer thinks it would be more relevant after treatment of differing hypsographic curves shows that sometimes vertical sidewall melt is important. Based on current results, which show dominance of freeboard loss. it struck the reviewer as disingenuous to discuss all these mechanisms and then admit at the end that the available evidence suggests none are important, only as an aside in the last sentence of the section. The discussion should focus on these rather than relict bits from prior version. An analysis of sensitivity to ice topography/roughnes/hypsographic curve would be very highly valuable to the community

here instead. The summary of the mechanisms from literature has some serious issues (detailed below). Authors are likely missing the strongest mechanism here as well. Albedo is strongly dependent on freeboard below about 20cm... therefore low topographic height areas will have lower albedo and will melt faster.

line 545 - thin layer of water... due to multiple light reflections. Reviewer thinks authors mean refractions and authors should clarify that the refraction occurs more strongly (and hence more backscatter occurs) when the ice is not covered by liquid and air occupies pore/intercrystalline space.

paragraph around line 550. Yes, this generally describes grain ripening, but this process is not significant for large scale 'jagged' features and 'smoothing' ice. It is a microstructural process, and one unlikely to be much more prevalent near ponds than away from them, therefore not a good candidate as described.

c- 557 - needs a reference, or replace 'often' with something softer like 'sometimes' and a personal communication.

L565 – Up to authors, but could be helpful to the reader to clarify why cloudy conditions have lesser effect (lower shortwave and more of surface radiative balance controlled by spatially uniform longwave absorption).

L 570 – 583. These mechanisms are possible, but not likely to be important. Reviewer suggests deleting and saving the words.

L575 – Frankly this seems unlikely

L580 - The ridge accelerated melt is largely fictitious. For every sun-ward side, there is a shaded side. Reviewer suggests deleting this or at least clarifying sunny/shady balance.

L599 – It is important to note that lateral melt does not depend on ice thickness... or ... IN THIS MODEL, THOUGH IT MAY DEPEND ON THESE IN NATURE TO SOME DEGREE. L 655 – C is the likely and observationally supported cause here. See estimates

of porosity in Polashenski et al., 2012. Suggest highlighting this option as more likely than the others. L667 through section 5.4 This section is fine, but would be stronger if it included discussion of how changes in topography (hypsographic curve) would impact pond growth. In this case, the stop growing case is dependent on very steep/vertical sidewalls which are common in rough ice. L709 " we also find. . ." this statement is true but authors need to fix up the pond albedo parameterization and make sure it holds L710 "Using the same.. " delete sentence, add error bars, or change to discussion of sign only L712 " The four modes. . . " This statement needs qualification to prevent over interpretation or mis interpretation, specifically indicating what terms (average initial pond coverage/thickness etc) are being held constant and which are changing. The next comment will help clarify the issue. L 713 " growth rate due to ponded ice and ice bottom melting increases with pond coverage" Reviewer thinks the results really indicate that growth rate due to these mechanisms increases with ice thinning. Starting at the same thickness with different pond coverage and the same average freeboard, bottom melting should case greater growth rate on lower pond coverage floes, because more of the floe surface will be closer to sea level.

---

## Author Comment (AC3) · 23 Feb 2017

**Response to reviewers**

Predrag Popovic[1] and Dorian Abbot[1]

[1]University of Chicago, Dept. of Geophysical sciences

*Correspondence to:* Predrag Popovic (ppopovic@uchicago.edu)

Dear reviewer and editor,

We found the reviewer's comments very helpful, and we have made several major adjustments to the manuscript which significantly improved the model. In particular, we have:

1. Changed the focus from processes that affect the melt rate near the pond perimeter to processes that depend on height above sea level. For this reason, we have removed the discussion of physical processes acting near the pond perimeter.

2. Discussed different hypsographic curves and included a parameterization of sea ice roughness.

3. Discussed the difference between first year ice and multiyear ice pond evolution.

10   Additionally, we have made many smaller modifications. Below we respond in detail to each of the reviewer's comments, and attach the revised manuscript. Thank you for your time and help improving this manuscript.

**Response to reviewer comments**

Thank you for your review! In response to your comments, we have substantially modified our paper.
15   In particular, we have:

1. Added a discussion on the effects of varying the hypsographic curve and ice roughness.

2. Removed the discussion about processes that influence the rate of melt near the pond perimeter (section 5.1).

3. Made a more careful comparison with observations.

20   In addition, we have changed the focus from processes that affect the melt rate near the pond perimeter to processes that depend on height above sea level. For this reason we have used the name "enhanced melting" instead of "vertical sidewall melting" in the paper and in this response. We respond (in black) in detail to all of your comments (in blue) below.

1) the statements suggesting that hypsographic curves are general across sea ice in line 240-
25   250 should be removed. It is likely that hypsographic curve of sea ice varies greatly due to many

influences on topography. Related discussion could suggest that this model would be made stronger by input of accurate curves.

**Response:** We have removed this sentence from the paper. Instead, now suggest that fitting the hypsographic curve with a tangent function that has two free parameters would likely be appropriate for many different hypsographic curves. We showed this by fitting the tangent function to a first year ice hypsographic curve measured by Landy et al, and to a multiyear ice hypsographic curve measured during SHEBA. On line 221, we now write

"*The above sea level part of every measured hypsographic curve we tested can be fit relatively well with a tangent function (Fig. 2a, red line). We will assume that this fit holds for a wide range of different sea ice floes, and use it to initialize our model with different physical parameters.*"

2) vertical sidewall melting appears to be unimportant, the freeboard mechanism dominates. That is the conclusion here. This additional sentence trying to justify vertical sidewall melting is unsupported. By itself this sentence would be innocent, but the first part of the discussion (5.1) which seems to be trying to focus on mechanisms of accelerated melt around pond edges really works this topic more. The extended discussion of vertical melt around pond edges doesn?t really seem to have any place here given the behavior of the model. Suggest removing entirely. If not... see below.

**Response:** Thank you for this comment. In response to the comments you made in the full review, we have modified the way we treat "enhanced melting" (equivalent of "vertical sidewall melting" in the new version of the paper). We now assume that instead of accelerated melt acting near the pond boundaries, the melt rate depends on the height above sea level. Furthermore, we now consider the effects of sea ice roughness on enhanced melting and freeboard sinking. It turns out that when these two effects are taken into account, enhanced melting becomes important, contributing to roughly half of the pond growth. We have removed section 5.1 entirely, and only kept a short discussion on mechanisms when introducing height-dependent enhanced melting. On line 263, we now write:

"*Next we need to consider the contribution from enhanced melting. Before doing so we need to make some assumptions about the nature of enhanced melt. There are multiple physical processes that can cause the melt rate to deviate from the mean. One process that stands out as being particularly important is albedo decrease due to ice wetting: ice close to sea level will likely be wet and therefore have a lower albedo compared to ice higher up. The deviation from the mean melt rate in this case depends primarily on the height above sea level. Another potential contribution to height-dependent enhanced melt may effectively come from random fluctuations in the melt rate around the average: ice near the sea level has a higher probability of falling below sea level due to random fluctuations than ice higher up. After falling below sea level, ice becomes ponded, melts faster, and is unable to return to its previous position. Other processes, such as lateral melt, may not depend on height above sea level, but for now we neglect this possibility (see section 7.1 for discussion).*"

3) Section 5.1 is problematic. It struck the reviewer as disingenuous to discuss all these mechanisms and then admit at the end that the available evidence suggests none are important, only as

an aside. The model suggests changes in freeboard, and though the authors gloss over it, changes in hypsographic curve are the important features. The discussion should focus on these rather than relict bits from prior version. An analysis of sensitivity to ice topography/roughness/hypsographic curve would be very highly valuable to the community here instead. There are also issues with the summary of the literature and applicability for several of the topics written as well.

**Response:** Thank you for this comment! We agree, and we have removed section 5.1 entirely. We added a section where we discuss the effects of ice roughness and different hypsographic curves. We also evaluate the difference between first year ice and multiyear ice topographies.

4) -SHEBA ponds are not first year ice, nor were they all permeable during the times the authors use for comparison. This creates several instances in needed of adjustment or revision.

**Response:** We now note this in section 3. The ponds did, however, drain to sea level at some point, which means that from that point on, our model should be appropriate. Since there is no obvious change in pond coverage trend in SHEBA ponds over the period we considered, we assume that even if they were not fully drained at the beginning of our comparison this had little effect on pond coverage evolution. We use a hypsographic curve measured during SHEBA when making the comparison, and we discuss the difference between first year ice and multiyear ice ponds.

5) The conclusion of 1.3 percent per month in a warming climate is way to shaky to be allowed in the abstract or conclusion as a strong take home point. There are numerous processes not accounted for here. Trumpeting a pond coverage trend in a warming climate based only on an idealized model of one of several pond formation processes is just too bold to pass peer review. The strong conclusions of this paper are with regard to the relative roles of various processes implied. This point may be presented only in the discussion as a possible outcome given particular assumptions, not as a main take home of the paper.

**Response:** We have removed the conclusion of 1.3 percent per month from the abstract and the conclusions. Instead, we only state that there would likely be a significant pond feedback in a warming climate. On line 15, we now write: "*
[revised manuscript text omitted]
_{\text{em}}/ds_{\text{fs}})^2 + (k-1)}{(1 + \frac{ds_{\text{em}}}{ds_{\text{fs}}})(k-1)} S_{\text{em}}^*\Big]\frac{F_{\text{sol}}}{\overline{F}_{\text{bi}}} \Delta \alpha_{\text{bi}} \approx -0.9 \frac{1}{\text{month}} \Delta \alpha_{\text{bi}} \,. \tag{30}$$

On the other hand, if the melt pond albedo changes by $\Delta \alpha_{\text{mp}}$, the change in growth rate would be roughly

560 $$\Delta S^* = -\Big[S_{\text{mp}}^* + \frac{(\rho_w - \rho_i)x_i(ds_{\text{em}}/ds_{\text{fs}})^2 \overline{F}_{\text{mp}}}{\rho_w(1 + \frac{ds_{\text{em}}}{ds_{\text{fs}}})(k-1)(1-x_i)\overline{F}_{\text{bi}}} S_{\text{em}}^*\Big]\frac{F_{\text{sol}}}{\overline{F}_{\text{mp}}} \Delta \alpha_{\text{mp}} \approx -0.2 \frac{1}{\text{month}} \Delta \alpha_{\text{mp}} \,. \tag{31}$$

It follows from these estimates that after a month the mean pond fraction would differ by roughly $4.5\%$ for a bare ice albedo difference of $0.1$, and by around $1\%$ for a pond albedo difference of $0.1$. Therefore, variation in pond albedo affects pond evolution roughly five times less than variation in bare ice albedo. This explains our observation from Fig. 5c that pond evolution is much more sensitive to variations in bare ice albedo than to variations in pond albedo. In this way, we also extract the dependence of sensitivity on physical parameters. A major difference between the two sensitivities is their dependence on the initial pond coverage: the sensitivity to pond albedo is proportional to $x_i$, whereas the sensitivity to bare ice albedo is proportional to $1 - x_i$. In the above example we used $x_i = 0.2$, which explains most of the large difference between the two sensitivities. If the pond coverage were higher, variations in the pond albedo could become more important than variations in bare ice albedo. For example, assuming no enhanced melting, the sensitivity to pond albedo would become greater than the sensitivity to bare ice albedo at $50\%$ pond coverage ($\frac{\Delta S_{\mathrm{mp}}^*}{\Delta S_{\mathrm{bi}}^*} = \frac{x_i}{1-x_i}\frac{\Delta\alpha_{\mathrm{mp}}}{\Delta\alpha_{\mathrm{bi}}}$).

**6.2 Under global warming, pond feedback could lead to significant ice thinning**

We now use the effective strengths to roughly estimate the impact of global warming on the pond coverage. At high latitudes, feedbacks due to changes in albedo, the atmospheric lapse rate, and clouds can amplify the forcing due to global warming (Holland and Bitz, 2006). For this reason forcing at high latitudes is generally larger than direct radiative forcing due to an increase in $CO_2$ concentration. In a global warming scenario, the pond growth rate would increase because the ice melts faster, but also because ice at the beginning of the melt would be thinner. We can emulate a global warming scenario by increasing the flux $F_r$ by a certain amount, $\Delta F_r$, and by assuming that the initial ice thickness decreases by $\Delta H \equiv \frac{\partial H}{\partial F_r}\Delta F_r$, where $\frac{\partial H}{\partial F_r}$ is the ice thinning per $1\ \mathrm{Wm}^{-2}$ of warming. Therefore, we split the change in pond growth rate due to global warming, $\Delta S^*$, into a contribution from direct forcing, $\Delta S_F^*$, and a contribution from ice thinning, $\Delta S_H^*$. Using the above formalism, we find

$$\Delta S_F^* \equiv \sum_i \frac{\partial S_i^*}{\partial F_r}\Delta F_r = \Big[\frac{S_{\mathrm{bi}}^*}{\overline{F}_{\mathrm{bi}}} + \frac{S_{\mathrm{mp}}^*}{\overline{F}_{\mathrm{mp}}} + \frac{\frac{\rho_w-\rho_i}{\rho_w}(ds_{\mathrm{em}}/ds_{\mathrm{fs}})^2 + (k-1)(1-x_i)}{(1+\frac{ds_{\mathrm{em}}}{ds_{\mathrm{fs}}})(k-1)(1-x_i)}\frac{S_{\mathrm{em}}^*}{\overline{F}_{\mathrm{bi}}}\Big]\Delta F_r \approx \frac{0.5\%}{\mathrm{W/m^2}\times\mathrm{
[revised manuscript text omitted]

---

## Author Comment (AC4) · 23 Feb 2017

**Response to reviewers**

Predrag Popovic[1] and Dorian Abbot[1]

[1]University of Chicago, Dept. of Geophysical sciences

*Correspondence to:* Predrag Popovic (ppopovic@uchicago.edu)

Dear reviewer and editor,

We found the reviewer's comments very helpful, and we have made several major adjustments to the manuscript which significantly improved the model. In particular, we have:

5    1. Changed the focus from processes that affect the melt rate near the pond perimeter to processes that depend on height above sea level. For this reason, we have removed the discussion of physical processes acting near the pond perimeter.

2. Discussed different hypsographic curves and included a parameterization of sea ice roughness.

3. Discussed the difference between first year ice and multiyear ice pond evolution.

10  Additionally, we have made many smaller modifications. Below we respond in detail to each of the reviewer's comments, and attach the revised manuscript. Thank you for your time and help improving this manuscript.

**Response to reviewer comments**

Thank you for the review, your comments were extremely useful! In response to your comments, we
15  have substantially modified our paper. In particular, we have:

1. Changed the focus from processes that affect the melt rate near the pond perimeter to height-dependent processes, thereby relaxing the restriction that new ponds cannot form.

2. Explored the effects of sea ice roughness.

3. Made a more rigorous comparison to observations, where we clearly state that observations
20    were made on multiyear ice. We have also discussed the difference in sea level pond evolution on FYI and on MYI.

We have removed or substantially changed many parts of the paper. For this reason, some of the specific comments you made no longer apply. Since we no longer focus on the processes acting near the pond perimeter, we change the name "vertical sidewall melting" to "enhanced melting." We
25  respond (in black) in detail to all of your comments (in blue) below.

1) Reviewer feels that the authors use of Multiyear ice dataset for comparison of a model that only tracks FYI processes is a significant problem. Authors must either change out comparison data to a FYI dataset or place clear statements throughout the text indicating the severe limitations of comparison to field observations that amount to a Frankenstein combination of FYI (e.g. Landy Hypsographic curve) and MYI (Perovich Albedos).

**Response:** Thank you for pointing this out! We have changed the "default" pond albedo from 0.4 to 0.2 to correspond to typical FYI values. When comparing our model to observations, we stated that the measurements were made on level MYI but that the ponds drained to sea level, which means that our model is appropriate. On line 380, we now write:

"*Ice along the albedo line was level multiyear ice, but the ponds drained to sea level after some time which makes them amenable to our model (Perovich et. al., 2003).*"

In addition, we now use a MYI topography measured along a topography profile during SHEBA. We found that ponds on multiyear ice grow almost entirely due to enhanced melting.

2) Reviewer feels the authors miss a substantial opportunity to comment on the impact that differing hypsographic curves would have on the ice. Reviewer suggests that vertical sidewall melting would be primarily controlled by height above sea level of the area.

**Response:** This was a very useful comment as well. We added a section where we discussed the effects of sea ice roughness (section 5). We represented a hypsographic curve with a "tangent function" and varied the parameters of this function to get hypsographic curves with different shapes and roughness. We have developed a parameterization that approximately captures the effects of roughness for freeboard sinking and enhanced melting and allows us to estimate mean pond coverage after a period of time without having to run the model. We show that, on ice with low roughness relative to the mean freeboard height, pond growth by freeboard sinking is suppressed while pond growth by enhanced melting is permitted (with lateral melting likely playing an important role). We discuss the difference between FYI and MYI pond evolution. On line 530, we write

"*Multiyear ice topography shown in Fig. 2a, dashed line, has $\hat{\sigma} \approx 0.25$ and is significantly smoother than first year ice topography shown in Fig. 2a, solid line, which has $\hat{\sigma} \approx 0.55$. From Eq. (28) it follows that freeboard sinking on multiyear ice is roughly 5 times less efficient in growing the ponds than on first year ice.*"

3) The conclusion of 1.3 percent per month in a warming climate is way to shaky to be allowed in the abstract or conclusion as a strong take home point. There are numerous processes not accounted for here. Trumpeting a pond coverage trend in a warming climate based only on an idealized model of one of several pond formation processes is just too bold to pass peer review. Authors have two options on this point. 1. Produce and include estimates of uncertainty with this number, 2. Drop the specific quantification and discuss the sign of the change.

**Response:** We have removed the exact number of $1.3\%$ per month per $\text{Wm}^{-2}$ from the abstract and the conclusion, but stated that ice thinning due pond feedback is likely of a similar order of magnitude as ice thinning due to direct forcing. In the abstract, on line 15, we write:

"*We also show that under a global warming scenario, pond coverage would increase, decreasing the overall ice albedo, and leading to ice thinning that is likely comparable to thinning due to direct forcing.*"

In the conclusions, on line 819, we write:

"*Under a global warming scenario, the pond feedback could lead to ice thinning comparable to thinning due to direct forcing.*"

4) Conclusions need to do a better job clarifying over what conditions the general interactions between melt location and pond coverage apply. Reviewer feels the model results only show how these are true for certain pond coverage and ice thickness pathways, dependent on the key initial assumptions, such as that ice all has the same hypsographic curve. As an example the conclusion that bottom melting's impact on growth rate increases with increasing pond coverage may not hold if comparing two floes with very different pond coverage and different topography. One floe may have very flat ice and low pond coverage at the time ponds drain to sea level, while another may have higher pond coverage and greater topography. Bottom melt will impact the level ice floe more than the rougher flow because of having more ice near sea level to flood, regardless of the initial pond coverage.

**Response:** We feel that some confusion may have arisen from discussing the dependence of different modes of growth on pond coverage. For this reason, we removed this from the conclusions. We decided against stating many caveats in the conclusions as we believe it would derail the reader from the main findings of our model. Nevertheless, we have rewritten the conclusion section, hopefully in a clearer way.

5) Clarify that this is all about FY ice. . . As the melt season progresses FIRST YEAR sea ice (delete floes) in the Arctic often become(S) permeable...

**Response:** Our model can be used whenever ponds are drained to sea level, regardless of whether this is on FYI or MYI. As we discuss the evolution of drained ponds on MYI later in the paper, we did not state that the model is only about FYI in the abstract. We deleted the word "floes."

6) Clarify that it is the surface relation to sea level that is of concern . . .and correspond to regions of the sea ice WHOSE SURFACE IS (delete that are) below sea level. . .

**Response:** Done

7) "relatively well" - relative to what? Can you replace this with a quantitative statement rather than this highly subjective one.

**Response:** We deleted "relatively" in the abstract. Later in the text gave a quantitative assessment. On line 403, we now write

*"The agreement between model and observation is excellent, with a maximum discrepancy of* $3\%$
*pond coverage at the end of the melt season."*

8) Suggest replacing "demonstrates the somewhat surprising result" with simply "indicates". I'm
not surprised by this! Also see comments later about some issues with the support for this statement.

**Response:** Done!

9) "Performing a similar. . . 1.3%... per month." This statement is too sketchy for the abstract.
This quantitative conclusion is not something that a reader could use as a takeaway number from
this paper. Reviewer could permit discussion of the <sign> of this impact here, but if the authors
would like to quantify it specifically outside the context of the section where this comes from in the
discussion, then error bars will need to be included. Reviewer expects these would be very large and
challenging to quantify.

**Response:** We removed $1.3\%$ per $\mathrm{Wm}^{-2}$ per month, and instead write (line 15):

*"We also show that under a global warming scenario, pond coverage would increase, decreasing
the overall ice albedo, and leading to ice thinning that is likely comparable to thinning due to direct
forcing."*

10) "Typical. . . proceeds in (delete a) fairly consistent STAGES (delete manner)." The stages are
predictable, and regular. Their duration is not.

**Response:** Done

11) You can actually be clearer here. The stage when ice. . . is typically the longest, OFTEN
LONGER THAN THE TWO FIRST STAGES COMBINED.

**Response:** Done

12) "we analyze the model RESULTS analytically" (I think its more the results than the model
you analyze)

**Response:** We disagree: we analyze both the model equations and the solutions to those equations.

13) ". . . and (delete all the meltwater created is) MELTWATER CAN BE." Need to be clear here
that meltwater is not necessarily all removed, just that it can be removed.

**Response:** Done

14) ". . .when (delete: pond coverage drops to its summer minimum, meaning that all) the melt-
water on the ice. . . " It is not necessarily true that the coverage is it its minimum at this point, though
it is likely. Better to just leave this point defined by water level and sea level relationship as the
remainder of the sentence does

**Response:** Done

15) " . . .which were above sea level to sink OR MELT below sea level". . . You allow for vertical
melt to cause creation of below sea level surface.

**Response:** Done

16) " . . . at the surface of the ice". Need to clarify that this is at EITHER surface, top or bottom.
Suggest ". . . at the surface or bottom of the ice" for clarity.

**Response:** Done

17) " we will further assume. . . " You do not follow this. New ponds can form by flooding isolated areas by freeboard loss based on your use of the hypsographic curve. This clarifying constraint applies only to your vertical melt enhancement around pond perimeters. If you were to change the vertical melt enhancement to be based on surface height above freeboard, which is probably its strongest dependence, you could remove this weakness altogether and freely permit new ponds to form in low topography areas by both freeboard loss and accelerated vertical melt. Reviewr strongly suggests you consider doing this - it will make your model much more robust.

**Response:** We followed up on this. Instead of focusing on melt near the perimeter, we introduced a height-dependent enhanced melting, thereby removing the constraint on new pond formation. This required that the strength of enhanced melting be modified, and changed the scaling of enhanced melting with physical parameters. In order to derive the strength of enhanced melting, we needed to construct an additional model which we now describe in appendix B.

18) ". . . when the ice at that point melts AT THE SURFACE" - must clarify that this is only surface melt impacted.

**Response:** Done

19) ". . . but up to its flexural wavelength. . . " please provide order of magnitude estimate of this and compare to melt pond size (e.g. 10 m) to show whether this assumption is supported.

**Response:** On line 109, we now write:

"*An ice floe is not a rigid body, but up to its flexural wavelength (roughly 30m on 1.5m thick ice) we can approximate it as such. As the flexural wavelength is larger than the typical scale of melt ponds (roughly 10m), rigid body approximation is likely good.*"

20) ". . . ponds can grow OR SHRINK as a result of freeboard CHANGES (delete sinking)." The freeboard changes work both ways, under certain circumstances. "Both rigid. . . and local. . . contribute to. . . sinking" This is inaccurate. Mass loss above sea level (local melting) contributes to freeboard rising in a relative sense.

**Response:** This is not true. We make a distinction between "freeboard sinking" and "rigid body motion." What the reviewer is describing here is actually rigid body motion which can be both upward and downward depending on whether mass is lost above or below sea level. Freeboard sinking, however, represents the average downward shift of the ice surface (the surface has to move downward on average, since the ice is thinning). It has a contribution from both the rigid body motion and the average local melt. Therefore, melting bare ice will induce an upward rigid body motion and a downward local melt resulting in a downward shift of the ice surface on average. In order to clarify this, we added on line 117:

"*Freeboard sinking should not be confused with rigid body motion: the average freeboard height always decreases as a response to ice thinning, whereas the rigid body motion can point both upward*

170  *and downward depending on whether mass is lost above or below sea level. Both rigid body motion and average local melting contribute to freeboard sinking.*"

We also added a figure that explains our model (Figure 3).

21) " assume no new ponds can form" You should clarify that this assumption only applies to this mode of growth and not to freeboard sinking as parameterized. As stated above strongly suggest

175  making vertical sidewall melting actually based on a reduced albedo as a function of freeboard, which removes the need for this constraint altogether.

**Response:** Now we allow for new pond formation, so we removed this sentence entirely.

22) ". . .not the horizontal EROSION (delete motion)"

**Response:** We removed this sentence from the paper.

180  23) "i) it is LIKELY near sea level"

**Response:** We removed this sentence from the paper.

24) ". . . pond perimeter melts slowly (delete enough) RELATIVE TO THAT FURTHER FROM THE PERIMETER" the rate is relative, not absolute and this wording was too qualitative anyhow.

**Response:** We removed this sentence from the paper.

185  25) How is mass above sea level determined? You have only height above sea level in the Hypsographic curve. You must use a density here as well. What is this density? (it should be lower than the density of ice below sea level)

**Response:** We made a distinction between pure ice density ($\rho_i$) and bulk ice density ($\rho_b$) in the text. The density used to determine the mass above sea level is the density of pure ice, $\rho_i =$

190  $916 \ \mathrm{kgm^{-3}}$. On line 168, we now write:

"*$\rho_b$ is the bulk ice density. This is the density of sea ice once all the brine has drained and is always less than $\rho_i$.*"

We assumed a uniform bulk ice density, and discussed the effects of vertical variations in bulk density together with the effects of internal melt in section 7.2 and appendix D.

195  26) This parameterization entirely neglects light transmission, and assumes all the change in albedo is seen by the surface. In actuality much of the difference in energy ends up being in short wavelengths deposited deeper in the ice. This would be a substantial extra complication. Reviewer would be OK if you address this simply by acknowledging that the assumption all extra energy of a lower albedo is not actually deposited at the surface and that this is a first order approximation.

200  **Response:** On line 172, we now write:

"*This parameterization neglects light transmission, and assumes that all of the energy is deposited in the surface. Much of the variation in albedo of ponded ice is due to the fact that the pond bottom is partially transparent, and energy is deposited in the ocean instead of directly in the ice. However, this does not make much difference in our model since the energy deposited in the ocean is likely*

205  *used for melting ice below sea level anyway.*"

27) ". . . contribute to pond (delete growth) AREA CHANGE. (delete Since we assume new pons cannot form)" This constraint only applies to vertical sidewall melting, not to freeboard loss.

**Response:** We removed this sentence from the paper.

28) "needs to equal one" This means that bare ice far away from the ponds must be greater than one if ice near ponds is less than one. The discussion here of some area being less than one, the rest = 1, and the overall average = 1 is not really sensible for any reasonably finite area. The reviewer feels the low albedo area subject to vertical sidewall melting is actually finite and of several to tens of percent of the surface. Therefore some adjustments are needed to this.

**Response:** Although it is true that it impossible some part of the ice to have $k < 1$, the rest to have $k = 1$, and $< k >= 1$, it is approximately true if the fraction of ice with $k < 1$ is small. Making this distinction would likely add little to the model, but would significantly complicate it. For this reason, we decided against changing this assumption. We write on line 292:

"*We note that the assumption that $k(\boldsymbol{r}) = 1$ high above the sea level and $k(\boldsymbol{r}) > 1$ near the sea level is strictly not true since averaged over all of bare ice $k(\boldsymbol{r})$ needs to equal one. However, it is approximately true if $\Delta s$ or $\delta$ are small, such that the area where $k(\boldsymbol{r}) \neq 1$ is small compared to the total area of bare ice.*"

29) You clarify this below on line 228, but suggest including here. . . "the average freeboard height OF THE BARE ICE FRACTION, h"

**Response:** Done

30) "not important" This is only true so long s ponds GROW and k>1. Need to add some clarification of this.

**Response:** On line 226, we now write:

"*The topography below sea level is not important for the evolution of pond coverage if the pond coverage grows, ...*"

31) "we believe that such a statistical description. . . should be general. . . across different ice types. . ., however" This statement is not supportable. At a minimum it must be deleted.

**Response:** We deleted this sentence. Instead, we used a "tangent function" to fit the hypsographic curve. The function has two free parameters that determine its exact shape, and we were able to fit all of several measured hypsographic curves with it. On line 221, we now write:

"*The above sea level part of every measured hypsographic curve we tested can be fit relatively well with a tangent function (Fig. 2a, red line). We will assume that this fit holds for a wide range of different sea ice floes, and use it to initialize our model with different physical parameters.*"

32) Reviewer thinks that the authors are sweeping what may be the single most important control over this process under the rug. Reviewer respects that authors many not want to tackle this now and leaves the option to authors as to whether to address the possibility of different hypsographic curves having impacts. Reviewer feels changing hypsographic curves would exert considerable influence over how ponds evolve late in the melt season as well as their overall coverage. Using this

model to comment on this would be a very powerful expansion of this paper. Also, Hypsographic curves evolve during the melt season with ponded and unponded ice increasingly creating a bimodal distribution. This would change the relative impacts of the growth mechanisms... vertical sidewall growth may "run out" of area to impact, for example. Since reviewer has revealed their identity to assist with review completion, reviewer is available to discuss this idea if authors choose to pursue it.

**Response:** Thank you for this comment, and for helpful discussions over the phone. We took your advice and committed a section to examining the effects of the hypsographic curve shape (section 5). We defined the non-dimensional bare ice roughness, $\hat{\sigma} \equiv \frac{\sigma}{h}$, as the standard deviation of the bare ice height above sea level to mean freeboard height, and showed that ponds grow slower on ice with lower $\hat{\sigma}$. We also showed that freeboard sinking gets suppressed if the roughness is low, while enhanced melting does not. In this way, we showed that ponds on MYI grow mainly due to enhanced or lateral melting. We also developed a parameterization to estimate the mean pond coverage after a period of time if $\hat{\sigma}$ is known.

33) ".. only changes close to the pond boundary..." Actually this isn't quite true. Changes near the pond boundary will also impact overall freeboard, resulting in an average reduction of hypsographic curve height across all heights.

**Response:** We removed this sentence from the paper.

34) "... pond perimeter... is a function of pond fraction" Hmm. This is not the primary control. See Melt pond fractal dimension paper that came out of Ken Golden's group. If authors switch to treating vertical sidewall melting to be based on freeboard level as suggested above authors could avoid dealing with the significant complications in the relationship between pond area and perimeter.

**Response:** We do not deal with "vertical sidewall melting" anymore.

35) "decreases as the ponds deepen". This is only true by correlation, not causation. More accurately albedo decreases as the thickness of the underlying ice decreases (also as any snow ice on the surface is removed). There are substantial issues with the next sentence. "We assume... " The difference in albedo of ponds caused by differences in radiative absorption in the water is negligible compared to the impact of pond bottom albedo caused by changing ice thickness. This incorrect parameterization of pond albedo calls into question your conclusion about the small effect of pond albedo on pond evolution. Reviewer believes its likely this small effect will remain, but this parameterization of pond albedo is not supportable and must be revised.

**Response:** Thank you for this comment. We now use a parameterization of pond albedo with depth developed by Morassutti and Ledrew (1996). It turns out there is even less dependence on pond depth now, since, according to their parameterization, the albedo becomes independent of depth when ponds become deep enough. The initial pond depth we use (15cm) is already close to this saturation depth, making the pond evolution independent on pond albedo variations.

36) Reviewer has some concerns about the use of this data, which is from multiyear ice, for a model exercise which is really only valid on FYI, where permeability becomes very large (MY ice ponds do not all drain to sea level). These ponds were not all permeable at the time they are discussed. The albedos here are high for first year ice and melt rates are low for FYI. In a way, the authors have created a sort of Frankenstein ice by combining FYI topography from Landy and MYI optical properties from Perovich. Would it be possible to do better using data from Polashenski et al 2012, Polashenski and Perovich 2012; and Landy et al 2014? There may also be some useful data from some of Marcel Nicolaus's papers.

**Response:** We now use a default pond albedo of 0.2 and bare ice albedo of 0.55.

37) Suggest using Timco et al '96 to refine these densities. Seawater should be more like 1025 kg/m3 and sea ice more like 920 kg/m3 below sea level and 0.87 above (or lower if surface scattering layer is included).

**Response:** We corrected the model to include a uniform bulk ice density different than the density of pure ice. The density of interest is the bulk ice density once all the brine has drained from the ice. For this reason it is always less than the density of pure ice. Many of the values in Timco et al suggest a large difference in bulk density above and below sea level, but this is largely due to ice below sea level being saturated with water. Once all the brine drains, the difference between ice above and below sea level is significantly less. Nevertheless, we discuss the potential effects of vertical variations in bulk density in section 7.2. We find that, if considered together with internal melt, the vertical structure of bulk ice density can likely be neglected.

38) "grows ponds. . . melting, and freeboard sinking" Not true. The melting of bare ice grows ponds through vertical sidewall melting but SHRINKS ponds through freeboard rising.

**Response:** This is not true. Freeboard (the average surface height) always sinks if the ice melts and there is no internal melting. Bare ice melting contributes to freeboard sinking since it removes a layer of the the ice surface (this is what we call "local melting") in addition to inducing and upward "rigid body motion." On average, local melting is stronger than rigid body motion if there is no internal melt.

39) Delete ". . . whereas freeboard sinking due to bare ice melting is independent of pond fraction" This is not accurate. The amount of freeboard sinking depends on how much area bare ice melting has to impact, which is dependent on pond fraction.

**Response:** We changed this to (line 461)

"*Furthermore, when pond coverage is low, rigid body motion due to ponded ice melting is less efficient than that due to bare ice melting because it is proportional to melt pond fraction.*"

40) Might state why shrinking ponds can't be represented well. Unrealistic pond bottom topography in hypsographic curve.

**Response:** The reason shrinking ponds cannot be represented well is that the equation for enhanced meting fails in that case. On line 473, we now write:

*"The evolution of such a pond coverage cannot be represented well in our model since the equation for enhanced melting becomes invalid in this case."*

41) ice thickness of 2 m is very thick FYI. These experiments would be better run on 1.5 m thick ice as a more typical FYI cover.

**Response:** We now use a default ice thickness of 1.5m.

42) Delete "in"

**Response:** We removed this sentence from the paper.

43) extra comma

**Response:** We removed this sentence from the paper.

44) . . . be ignored, AS LONG AS THE HYPSOGRAPHIC CURVE INCLUDES SIGNIFICANT LOW TOPOGRAPHY AREAS

**Response:** A conclusion that the complicated physics of lateral melting can be ignored does not depend on our choice of a hypsographic curve. We showed here that the lateral melt flux, $F_{\text{lat}}$, that is in general difficult to find, can be related with a single constant to a bare ice melting flux, $F_{\text{bi}}$, which is easily found using shortwave, longwave, sensible, and latent heat fluxes. In the new version of the paper we found that a slightly better agreement with Skyllingstad et. al. (2009) is made by relating $F_{\text{lat}}$ with the ponded ice melting flux, $F_{\text{mp}}$. We have reorganized the discussion on lateral melting, hopefully making it more clear.

45) pond growth (ADD COMMA)

**Response:** We removed this sentence from the paper.

46) . . this is Multiyear ice. SHEBA ponds are not first year ice, nor were they all permeable during the times the authors use for comparison. This poses considerable issues for this comparison. Suggest finding a FYI comparison in literature. If not, authors must acknowledge specifically and clearly that this is MYI, and not the same as what the model is set up for (FYI).

**Response:** Thank you for pointing this out. We were not able to find adequate FYI data, so we used the same data. Although these ponds were on MYI, at some point they drained to sea level (Perovich et al 2003), after which our model is applicable. Even if they were not completely drained at the time beginning of our comparison, this does not seem to have affected pond evolution significantly as there is a roughly constant pond coverage trend throughout the comparison period. We now write on line 380:

*"Ice along the albedo line was level multiyear ice, but the ponds drained to sea level after some time which makes them amenable to our model (Perovich et. al., 2003)."*

47) This is all pretty optimistic. There is likely a significant disagreement between the hypsographic curve used by the authors and that of this MY ice. Must definitely state the possibility of such given the Frankenstein nature of this dataset. Also " This difference in topography could explain the remaining discrepancy. . ." Yes, but its unlikely that it would completely explain it and

**Response:** In section 3, we now make a more careful comparison with the observations. We show that the pond coverage trend along a nearby topography line is very similar to the albedo line, so we decided to use the hypsographic curve and ice thickness measured along the topography line to initialize our model. We treated the strength of enhanced melting as a fitting parameter. We obtained a better fit than before, with a discrepancy in pond coverage at the end of melt season of around 3% as opposed to 7%, which we had before. Also we showed that pond growth is almost entirely due to enhanced melting, and we attributed this to strong lateral melting.

48) add clarification. . . this is not a universal property of ice thickness, since changes in topography that may be related to ice thickness could impact. " meaning that. . . on thicker ice OF THE SAME SURFACE HYPSOGRAPHIC CURVE."

**Response:** We now write on line 614:

"*meaning that, all else equal, ponds grow proportionally slower on thicker ice.*"

49) "Using . . . we get. . ." This is also only true at a specific ice thickness and hypsographic curve. These should be specified as well to clarify that this is not a general result for all ice at this bottom melt flux.

**Response:** We now write on line 646

"*Using $\overline{F}_{bot} = 20 \ Wm^{-2}$ and other parameters same as above, we get.*"

50) " contributes to roughly. . . both to. . . melting to. . ." Multiple split infinitive. . . way hard to read.

**Response:** We now write on line 689

"*Using the above values of strengths, we find that after a month of growth bare ice melting contributes to roughly 25% of mean pond coverage, ponded ice melting contributes to around 13%, ice bottom melting contributes to around 7%, and enhanced melting contributes to roughly 55%.*"

51) here is a great example of where the relative impact stated would be highly dependent on Hypsographic curve. In this particular case, it would seem also that vertical sidewall melting appears to be unimportant, the freeboard mechanism dominates. This additional sentence trying to justify vertical sidewall melting importance seems poorly supported. Reviewer thinks that vertical sidewall melting importance in some part of the regime would be clarified by use of differing hypsographic curves.

**Response:** We removed this sentence from the text. Interestingly, however, it turns out that including a height-dependence instead of distance from the pond edge and quantifying the effects of ice roughness makes enhanced melting important. It contributes roughly the same to mean pond coverage as freeboard sinking on FYI and dominates on MYI.

52) this is an accurate conclusion and likely will stand after the revision, but for right now it is built on a very poor pond albedo parameterization which should be revised.

**Response:** We revised the albedo parameterization, ice thickness, and enhanced melting, but found that bare ice albedo is still contributes roughly 5 times more than pond albedo if the initial pond coverage is the same ($x_i = 0.2$).

53) This is an interesting model exercise. Reviewer will allow it here. The quantitative result, however, cannot be discussed outside this section (i.e. abstract and conclusion) unless the authors are able to produce reasonable estimates of the uncertainty. Reviewer feels the result is simply too sketchy for placing it in these areas without error estimates and risking a member of the modeling community pick it up as fact. Reviewer instead suggest discussion what sign these results indicate the melt pond change should have in a changing climate and highlighting this in the abstract and conclusion.

**Response:** We removed the specific number from the abstract and the conclusions, and now only state that the pond albedo feedback is important and its contribution to ice thinning is likely of a similar order of magnitude as contribution from direct forcing.

54) Start of section 5.1 is a non-sequitur from previous section, suggest a couple sentences to introduce why you are talking about this.

**Response:** We removed this section from the paper entirely.

55) Section 5.1 is problematic in topic based on conclusions. Reviewer thinks it would be more relevant after treatment of differing hypsographic curves shows that sometimes vertical sidewall melt is important. Based on current results, which show dominance of freeboard loss. it struck the reviewer as disingenuous to discuss all these mechanisms and then admit at the end that the available evidence suggests none are important, only as an aside in the last sentence of the section. The discussion should focus on these rather than relict bits from prior version. An analysis of sensitivity to ice topography/roughness/hypsographic curve would be very highly valuable to the community here instead. The summary of the mechanisms from literature has some serious issues (detailed below). Authors are likely missing the strongest mechanism here as well. Albedo is strongly dependent on freeboard below about 20cm. . . therefore low topographic height areas will have lower albedo and will melt faster.

**Response:** We agree, and we removed this section entirely. We only briefly discuss the dependence of albedo on height above sea level. On line 264, we now write:

"*There are multiple physical processes that can cause the melt rate to deviate from the mean. One process that stands out as being particularly important is albedo decrease due to ice wetting: ice close to sea level will likely be wet and therefore have a lower albedo compared to ice higher up. The deviation from the mean melt rate in this case depends primarily on the height above sea level. Another potential contribution to height- dependent enhanced melt may effectively come from random fluctuations in the melt rate around the average: ice near the sea level has a higher probability of falling below sea level due to random fluctuations than ice higher up. After falling below sea level, ice becomes ponded, melts faster, and is unable to return to its previous position.*"

56) It is important to note that lateral melt does not depend on ice thickness. . . or . . . IN THIS MODEL, THOUGH IT MAY DEPEND ON THESE IN NATURE TO SOME DEGREE.

**Response:** Done

57) C is the likely and observationally supported cause here. See estimates of porosity in Polashenski et al., 2012. Suggest highlighting this option as more likely than the others.

**Response:** We removed this sentence from the text. We significantly revised this section to consider the effects of vertical density structure together with internal melt. We found that these effects likely cancel each other out to a significant degree.

58) through section 5.4 This section is fine, but would be stronger if it included discussion of how changes in topography (hypsographic curve) would impact pond growth. In this case, the stop growing case is dependent on very steep/vertical sidewalls which are common in rough ice

**Response:** We shortened the discussion about pond growth stopping, as the paper was getting too long. Now we only mention the possibility of pond growth stopping due to upward rigid body motion temporarily induced by internal melting. On line 780, we now write:

"*Here, we will entertain the possibility of pond growth by vertical motion of the topography stopping entirely for a period of time. This is an example of a possible transient effect of internal melting, which, although interesting, seems unlikely.*"

Other mechanisms for slowing down pond growth, such as steep topography, are now discussed in sections 3 and 5.

59) " we also find. . ." this statement is true but authors need to fix up the pond albedo parameterization and make sure it holds

**Response:** We fixed the pond parameterization, and found a similar result.

60) "Using the same.. " delete sentence, add error bars, or change to discussion of sign only

**Response:** We removed the exact number, and wrote only (line 819)

"*Under a global warming scenario, the pond feedback could lead to ice thinning comparable to thinning due to direct forcing.*"

61) " The four modes. . . " This statement needs qualification to prevent over interpretation or mis interpretation, specifically indicating what terms (average initial pond coverage/thickness etc) are being held constant and which are changing. The next comment will help clarify the issue.

[revised manuscript text omitted]